# Nardilysin-regulated scission mechanism activates polo-like kinase 3 to suppress the development of pancreatic cancer

Jie Fu [1,10] ✉, Jianhua Ling [1,10], Ching-Fei Li [1], Chi-Lin Tsai [1], Wenjuan Yin[1], Junwei Hou [1], Ping Chen[1], Yu Cao[1], Ya'an Kang[2], Yichen Sun[1], Xianghou Xia [1], Zhou Jiang [1], Kenei Furukawa[1], Yu Lu[1], Min Wu[1], Qian Huang[1], Jun Yao[1], David H. Hawke[3], Bih-Fang Pan[3], Jun Zhao[4], Jiaxing Huang[1], Huamin Wang [4,5], E. I. Mustapha Bahassi[6], Peter J. Stambrook[6,11], Peng Huang[3,7], Jason B. Fleming[2,8], Anirban Maitra [4,5], John A. Tainer [1], Mien-Chie Hung [1,9], Chunru Lin [1,5] ✉ & Paul J. Chiao [1,5] ✉

Pancreatic ductal adenocarcinoma (PDAC) develops through step-wise genetic and molecular alterations including Kras mutation and inactivation of various apoptotic pathways. Here, we find that development of apoptotic resistance and metastasis of *Kras$^{G12D}$*-driven PDAC in mice is accelerated by deleting *Plk3*, explaining the often-reduced Plk3 expression in human PDAC. Importantly, a 41-kDa Plk3 (p41Plk3) that contains the entire kinase domain at the N-terminus (1-353 aa) is activated by scission of the precursor p72Plk3 at Arg354 by metalloendopeptidase nardilysin (NRDC), and the resulting p32Plk3 C-terminal Polo-box domain (PBD) is removed by proteasome degradation, preventing the inhibition of p41Plk3 by PBD. We find that p41Plk3 is the activated form of Plk3 that regulates a feed-forward mechanism to promote apoptosis and suppress PDAC and metastasis. p41Plk3 phosphorylates c-Fos on Thr164, which in turn induces expression of Plk3 and pro-apoptotic genes. These findings uncover an NRDC-regulated post-translational mechanism that activates Plk3, establishing a prototypic regulation by scission mechanism.

PDAC is projected to surpass breast, colorectal, and prostate cancer in incidence to become the second leading cause of cancer-related deaths by 2030[1,2]. Genetic analysis and gene expression profiling of PDAC identified several common mutations and molecular alterations, such as mutational activation of Kras, inactivation of Ink4a/ARF, and NF-κB activation[3–5]. Modeling PDAC with mutant Kras and deleting Ink4a/ARF, IKK2/β, or p53 in genetically engineered mouse models unequivocally demonstrated that these alterations are required for

[1]Department of Molecular and Cellular Oncology, The University of Texas MD Anderson Cancer Center, Houston, TX 77030, USA. [2]Department of Surgical Oncology, The University of Texas MD Anderson Cancer Center, Houston, TX 77030, USA. [3]Department of Systems Biology, The University of Texas MD Anderson Cancer Center, Houston, TX 77030, USA. [4]Department of Pathology, The University of Texas MD Anderson Cancer Center, Houston, TX 77030, USA. [5]Cancer Biology Program, The University of Texas MD Anderson UTHealth Graduate School of Biomedical Sciences, Houston, TX 77030, USA. [6]Department of Molecular Genetics, University of Cincinnati Cancer Institute, Cincinnati, OH 45267, USA. [7]Present address: State Key Laboratory of Oncology in South China, Collaborative Innovation Center for Cancer Medicine, Sun Yat-sen University Cancer Center, Guangzhou, Guangdong 510060, China. [8]Present address: Department of Gastrointestinal Oncology, Moffitt Cancer Center, Tampa, FL 33612, USA. [9]Present address: Graduate Institute of Biomedical Sciences, Institute of Biochemistry and Molecular Biology, Research Center for Cancer Biology, Cancer Biology and Precision Therapeutics Center, and Center for Molecular Medicine, China Medical University, Taichung 406, Taiwan. [10]These authors contributed equally: Jie Fu, Jianhua Ling. [11]Deceased: Peter J. Stambrook. ✉e-mail: jfu3@mdanderson.org; clin2@mdanderson.org; pjchiao@mdanderson.org

mutant Kras to induce PDAC[6–12]. Alterations in these and other onco-genic signals disrupt the regulation of apoptosis, leading to cancer progression and metastasis[4,13,14]. PDAC cells gain the capacity to resist apoptosis, which contributes to the development of a more aggressive tumorigenic and metastatic phenotype in vivo[15,16]. Our previous finding showed that inhibition of Polo-like kinase 3 (Plk3) expression by siRNA suppressed superoxide-mediated apoptosis in PDAC cells, suggesting Plk3 may have a regulatory role in induction of apoptosis[17]. We recently demonstrated that the apoptosis resistance of PDAC in a $Kras^{G12D}$-dri-ven mouse model was accelerated by deleting $Plk3$, consistent with the observation of often-reduced Plk3 expression in human PDAC, and suggesting that mutational activation of the Kras oncogene and inac-tivation of the Plk3 tumor suppressor gene are defects leading to resistance to apoptosis. However, the regulatory mechanism of Plk3 activation remained to be discovered.

Plk3 is a member of the Polo-like kinase family and plays pivotal roles in the regulation of cell-cycle progression and apoptosis[18–20]. However, little is known about Plk3's activation process or its regula-tion of cell proliferation and programmed cell death. In contrast, several hypotheses for the regulation of Plk1 activation have been proposed[21–24]. For example, a study of Plk1 revealed that the Polo-box domain (PBD) altered its conformation upon binding phosphopep-tides, illuminating possible activation mechanisms of Plk1 by phos-phorylation or phosphopeptide binding[25]. Yet, Perez et al. recently showed that phosphorylation at Thr219 in the T-loop does not increase Plk3 kinase activity, suggesting that activation of Plk3 is regulated by different mechanisms from those of Plk1 regulation[26]. Thus, we aimed to determine how Plk3 is activated and whether Plk3 activation mod-ulates PDAC development and metastasis.

In the present study, we carry out our investigation to (a) examine the effects of Plk3 deletion in regulation of apoptosis in $Kras^{G12D}$-induced PDAC development of genetically engineered mouse models; (b) better understand the function of Plk3 in activation of apoptosis in pancreatic cancer patient-derived cells (PDX); (c) elucidate the underlying mechanisms by which Plk3 is activated; and (d) most importantly, define the potential function of nardilysin (NRDC)-regu-lated post-translational modification.

## Results

### Plk3 expression is reduced in human PDAC, and deleting $Plk3$ promoted $Kras^{LSL-G12D}$-driven PDAC development and metastasis in mice

To investigate the importance of Plk3 expression in regulation of apoptosis and tumor development, we compared the Plk3 mRNA level in 16 tumor types and corresponding normal tissues in The Cancer Genome Atlas (TCGA) and found that the Plk3 mRNA level is reduced in most of these tumor types[27] (Supplementary Fig. 1a). To analyze Plk3 expression in human PDACs, we performed immunohistochemistry staining for Plk3 using a tissue microarray containing 33 PDAC samples and 56 normal pancreatic tissue samples. Whereas 45% (25/56) of the normal tissue samples exhibited high levels of Plk3 expression, 82% (27/33) of the PDACs had low expression of Plk3 ($p = 0.0114$) (Fig. 1a). We validated low Plk3 expression in PDAC by evaluating the level of Plk3 mRNA in fresh surgically resected primary human PDAC and adjacent normal pancreatic tissues (Supplementary Fig. 1b). Further-more, we demonstrated that Plk3 protein levels were remarkably decreased in 16 newly established PDAC PDX cell lines[28] compared to nontumorigenic human pancreatic duct epithelial (HPDE) cells (Fig. 1b and Supplementary Fig. 1c). Consistently, Plk3 expression was also significantly lower in commonly used PDAC cell lines and in tumori-genic HPDE/Kras$^{G12V}$/Her2/shp16shp14/shSmad4 (HPDE/T$^+$) cells[29] than in HPDE cells (Fig. 1c). Immunoblotting analysis showed that Plk3 expression was lost or significantly reduced in PDAC cells isolated from $p48$-$cre;Kras^{LSL-G12D};INK4a/Arf^{F/F}$ mice (KIC) and $Pdx1$-$Cre;Kras^{LSL-G12D};p53^{LSL-R172H}$ mice (KPC) compared to that in normal mouse

pancreatic epithelial cells (Fig. 1d). Furthermore, the reduced Plk3 expression in PDAC from the Segara and Logsdon cancer microarray datasets in the Oncomine database also supports our findings (Fig. 1e). Based on these data, we hypothesized that Plk3 is a potential tumor suppressor.

We next generated $p48$-$cre;Kras^{LSL-G12D};Plk3^{+/+}$ and $p48$-$cre;Kras^{LSL-G12D};Plk3^{-/-}$ mice (Table 1), and deletion of Plk3 was confirmed by IHC analysis (Fig. 1g). In agreement with Hingorani et al.[11,12], $p48$-$cre;Kras^{LSL-G12D};Plk3^{+/+}$ mice developed the full spectrum of pancreatic intraepithelial neoplasia (PanIN) lesions, but only 1 of the 28 mice spontaneously developed PDAC at 8–12 months of age (Table 1). In contrast, the $p48$-$cre;Kras^{LSL-G12D};Plk3^{-/-}$ mice developed PanIN as early as 3 months of age. Histological analysis revealed strongly increased PanIN area and grade (Fig. 1f and Supplementary Fig. 1d). Deletion of $Plk3$ provoked the advancement of lesions, leading to PDAC develop-ment in 11 of 29 (38%) mice, with 7 of the 11 (64%) having metastatic lesions in the liver or the lung by 8–12 months of age (Table 1 and Supplementary Fig. 1e). Tumor development in the $p48$-$cre;Kras^{LSL-G12D};Plk3^{-/-}$ mice was accompanied by an increase in Ki-67 staining and a decrease in apoptosis, as indicated by levels of caspase-3 activation (Fig. 1g). These results suggest that reduction in apoptosis and increase in tumor growth promoted by the loss of Plk3 are likely involved in PDAC tumor progression and metastasis.

### Plk3-induced anoikis is associated with proteolytic processing of p72Plk3 to generate p41Plk3

We expressed Plk3 in HEK293T cells in a pilot study to determine the effect of Plk3 expression on apoptosis. Remarkably, a significant number of cells rounded up and floated (Supplementary Fig. 2a). When these cells were plated onto PolyHEMA-coated plates, they remained in suspension and showed a notable increase in apoptosis compared to cells in attached culture (Supplementary Fig. 2a, b). We next grow GFP-Plk3 transfected HPDE cells under both attached and suspension condition. Immunofluorescence analysis demonstrated that cells under attachment culture displayed minimal nuclear fragmentation with or without Plk3 overexpression. Conversely, Plk3 expression in suspension culture revealed a condensed and fragmented nuclear chromatin (Supplementary Fig. 2c, d). Consistently, Plk3 transfected HPDE cells in suspension exhibited higher levels of cleaved PARP compared to attached cells (Supplementary Fig. 2e). Together, our findings suggest that Plk3-induced cell death may occur through cell detachment–induced apoptosis, known as anoikis. We assayed hTERT-immortalized HPNE cells with Dox-inducible Plk3 expression, finding that Plk3 induction triggered apoptosis (Fig. 2a). A cell detachment assay using HPDE and PDAC cells revealed significantly lower levels of PARP cleavage in all three PDAC cell lines compared to HPDE cells (Fig. 2b, c). This indicates that HPDE cells might be susceptible to anoikis, while PDAC cell lines were resistant to anoikis. We hypothe-sized that reduced Plk3 expression in PDAC could lead to this resis-tance. To further investigate the role of Plk3 in inducing anoikis, we overexpressed, knocked down, or knocked out Plk3 in various cell lines under suspension conditions. Knockdown of Plk3 expression in HPDE cells or knockout of Plk3 in PANC-1 and PATC50 cells substantially reduced PARP cleavage and cell death, indicating suppression of anoikis (Fig. 2d and Supplementary Fig. 2f). Rescuing Plk3-knockdown HPDE cells and overexpressing Plk3 in PANC-1 cells markedly induced PARP cleavage, facilitating anoikis (Fig. 2e and Supplementary Fig. 2g). MEFs isolated from Plk3$^{-/-}$ mice exhibited increased survival compared to Plk3$^{+/+}$ and Plk3$^{+/-}$ MEFs (Supplementary Fig. 2f). Additionally, the level of cleaved caspase-3 was notably reduced in Plk3 knockout PANC-1 cells, suggesting that Plk3-mediated anoikis is triggered via caspase-dependent apoptosis (Supplementary Fig. 2h).

To explore the regulatory mechanisms governing the pro-apoptotic function of Plk3, we expressed N-terminal Flag-tagged Plk3 in PANC-1 cells (Fig. 2f). Alongside the expected 72-kDa Plk3, we

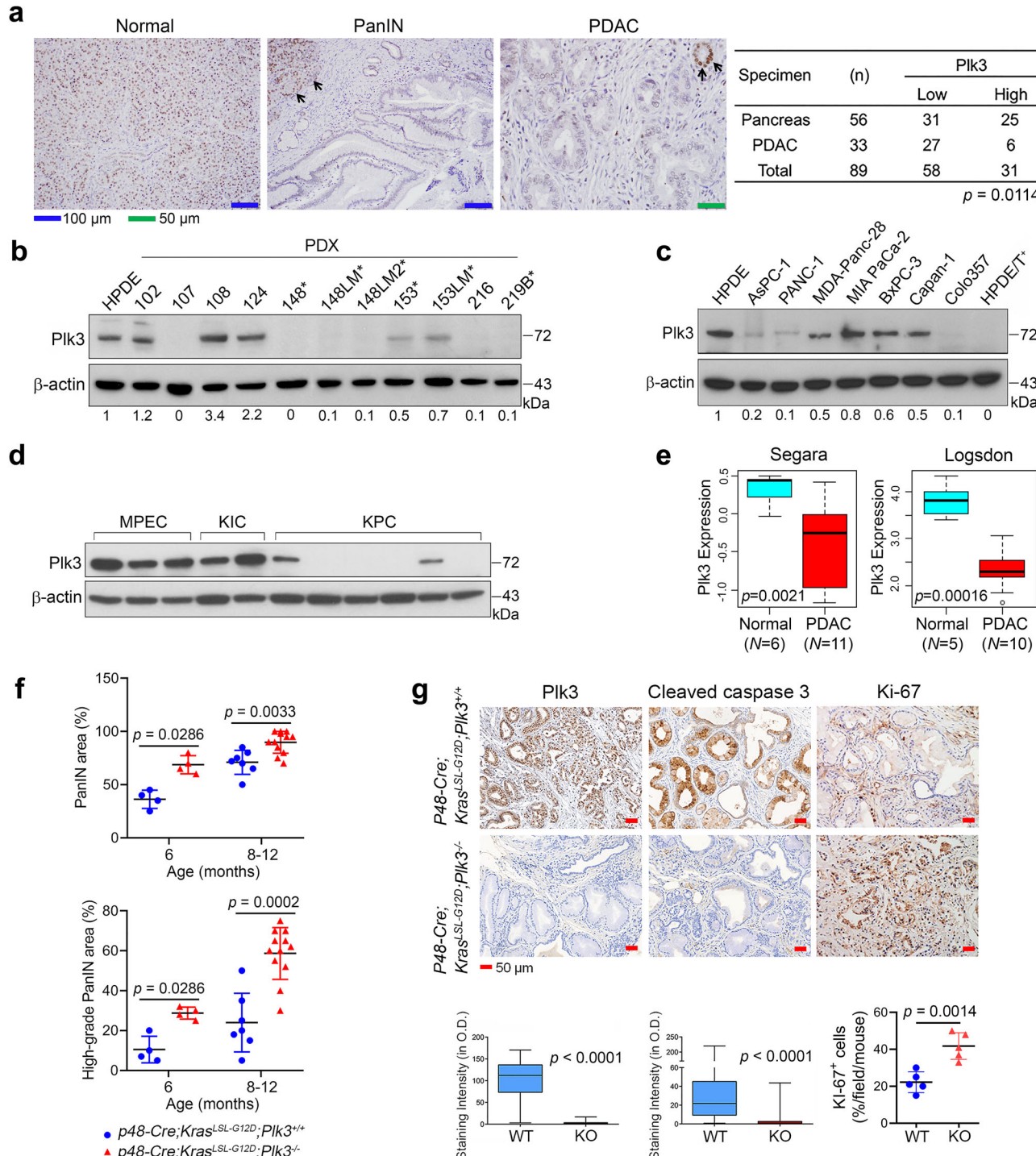

**Fig. 1 | Plk3 expression is reduced in human PDAC, and deleting *Plk3* promoted Kras^LSL-G12D^-driven PDAC development and metastasis in mice. a** Representative micrographs of Plk3-stained tissue sections showing strong nuclear Plk3 expression in normal pancreas and loss of Plk3 expression in PDAC and PanIN lesions. Strong nuclear expression of Plk3 in normal pancreas serves as an internal positive control in PanIN and PDAC lesions (middle and right panels, arrows). $p = 0.0114$, Fisher's exact test. **b, c** Immunoblots of p72Plk3 in a panel of PDX cell lines (**b**), in human PDAC cell lines and tumorigenic HPDE/T⁺ cells (**c**). * indicates cells derived from liver metastases of primary PDAC. The Plk3:β-actin ratios are shown at the bottom. **d** Immunoblot of p72Plk3 in three mouse pancreatic epithelial cell lines (MPEC) and PDAC cell lines derived from multiple KIC and KPC mouse models. **e** Box plots of Plk3 expression in the Segara and Logsdon cancer microarray datasets from Oncomine (https://www.oncomine.com), student *t*-test. **f** Quantification of the PanIN areas and PanIN grading in pancreatic tissues, obtained from 6- and 8- to

12-month-old *p48-cre;Kras^LSL-G12D^;Plk3^+/+^* (Plk3-WT) and *p48-cre;Kras^LSL-G12D^;Plk3^−/−^* (Plk3-KO) mice. Higher-grade PanIN lesions are defined as PanIN-1B, PanIN-2, and PanIN-3 lesions; $n = 4$ Plk3-WT and 4 Plk3-KO independent mice of 6 months old; $n = 7$ Plk3-WT and 12 KO independent mice of 8–12 months old. Error bars, mean ± SD; unpaired Student *t*-test (two tailed) with Welch's correction for PanIN areas and Mann−Whitney test (two tailed) for grading. **g** IHC staining for Plk3, cleaved caspase-3, and Ki-67 in Plk3-WT and -KO mice ($n = 5$). The quantification shows the number or proportion of cells positive for each marker. O.D., optical density. Data in (**b**–**d**) are representative of two independent experiments with similar results. Error bars, mean ± SD (**f**, ki-67 in **g**), two-tailed unpaired *t*-test (**f**, ki-67 in **g**). Box plots indicate minima (lower end of whisker), maxima (upper end of whisker), median (center), 25th percentile (bottom of box), and 75th percentile (top of box) (**e**, **g**). Source data are provided as a Source Data file.

**Table 1 | Suppression of oncogenic *Kras^G12D*-induced histological progression of PanIN and PDAC with Plk3**

| Genotype | Total Mouse Number | Pancreatic Lesions | | | | | | | |
|---|---|---|---|---|---|---|---|---|---|
| | | Pancreatitis | No-pancreatitis | PanIN | No-PanIN | PDAC | No-PDAC | Metastasis | No-metastasis |
| Control (*p48-Cre*, or *Kras^LSL-G12D*, or *Plk3^-/-*) | 15 | 0 | 15 | 0 | 15 | 0 | 15 | | |
| *p48-Cre;Kras^LSL-G12D;Plk3^+/+* | 28 | 28 | 0 | 28 | 0 | 1 | 27 | 0/1 | 1/1 |
| *p48-Cre;Kras^LSL-G12D;Plk3^-/-* | 31 | 29 | 2 | 29 | 2 | 11 | 20 | 7/11 | 4/11 |
| | | *p* = 0.493 | | *p* = 0.493 | | *p* = 0.0029 | | *p* = 0.0177 | |

Chi-square analysis of the association between *p48-Cre;Kras^LSL-G12D;Plk3^+/+* and *p48-Cre;Kras^LSL-G12D;Plk3^-/-*, and the observed phenotypes. Pancreatitis was found in PanIN, and PDAC; PanIN was coexisted with PDAC.

identified a 41-kDa polypeptide (p41Plk3) using an anti-Flag antibody. Antibodies specific to the N-terminus of Plk3 also detected p41Plk3, while those specific to the C-terminus did not (Supplementary Fig. 2i). Moreover, multiple PDAC cells cultured in suspension for up to 48 h also exhibited a 41 kDa band, as recognized by an N-terminus–specific anti-Plk3 antibody (Fig. 2g). Three PDX PDAC cell lines treated with siRNA against Plk3 all exhibited markedly reduced p41Plk3 (Supplementary Fig. 2j). These results indicate that p41Plk3 originates from p72Plk3. Additionally, p41Plk3 was present in tissues from Plk3^+/+ mice but absent in those from Plk3^-/- mice (Supplementary Fig. 2k). A time-course suspension culture of Plk3-overexpressing HPDE, PATC50, PANC-1, and 293T cells revealed an increase in p41Plk3 but a decrease in p72Plk3 over the time (Fig. 2h–j). PDAC cells exhibited a lower basal level of p41Plk3 compared to HPDE and 293T cells. We reasoned that the high resistance to anoikis in PDAC cells (Fig. 2c) might be attributed to low levels of precursor p72Plk3 and p41Plk3 generation. Interestingly, Plk3-overexpressing HPDE cells in suspension, but not in the attached state, exhibited p41Plk3 (Supplementary Fig. 2l). Given the higher apoptosis rate in suspension versus attachment (Supplementary Fig. 2a–e, h), we hypothesized that the occurrence of p41Plk3 promotes anoikis. As a support, knockout of Plk3 expression in PDAC cells led to a decrease in p41Plk3 expression and a reduction in apoptosis-inducing activity (Fig. 2k, l). Notably, PATC43 and 66 cells exhibited a high apoptosis rate even in attached condition, resulting in a similar attenuation of apoptosis in both attachment and detachment upon Plk3 knockout. This is possibly due to cell-specific heterogeneity and potential phenotypic changes in culture for these primary PDX lines[28]. Together, these results suggest that Plk3-induced anoikis is associated with proteolytic processing of p72Plk3 to generate p41Plk3, regulating apoptosis.

When we probed Plk3 using antibodies detecting its C-terminus, a 32-kDa Plk3 protein (p32Plk3) was detected (Fig. 2m, n). Interestingly, overexpression of the N-terminal Myc-tagged Plk1 similarly generated a 38-kDa polypeptide (p38Plk1) containing the Plk1 kinase domain and a 30-kDa polypeptide (p30Plk1) containing the C-terminus PBD of Plk1 (Fig. 2o and Supplementary Fig. 2m). The Plk1 knockdown experiments validated that the p38Plk1 protein was derived from Plk1 (Fig. 2p). Taken together, these findings establish the existence of p41Plk3 and p38Plk1, which appear to arise as a consequence of post-translational modification of Plk3 and Plk1, respectively.

**Identification of the proteolytic cleavage site in the DR1 domain of Plk3**

A clue for determining the proteolytic cleavage site of p72Plk3 came from a functional screening of an open reading frame library to identify molecules that inhibit Plk3-induced apoptosis (Supplementary Table 1). A common feature of these open reading frames isolated from 6 out of 60 surviving colonies is the presence of multiple nardilysin (NRDC) cleavage sites. Nardilysin belongs to the family of zinc-metalloendopeptidase that cleaves peptide substrates at the N-terminus of arginine residues in dibasic moieties comprising the -X-

Arg-Lys- motif. Our results demonstrate a decoy mechanism for interfering or competing with nardilysin-mediated cleavage of p72Plk3 to p41Plk3. Functionally mapping the domain that mediates the proteolytic cleavage of p72Plk3 identified amino acids 307–377, termed Domain 1–Related to Cell Death (DR1), as a region that mediates the apoptosis-inducing activity of Plk3 (Supplementary Fig. 3a–d). The DR1 sequence is well conserved amongst human Plk family members and in Plk3 across species (Supplementary Fig. 3e, f). Furthermore, the nardilysin cleavage site Arg354 of -RK- within the Plk3 DR1 region was identified using the Eukaryotic Linear Motif resource for Functional Sites in Protein (http://elm.eu.org). A closer look at the primary sequences of Plk3 and Plk1 shows that they contain four and three -RK-sites, respectively (Supplementary Fig. 3g). Among these, Only Arg354 of -RK- site is located within the DR1 domain, whereas other sites (Arg144, Arg216, Arg501) exist in the kinase domain or PBD (Supplementary Fig. 3g) and are mostly buried within these domains. A prediction of protein disorder demonstrated that the location of the DR1 domain is within a highly disordered region, exposing NRDC cleavage site to the solvent and accessible for protein interaction (Fig. 3a). We therefore hypothesize that the -RK- site in the DR1 region of Plk3 or Plk1 is likely to be the NRDC cleavage site.

MALDI-mass spectrometry analysis showed that purified NRDC cleaves p72Plk3 at Arg354 in the DR1 region (Supplementary Fig. 3h). Likewise, Plk1 is cleaved by NRDC at Arg337 (Supplementary Fig. 4). We further generated four Plk3 mutants (Plk3NT^1-353 [p41Plk3]; Plk3CT^354-646; Plk3^Δ326-377; and Plk3^R354G) to determine which had retained or had lost the capacity to induce apoptosis using a colony-formation assay in PANC-1 and 293 T cells respectively (Fig. 3b–d and Supplementary Fig. 5a–c). The results demonstrated that overexpression of p72Plk3 and p41Plk3 (Plk3NT^1-353) induced cell death, whereas Plk3CT^354-646, Plk3^R354G, and Plk3^Δ326-377 failed to induce cell death, exhibiting the same survival rate as that for control vector–transfected cells. p72Plk3-overexpressing PANC-1 cells only partially inhibited colony formation, suggesting low expression of p41Plk3 in PDAC cells. Consistently, flow cytometry analysis confirmed that Plk3, especially p41Plk3, mediated apoptosis-inducing activity (Fig. 3e and Supplementary Fig. 5d, e). Furthermore, a set of Plk3 mutation studies revealed that NRDC indeed cleaved Plk3 at residue Arg354 (Fig. 3f and Supplementary Fig. 5f–h). To fully evaluate the role of NRDC cleavage of Plk3 in promoting apoptosis, we transfected p72Plk3-WT and p72Plk3^R354G containing a mutated NRDC cleavage site in four PDAC cell lines under suspension culture for 36 h (Fig. 3g, h). The generation of p41Plk3 by NRDC cleavage was clearly inhibited to a greater degree in cells transfected with Plk3^R354G than in cells transfected with Plk3-WT (Fig. 3g). Consistently, the NRDC cleavage site mutation in Plk3 triggered less apoptosis (Fig. 3h). Knockdown or knockout of NRDC expression, or mutation of the NRDC cleavage site on Plk3 and Plk1, inhibited the generation of p41Plk3 and p38Plk1, respectively, (Fig. 3i, j and Supplementary Fig. 5i).

To further establish the enzyme/substrate relation, we mutated the enzymatic active sites in NRDC (NRDC H233G/H237G/E236G)[30] and

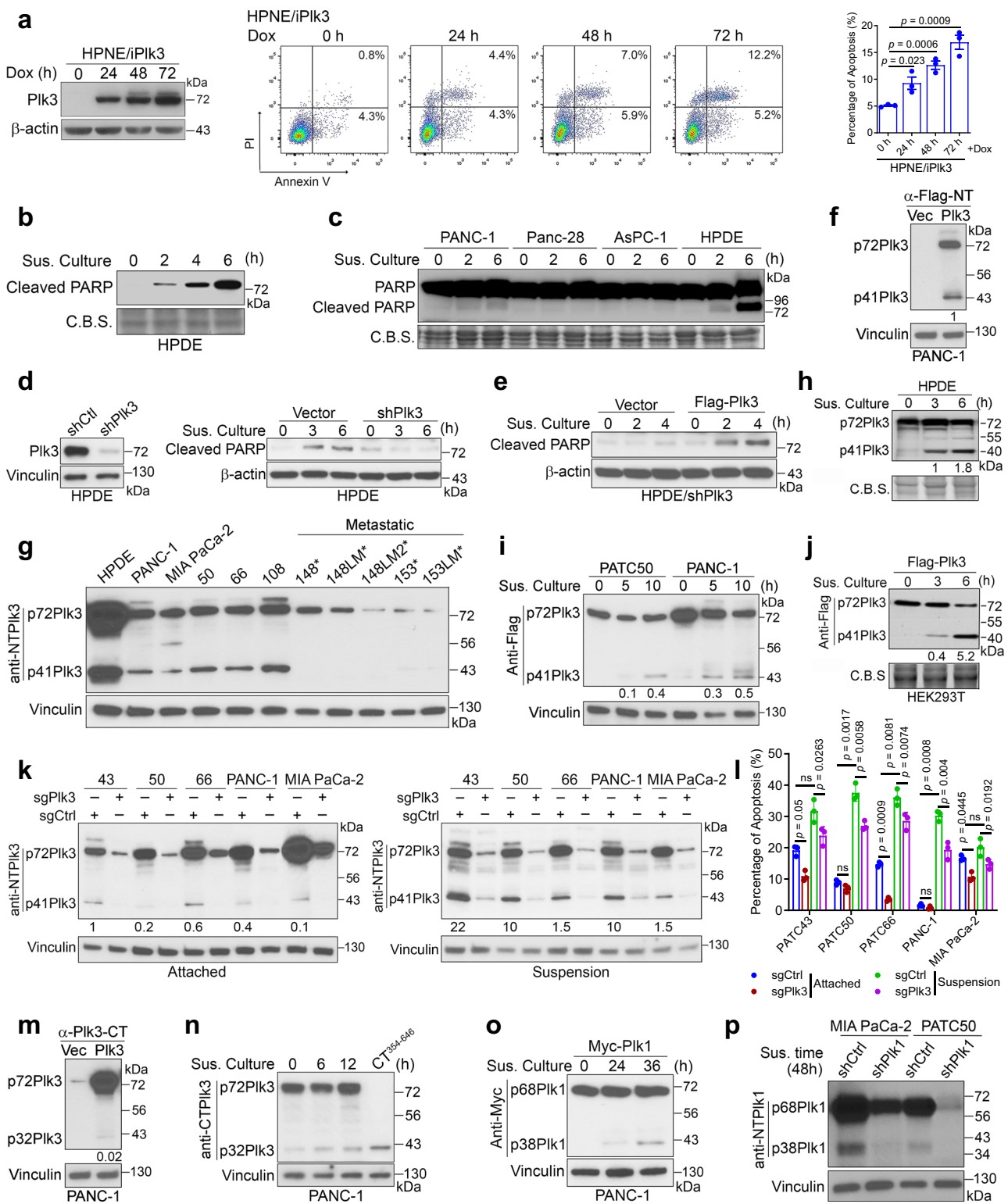

found that the cleavage of Plk3 in cells expressing this NRDC mutant was significantly reduced compared to cells expressing WT NRDC (Fig. 3k). Interestingly, Expression of a mutant form of p72Plk3 with the Arg354 -RKKK- site replaced with tobacco etch virus (TEV) protease cleavage sequence ENLYFQG (354-357TEV) failed to generate p41Plk3 (Supplementary Fig. 5j). Proteolysis of Plk3 354-357TEV by TEV protease exhibited increased p41Plk3 over the time (Supplementary Fig. 5j). Together, these results suggest that NRDC cleavage of p72Plk3 generates p41Plk3, which plays a key role in inducing apoptosis.

## Phosphoinositide 3-kinase regulates NRDC activity for Plk3 and Plk1 activation via centaurin-α1

To determine how NRDC is regulated to cleave p72Plk3, we first demonstrated that p72Plk3 forms a complex with NRDC and that amino acids 326–377 of Plk3 are important for this interaction (Fig. 4a). Next, we examined the subcellular localization of p72Plk3, p41Plk3, and NRDC (Fig. 4b–d) in PATC148 cells by expressing Dox-inducible p72Plk3 and p41Plk3, respectively. Immunofluorescence and fractionation experiments revealed that NRDC is present predominantly in the cytoplasm (Fig. 4c, d). p72Plk3 is localized in both the cytoplasm

**Fig. 2 | Proteolytic processing of p72Plk3 to generate p41Plk3 is important for induction of anoikis. a** Left, immunoblot of Plk3 expression under a Dox-inducible system in HPNE cells treated with Dox for the indicated times. Middle and right, apoptosis-inducing activity of HPNE/iPlk3 as evaluated using Annexin V/PI staining and flow cytometry analysis. **b, c** Immunoblots of cleaved PARP in HPDE cells (**b**) and PDAC cells (**c**) grown in suspension (Sus. Culture) on polyHEMA pre-coated plates at the indicated times. C.B.S., Coomassie blue–stained protein bands as a loading control. **d, e** Immunoblots of cleaved PARP in HPDE cells with stable shRNA-mediated knockdown of Plk3 (**d**) and stably reconstituted with Flag-Plk3 (**e**). **f** Immunoblot of p72Plk3 and p41Plk3 in PANC-1 cells transfected with N-terminal Flag-p72Plk3. **g** Immunoblot of p72Plk3 and p41Plk3 in a panel of PDX cell lines grown in suspension culture for 48 h. **h–j** Immunoblots of p72Plk3 and p41Plk3 in HPDE cells (**h**), PATC50 cells, PANC-1 cells (**i**), and 293T cells (**j**) grown in suspension for the indicated times. **k, l** Immunoblot of p72Plk3, p41Plk3 expression (**k**) and flow cytometry analysis of apoptosis-inducing activity (**l**) of indicated PDAC cells that were lentivirally transduced to express the sgRNA targeting Plk3 or non-targeting control sgRNA. Cells grow on tissue culture plates (Attached) or polyHEMA-coated plates in suspension (Suspension). The p41Plk3:p72Plk3 ratios are shown at the bottom. **m, n** Immunoblots of p72Plk3 and p32Plk3 in PANC-1 cells grown in suspension condition. Expression of Plk3 aa 354-646 in lane 4 in (**n**) was used as a positive control. **o** Immunoblot of p38Plk1 expression in PANC-1 cells transfected with Myc-p68Plk1 in suspension culture for the indicated times. **p** Immunoblot of Plk1 cleavage in the indicated PDAC cells transfected with Plk1 shRNA. The p41Plk3:p72Plk3 ratios or p32Plk3:p72Plk3 ratios are shown at the bottom in (**f, h–j,** and **m**). Data are representative of two independent experiments with similar results (**a** left, **b–k, m–p**). n = 3 independent experiments (**a, l**). Error bars, mean ± SEM (**a** right, **l**), two-tailed unpaired t-test (**a** right), ANOVA analysis (**l**). Source data are provided as a Source Data file.

and nucleus, and p41Plk3 is predominantly localized in the nucleus, suggesting p72Plk3 may interact with NRDC in the cytosol for proteolytic cleavage, followed by translocation of p41Plk3 to the nucleus for kinase function.

A previous study demonstrated that $p42^{IP4}$, also known as centaurin-α1 and phosphatidyl inositol-(3,4,5)-trisphosphate binding protein (PIP$_3$BP), binds to NRDC and that this interaction is controlled by the cognate cellular ligand of centaurin-α[31]. To determine whether centaurin-α1 and its cognate cellular ligands regulate NRDC-mediated Plk3 cleavage, we used the phosphoinositide 3-kinase (PI3K) inhibitor LY294002, with the results demonstrating that LY294002 reduced p41Plk3 production (Fig. 4e, f and Supplementary Fig. 5k), whereas expression of constitutive PI3K catalytic unit p110α enhanced the generation of p41Plk3 (Fig. 4g). Notably, treatment of PDAC cells with LY294002 was effective in inducing cell apoptosis, but it reduced p72Plk3 cleavage into the apoptosis inducer p41Plk3 (Fig. 4e, f). Considering oncogenic Kras activates PI3K and other signaling pathways driving pancreatic tumorigenesis, our results suggest Plk3 cleavage may not be required for the effects of PI3K in PDAC development. Furthermore, suspension culture of HPDE and a panel of PDAC cell lines revealed that the cleavage of p72Plk3 positively correlated with expression levels of NRDC and p110α in general (Fig. 4h). Consistently, we found p41Plk3 expression was significantly increased in PTEN-KO compared to PTEN-WT MEF cells, suggesting that NRDC and PI3K are required to cleave p72Plk3 (Fig. 4i). Co-IP of NRDC indicated that LY294002 increased the binding of NRDC to centaurin-α1 in the cytoplasm and decreased its binding to p72Plk3 (Fig. 4j). These findings suggest that LY294002 reduces the levels of PIP$_3$ by inhibiting PI3K in the cytoplasm (Fig. 4d) and thereby promoting the binding of centaurin-α to NRDC while inhibiting its binding to p72Plk3. Similarly, our results show that NRDC binds to Plk1 (Fig. 4k) as well, LY294002 indirectly impedes cleavage of Plk1 by NRDC by inhibiting PI3K (Fig. 4l), and Plk1 cleavage is markedly increased in PTEN-KO but not WT MEFs (Fig. 4m). In aggregate, our findings demonstrate that PI3K regulates post-translational processing of Plk1 and Plk3 involving the PI3K/centaurin/NRDC signaling axis to generate functional p38Plk1 and p41Plk3, respectively.

## C-terminal p32Plk3 inhibits N-terminal p41Plk3 kinase activity and apoptosis-inducing activity

Previous studies suggested that the N-terminal kinase domain of Plk1 is inhibited by its C-terminal Polo-box domain, but mechanisms for autoinhibition remained unclear[32,33]. Our structure analysis and modeling of human Plk3 using Plk1 complex ortholog from zebrafish[25,34] indicated that the flexible linker region (aa 316–459) containing the NRDC cleavage site may act as molecular rigging to regulate kinase activity (Fig. 5a). The IP experiment showed a direct interaction between Plk3NT$^{1-353}$ and CT$^{354-646}$ (Fig. 5b). An in vitro kinase assay using a newly identified Plk3 substrate (substrate identification is detailed in Fig. 6) revealed that Plk3 C-terminal region inhibits N-terminus kinase activity (Fig. 5c). We reasoned that the flexible linker region may strengthen the interaction between PBD and kinase domain, which in turn reduces the hinge movement between the two lobes of the kinase domain that inhibits the ATP hydrolysis (Fig. 5a). NRDC cleavage would weaken the PBD and kinase domain interaction and thereby activate the hinge movement for ATP catalysis in the kinase domain. Supporting this idea, the zebrafish Plk1 complex structure shows that the loop of the PBD reduces the lobe movements to inhibit the kinase activity when PBD binds to the kinase domain[25]. To further validate our model, we mutated L461R and L601R, which are located in the interface of PBD and the kinase domain (Fig. 5d), finding that the mutations significantly reduced the interaction between NT$^{1-353}$ and CT$^{354-646}$ of Plk3 and substantially increased the kinase activity (Fig. 5e, f). We also tested the P454R mutant, which was anchored near the ATP binding site (Fig. 5d). IP results showed P454R moderately reduced the interaction of NT$^{1-353}$ and CT$^{354-646}$ and slightly increased kinase activity (Fig. 5e, f). As a control, residue K473 at PBD was selected, as it is located away from the proposed binding interface between PBD and the kinase domain (Fig. 5a, right). K473A mutation did not affect the interaction of NT$^{1-353}$ and CT$^{354-646}$ (Fig. 5e). Our results support the Plk3 structural model and identify separation-of-function mutations that disrupt interaction of NT$^{1-353}$ and CT$^{354-646}$. Importantly, over-expression of Plk3CT$^{354-646}$ disrupted p41Plk3-induced apoptosis (Fig. 5g, h). These data highlight a possible Plk3 activation mechanism in which NRDC-mediated cleavage relieves an intramolecular inhibition of the N-terminus by the C-terminal region of Plk3. The time course of cycloheximide (CHX) treatment in PATC148 cells expressing NT$^{1-353}$- and CT$^{354-646}$-Plk3 revealed that NT$^{1-353}$ was stable, whereas CT$^{354-646}$ was subjected to degradation with a half-life of 14–15 h (Fig. 5i, j). We further demonstrated that chloroquine treatment had no effect on CT$^{354-646}$, whereas MG132, a proteasome inhibitor, blocked the degradation of CT$^{354-646}$ (Fig. 5k, l), suggesting that the stability of CT$^{354-646}$ might be regulated by polyubiquitin. Based on these findings, we surmised that scission of Plk3 to remove its C-terminus, which harbors an inhibitory effect on the N-terminus, is required to activate Plk3.

## Activated p41Plk3 phosphorylates c-Fos at Thr164 and regulates a Plk3/c-Fos feed-forward pathway to promote apoptosis

A previous report suggests that Plk3-mediated phosphorylation of caspase-8 can modulate its apoptotic activity[35]. Additionally, c-Fos, a member of the Fos family of transcription factors, was previously described to promote cell apoptosis induced under anti-proliferative and other conditions[36,37]. We sought to ask whether the pro-apoptotic functions of Plk3 and c-Fos participate in similar pathways. Firstly, co-IP experiment and Duolink assay showed a direct interaction between p41Plk3 (Plk3NT$^{1-353}$) and c-Fos (Fig. 6a and Supplementary Fig. 6a, b).

We then profiled the expression of c-Fos–regulated pro-apoptotic and anti-apoptotic gene expression in HPNE/ip72Plk3 and HPNE/

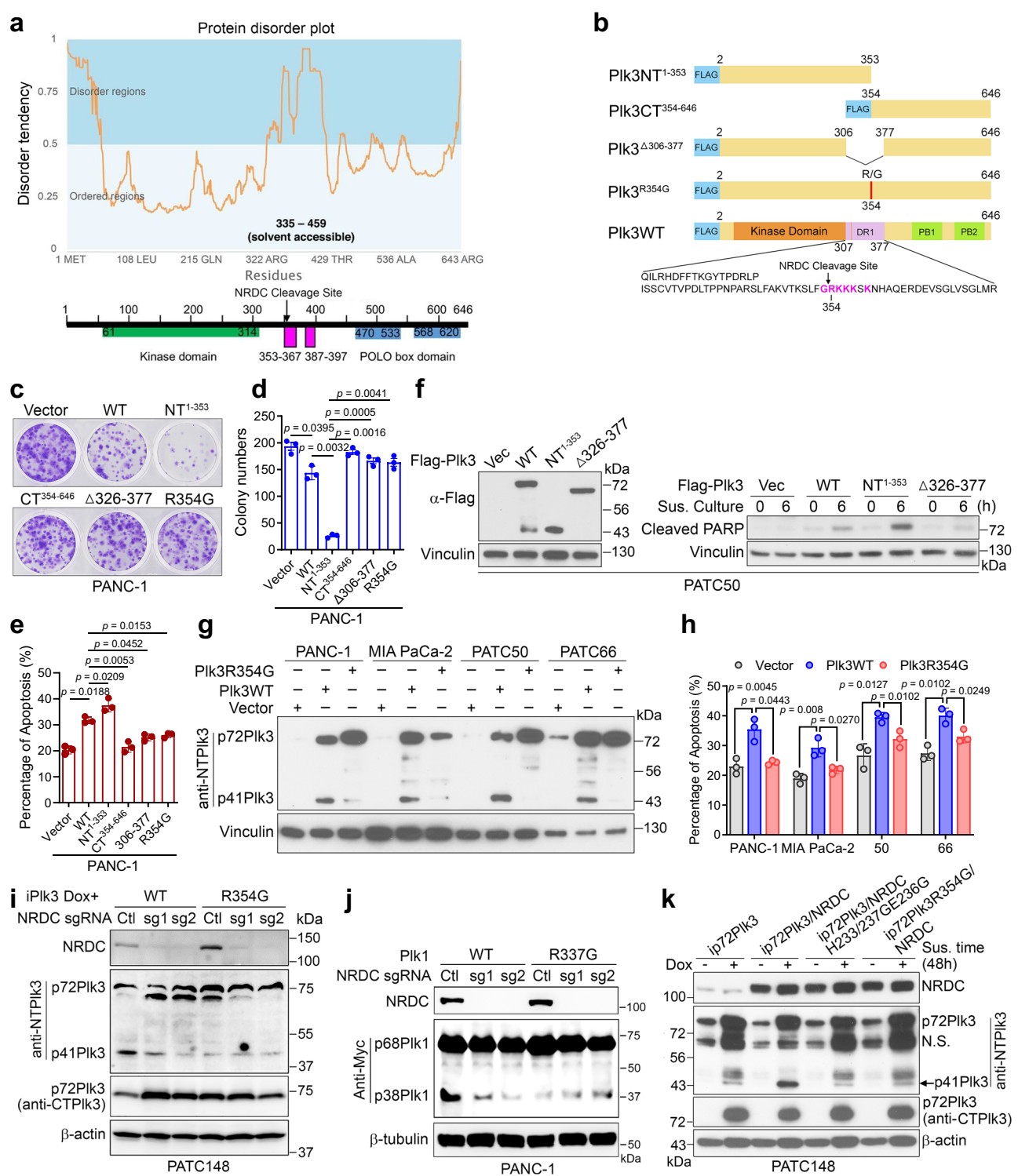

ip41Plk3 cells. Induction of p41Plk3 by Dox increased the expression of most pro-apoptotic genes compared with induction of p72Plk3 (Fig. 6b, c), and p41Plk3 also suppressed the expression of three of six anti-apoptotic genes (Fig. 6d, e). These data support our observations that activated p41Plk3 has significantly enhanced pro-apoptotic function. Expression of pro-apoptotic genes was substantially induced in 293T cells transfected with p72Plk3 and p41Plk3, but not Plk3$^{\Delta306-377}$ (Supplementary Fig. 6c). Furthermore, co-transfection of cells with p72Plk3 and c-Fos increased apoptosis to a far greater extent than when either p72Plk3 or c-Fos was transfected individually into 293T, PANC-1, and PATC50 cells (Supplementary Fig. 6d). Conversely,

knockdown of c-Fos expression in HPNE/ip72Plk3, HPNE/ip41Plk3, PATC148/ip41Plk3, or PATC153/ip41Plk3 cells diminished p72Plk3- or p41Plk3-induced apoptosis (Supplementary Fig. 6e–i). Together, these results suggest that Plk3, specifically p41Plk3, activates c-Fos to induce apoptosis in part by increasing expression of pro-apoptotic genes.

Next, we investigated whether c-Fos is a substrate of Plk3. An in vitro kinase assay demonstrated that both p72Plk3 and p41Plk3 phosphorylated c-Fos (Fig. 6f). Most strikingly, p41Plk3 showed significantly increased kinase activity compared with p72Plk3 (Fig. 6f, bottom, p72Plk3-SFB vs. Flag-p41Plk3), supporting the notion that NRDC-mediated cleavage to generate p41Plk3 is required to stimulate

**Fig. 3 | Identification of the proteolytic cleavage site in the DR1 domain of Plk3. a** Top, the Plk3 protein disorder plot was generated by GeneSilico MetaDisorder server (http://genesilico.pl/metadisorder/). The Solvent accessibility was predicted by RaptorX server. Bottom, the Plk3 domain diagram matching the disorder tendency is shown. **b** Schematic diagram of Flag-tagged Plk3 and mutants with DR1 region and the NRDC cleavage site depicted. **c–e** Colony-formation assay (**c**, **d**) and flow cytometry analysis of apoptosis-inducing activity (**e**) from PANC-1 cells transfected with indicated Flag-tagged Plk3 or mutants (Plk3NT$^{1-353}$ is p41Plk3). **f** Immunoblots of p72Plk3 and p41Plk3 (left) and cleaved PARP (right) at the indicated times in PATC50 cells transfected with Flag-tagged WT p72Plk3 or Plk3 mutants and grown under suspension conditions. **g**, **h** Immunoblot of p72Plk3 and p41Plk3 expression (**g**) and flow cytometry analysis of apoptosis-inducing activity (**h**) from PDAC cells transfected with indicated Flag-tagged Plk3-WT or mutants and grown under suspension conditions for 36 h. **i** Immunoblot of p72Plk3 and p41Plk3 expression in PATC148 cells that were lentivirally transduced to express the sgRNA targeting NRDC or non-targeting control sgRNA, and then stably transfected with p72Plk3 or p72Plk3$^{R354G}$ under a Dox-inducible system. **j** Immunoblot of p68Plk1 and p38Plk1 expression in PANC-1 cells that were stably transfected with Plk1 or Plk1$^{R337G}$ after the CRISPR/Cas9 deletion of NRDC. Cells in **i** and **j** were grown in suspension culture for 48 h. **k** PATC148 cells stably expressing Dox-inducible p72Plk3 or p72Plk3$^{R354G}$ were lentivirally transduced to express NRDC WT or H233G/H237G/ E236G mutant. Lysates were subjected to immunoblot to assess the cleavage of Plk3. Data are representative of two independent experiments with similar results (**f**, **g**, **i–k**). Error bars, mean ± SEM (**d**, **e**, **h**), $n = 3$ independent experiments, two-tailed unpaired $t$-test (**d**, **e**, **h**). Source data are provided as a Source Data file.

the kinase activity. Similarly, p38Plk1 exhibited ~4-fold increased kinase activity compared with p68Plk1 (Supplementary Fig. 6j). Furthermore, we performed an in vitro kinase assay using purified Plk3 protein from a commercial source (Fig. 6g), which is a mixture of p72Plk3 and p41Plk3 (Lane 1). Mass spectrometry identified Thr164, Ser122, Ser261, Ser362, and Thr376 as five putative residues on c-Fos that are phosphorylated by Plk3 (Supplementary Fig. 7a).

To determine whether the above phosphorylation sites have biological function, we converted the serine and threonine residues to alanine to prevent phosphorylation, or to aspartate or glutamate to mimic phosphorylation. As shown in Supplementary Fig. 7b, Thr164 serves as a major regulatory phosphorylation site on c-Fos that promotes c-Fos−mediated apoptosis. We generated a mouse monoclonal antibody that specifically recognizes Thr164-phosphorylated c-Fos. IP results demonstrated that p41Plk3 stimulated the Thr164 phosphorylation of WT c-Fos, but not c-Fos T164A mutant (Supplementary Fig. 7c). Immunofluorescence and cell fractionation experiments showed that Thr164-phosphorylated c-Fos is predominately localized in the nucleus (Supplementary Fig. 7d, e). Furthermore, transfection of PANC-1 cells with c-Fos resulted in substantial elevation of p72Plk3 mRNA and protein levels compared with control cells (Fig. 6h, i), suggesting that Plk3 is a downstream target gene of c-Fos. Subsequent chromatin IP experiments demonstrated that c-Fos binds to three AP-1 sites within the 5-kilobase *PLK3* promoter region (Fig. 6j). we performed luciferase reporter gene assays in PATC148/ip41Plk3 cells, finding that c-Fos reporter activity was upregulated upon p41Plk3 expression, but decreased by mutation of c-Fos binding sites 1 and 2 on Plk3 promoter, as well as by mutation of Thr164 on c-Fos (Fig. 6k). These data suggest that c-Fos transcriptional activation activity is mediated by Plk3 phosphorylation on c-Fos. qRT-PCR analysis of PATC148/ip41Plk3 cells revealed that expression of pro-apoptotic genes and anti-apoptotic gene was regulated by WT c-Fos. The transcription activation of these c-Fos responsive genes was impaired by c-Fos T164A mutant (Supplementary Fig. 7f). We further explored the functional role of Plk3-mediated phospho-c-Fos activation on PDAC cell proliferation. The anoikis assay revealed that c-Fos Thr164 mutation greatly reduced cell apoptosis compared with c-Fos WT cells with the presence of p41Plk3 expression (Fig. 6l–n). Taken together, these results suggest that Plk3 activates c-Fos by phosphorylating it at Thr164; nuclear accumulation of the activated phospho-c-Fos concomitantly induces the transcription of *PLK3* to regulate a Plk3/c-Fos feed-forward loop that promotes apoptosis in PDAC cells.

## Activated p41Plk3-induced apoptosis and suppressed PDAC tumorigenesis and metastasis

Next, we investigated whether scission of p72Plk3 by NRDC is functionally involved in PDAC development. It has been reported that Plk3 is implicated in the regulation of various stages of the cell cycle[38,39]. Altered checkpoint response in a cell-division transition may impact cell proliferation or apoptosis, providing a potential mechanism for Plk3 to modulate cancer development. We compared the effects of inducible p72Plk3 and p41Plk3 expression on the cell-cycle progression in PATC148 cells. Flow cytometry analysis revealed that overexpression of p41Plk3-induced G2-M arrest, whereas p72Plk3 had no effect on cell-cycle distribution (Supplementary Fig. 8a). We then determined the duration of the mitotic arrest induced by p41Plk3 in PATC148 cells synchronized by the double-thymidine block (Supplementary Fig. 8b–e). Whereas most p72Plk3-on cells had returned to G1 phase within 12 h after release from the thymidine block, the proportion of p41Plk3-on cells in mitosis remained above 80% at this time, and they progressed to G1 by 31 h. Consistently, the amounts of cyclin B in p72Plk3-on cells were greatly reduced 6 h after release, whereas the accumulation of cyclin B apparent in p41Plk3-on cells was maintained for more than 28 h after release (Supplementary Fig. 8b–e). Conversely, CRISPR/cas9 deletion of Plk3 in Panc-28 cells was able to accelerate mitotic progression (Supplementary Fig. 8f, g). BrdU incorporation analysis revealed that p41Plk3 expression significantly suppressed S phase progression after release from nocodazole block compared with p72Plk3 (Supplementary Fig. 8h–j). Multiple lines of evidence suggest that prolonged mitotic arrest triggers the activation of apoptotic pathway[40,41]. p41Plk3-mediated mitotic arrest might play critical roles in suppressing tumor growth by inducing cell apoptosis.

Stable expression of NRDC in PATC148 cells with Dox-induced expression of p72Plk3 resulted in slower tumor growth in an orthotopic xenograft mouse model (Fig. 7a). Furthermore, the reduction of PDAC growth was lost when overexpression of either enzymatic inactive NRDC H233G/H237G/E236G mutant or WT NRDC together with the cleavage-resistant p72Plk3 R354G mutant (Fig. 7a). We have shown that NRDC expression produces slightly higher levels of p41Plk3 compared with expression of NRDC or p72Plk3 mutants (Fig. 3k); thus, reduced tumor burden in NRDC-overexpressing mice is likely due to increased expression of p41Plk3 generated from NRDC cleavage, and endopeptidase activity of NRDC is required for Plk3-mediated suppression of PDAC. Hence, we further assessed the impact of the activated form of Plk3, p41Plk3, on PDAC growth and metastasis. Dox-inducible p41Plk3 expression dramatically inhibited PATC148 cells growth, colony formation, cell migration, and invasion while increasing apoptosis. Conversely, PATC148 cells stably transduced with p72Plk3 did not exhibit pro-apoptotic effects upon Dox induction (Fig. 7b, c and Supplementary Fig. 9a–e). As expected, we observed that the "Dox on" p72Plk3 group still developed metastatic PDAC compared with the "Dox off" group among nude mice orthotopically injected with highly metastatic PATC148 cells, whereas Dox-inducible p41Plk3 expression profoundly impaired tumor growth and subsequent metastasis (Fig. 7d–f). We also established Dox-regulated p41Plk3 and p72Plk3 in another metastatic PDX PATC153 cell line and revealed the same induction of apoptosis and inhibition of PDAC and metastasis for p41Plk3, but not p72Plk3 (Supplementary Fig. 9f–l). Furthermore, we showed that p41Plk3 expression prolongs survival compared with control or WT p72Plk3 expression in a subcutaneous xenograft mouse model (Fig. 7g), thus establishing that p41Plk3, not

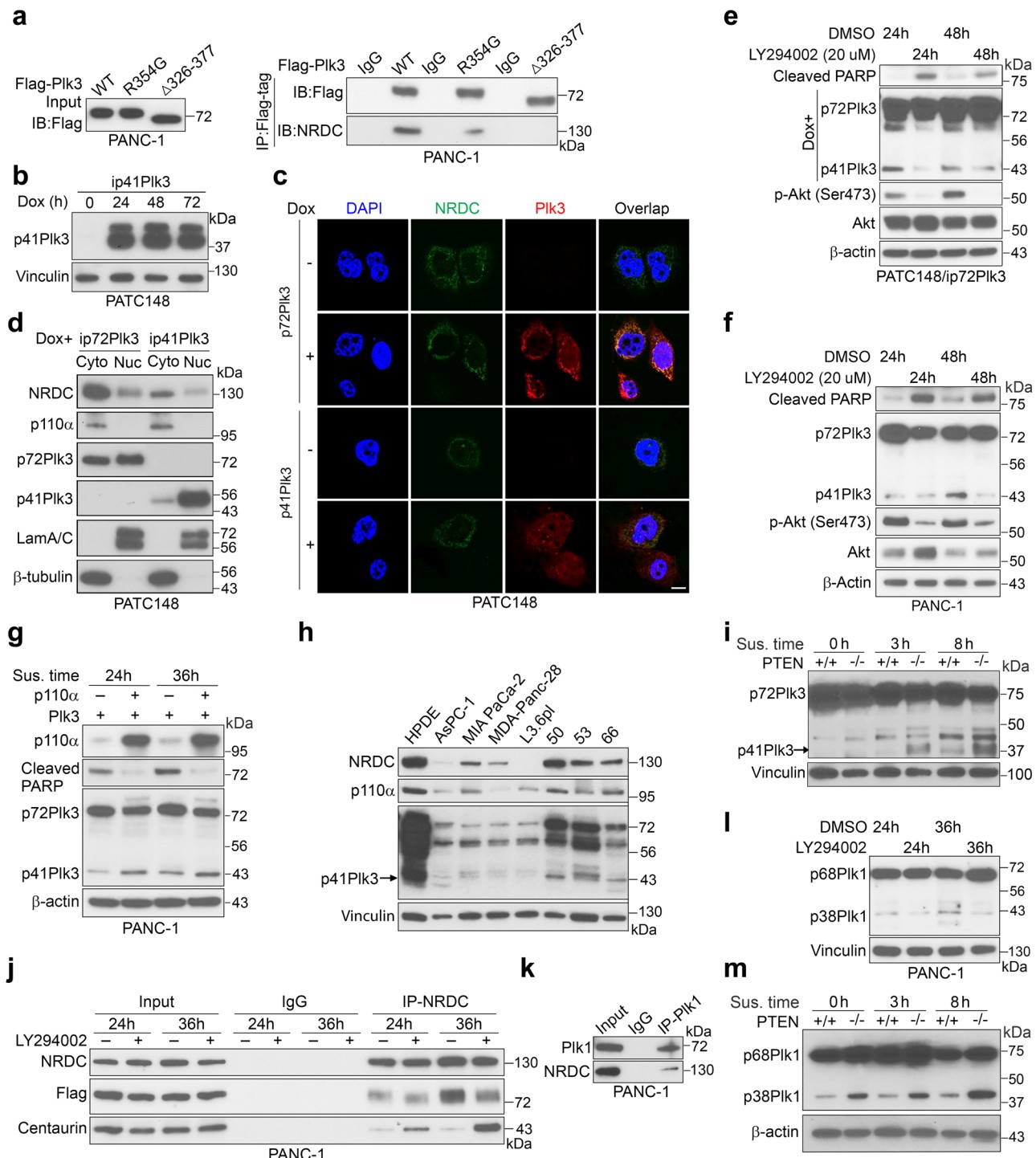

**Fig. 4 | Phosphoinositide 3-kinase regulates NRDC activity for Plk3 and Plk1 activation via centaurin-α1. a** Immunoprecipitation and IB detection of NRDC-Plk3 interactions in PANC-1 cells transfected with Flag-tagged Plk3-WT, R354G, or Δ326−377 mutants. **b** p41Plk3 expression under a Dox-inducible system in PATC148 cells treated with Dox for the indicated times. **c, d** Results of immunofluorescence (**c**) and immunoblot (**d**) performed to show localization of p72Plk3, p41Plk3, and NRDC and to detect interactions between p72Plk3 or p41Plk3 and NRDC in PATC148 cells with Dox-inducible expression of p72Plk3 or p41Plk3. Bar, 10 μM. **e, f** Immunoblots of p72Plk3, p41Plk3, and cleaved PARP expression in PATC148/ip72Plk3 cells (**e**) and p72Plk3-transfected PANC-1 cells (**f**) treated with PI3K inhibitor LY294002 (20 μM) for indicated times. **g** p72Plk3 and p41Plk3 expression in PANC-1 cells co-transfected with Plk3 and a vector control or the PI3K catalytic unit

p110α. **h** p41Plk3, NRDC, and p110α expression in a panel of PDAC cell lines and HPDE cells grown in suspension for 36 h. **i, m** Immunoblots of p72Plk3 and p41Plk3 (**i**) or p68Plk1 and p38Plk1 (**m**) in PTEN[+/+] and PTEN[−/−] MEF cells transfected with p72Plk3 or p68Plk1 in suspension culture for indicated times. **j** IP and IB detection of increased binding of NRDC with α-centaurin and decreased binding of Plk3 with NRDC in PANC-1 cells transfected with Flag-p72Plk3 and treated with or without LY294002 (32 μM). **k** IP and IB detection of Plk1-NRDC interactions in PANC-1 cells. **l** Immunoblot of p68Plk1 and p38Plk1 in Plk1 transfected PANC-1 cells treated with LY294002 (20 μM) for different times. Data in (**a, b, d−m**) are representative of two independent experiments with similar results. Source data are provided as a Source Data file.

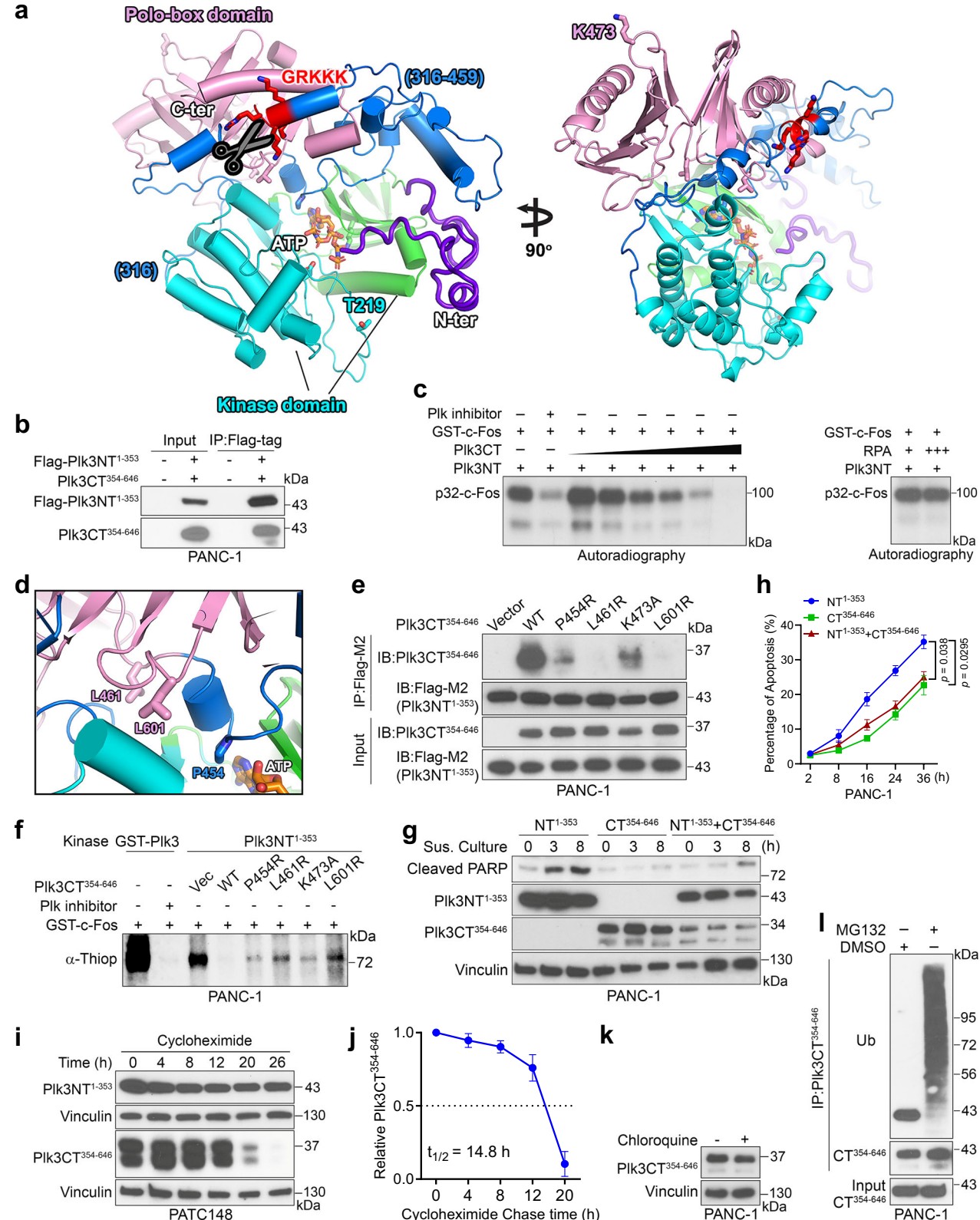

uncleaved p72Plk3, elevates its kinase activity and promotes apoptosis, G2/M arrest, and PDAC suppression.

Multiple lines of evidence suggest that Plk1 is overexpressed and plays an oncogenic role by inhibiting apoptosis in pancreatic and other cancers[19,42]. To directly test whether p38Plk1, a cleaved form of Plk1, has anti-apoptotic function in pancreatic cancer, we established PATC148/ip38Plk1 and PATC148/ip68Plk1 cells and found that

expression of p38Plk1 significantly increased cell proliferation while reducing apoptosis compared with p68Plk1 (Supplementary Fig. 10a–d), demonstrating the more pronounced function of p38Plk1 in controlling apoptosis.

To further investigate the opposing effects of Plk3 and Plk1 in regulation of apoptosis, we first grew multiple PDAC cell lines in attached and suspension culture conditions, finding that expression of

**Fig. 5 | C-terminal p32Plk3 inhibits N-terminal p41Plk3 kinase activity and apoptosis-inducing activity. a** Model of Plk3 structure predicted by Rosetta server from a close Plk1 structural homolog (PDB: 4j7b) shows interactions between kinase domain (two lobes in cyan and green ribbon), flexible N-terminus region (purple) and Polo-box domain (pink), connected by a flexible linker region (blue) containing the NRDC cleavage site (GRKKK). ATP analog, AMPPNP (orange sticks), is modeled at the hinge between the two lobes of the kinase domain by superimposition with Plk1 kinase domain (PDB: 2ou7). **b** IP and IB detection of N- and C-terminus Plk3 interaction in cells transfected with Flag-Plk3NT[1-353] and Plk3CT[354-646]. **c** In vitro kinase assay of recombinant GST-Plk3NT in the presence of increased MBP-Plk3CT-His or a non-relevant protein, RPA with or without treatment with Plk inhibitor. Substrate c-Fos identification is described in Fig. 6. **d** Selected separation-of-function residues at the interface of kinase and PBD are shown in sticks. **e, f** IP and IB detection of N- and C-terminus Plk3 interactions (**e**) and in vitro kinase assay of

N-terminus Plk3 (**f**) in PANC-1 cells transfected with Flag-Plk3NT[1-353] and Plk3CT[354-646] or mutants. **g, h** Immunoblot of cleaved PARP (**g**) and apoptosis-inducing activity (**h**) in PANC-1 cells transfected with NT[1-353], CT[354-646] of Plk3 alone, or co-transfected with NT[1-353] and CT[354-646] of Plk3. **i** N- and C-terminus Plk3 expression in NT[1-353]- or CT[354-646]-Plk3 transfected PATC148 cells treated with cycloheximide (10 μg/ml) at the indicated time points. **j** Quantification of cycloheximide assays to depict Plk3CT[354-646] expression vs. timed cycloheximide treatment. Dotted line indicates 50% of initial Plk3CT[354-646] protein signal. **k** Immunoblot of C-terminus Plk3 in PANC-1 cells transfected with Plk3CT[354-646] followed by chloroquine treatment (10 μM). **l** Immunoblot of ubiquitination of C-terminus Plk3 in Plk3CT[354-646]-transfected PANC-1 cells treated with DMSO or MG132 (10 μM). Each experiment was repeated an additional time with similar results (**b, c, e–g, i, k, l**). Error bars, mean ± SEM (**h, j**), $n = 3$ independent experiments, two-tailed unpaired $t$-test (**h, j**). Source data are provided as a Source Data file.

---

p41Plk3 was increased in the detached HPDE, PATC50, 66, and 108 cells compared with attached cells, but remained the same in Panc-28, MIA-PaCa-2, and PATC43 cells. The expression of p68Plk1 was decreased in all six PDAC cell lines cultured in suspension except HPDE (Supplementary Fig. 10e). It is worth noting that the response to apoptosis in these cells coincided with these observed alterations of p41Plk3 and Plk1 expression (Supplementary Fig. 10f). These data suggest that the upregulation of p41Plk3 and downregulation of p68Plk1 in detached PDAC cells are critical in driving cells towards apoptosis.

We then explored whether there is inter-regulation between Plk1 and Plk3 by stably expressing both Plk1 and Plk3 together. The increased p41Plk3 expression may suppress the expression of p68Plk1, resulting in decreased levels of p38Plk1 compared to p68Plk1 over-expression alone, which consequently led to enhanced apoptosis activity in co-transfected cells (Supplementary Fig. 10g, h). Altogether, we identified a common regulatory mechanism of both Plk3 and Plk1 via NRDC proteolytic cleavage in controlling PDAC cell apoptosis and proliferation.

## Discussion

Here, we have uncovered the long-sought post-translational mechanism of Plk3 activation. The activation of Plk3 is regulated by the NRDC scission mechanism (Fig. 7h). p41Plk3, but not the noncleaved p72Plk3, controls cell proliferation and cell-cycle progression. This regulation may contribute to a pro-anoikis response. Specifically, p41Plk3 phosphorylated c-Fos on Thr164, which in turn increases the transcription of Plk3 and the pro-apoptotic genes, resulting in a feed-forward regulation to promote apoptosis, G2/M arrest, PDAC, and metastasis suppression. In human PDACs, the simultaneous detection of loss of Plk3 expression and overexpression of Plk1 might disrupt the regulation of anoikis, subsequently fueling cancer progression and metastasis. We demonstrated that deleting Plk3 in *p48-cre;Kras^LSL-G12D* mice inhibited caspase-3 activation. Building on the findings of Ikuta et al. showing that pancreatic deletion of NRDC in *Kras^G12D*-driven mice promoted PDAC tumorigenesis, and considering the absence of NRDC expression in a subset of human PDAC[43], the metastasis observed in Plk3-knockout mice might be attributed to deficient NRDC-mediated cleavage on Plk3 in PDAC cells. Our data suggest that, in addition to its reduced expression level, Plk3 in PDAC cells is more resistant to NRDC-mediated cleavage into activated p41Plk3, conferring PDAC cells with apoptosis resistance, contributing to their progression towards malignancy and metastasis. Taken together, our study sheds light on a potential mechanism wherein activated Plk3 induces anoikis, thereby suppressing PDAC progression and metastasis.

PI3K-generated PIP₃ is the second messenger to activate several pathways that include AKT/mTORC1[44,45]. We identified PI3K-regulated NRDC cleavage of Plk1 and Plk3 as a post-translational modification to render them active, revealing a paradigm-shifting mechanism for regulating Plk activation. The far-reaching significance is that PI3K orchestrates both survival and apoptotic signals through the NRDC regulatory mechanism of Plk1 and Plk3 to balance survival and apoptotic pathways for the sake of maintenance of cell homeostasis.

Conformational plasticity is essential for kinase activity. The flexible linker acts as molecular rigging to interlink PBD and the kinase domain that inhibits ATP catalysis by reducing the conformational plasticity of the kinase domain (Fig. 5a). Cleavage by NRDC would relax the C-terminal linker that may in turn be destabilized. We have showed that p32Plk3 stability is regulated by the ubiquitin (Ub)-proteasome system (Fig. 5i–l). When Plk3 is cleaved and p32Plk3 is removed by degradation, the N-terminal p41Plk3 gained kinase activity and biological function to induce apoptosis, implying a possible regulation of p72Plk3 by p32Plk3 in trans and *in cis*, and a potential autoregulation mechanism for p41Plk3 activity.

Our bioinformatics analysis revealed that 10,947 (53.6%) out of 20,417 protein sequences have 1–3 NRDC cleavage sites in the UniProt/Swiss-Prot subset (Supplementary Table 2). Our biochemical data are consistent with the structural analysis of the Plk3 kinase molecule and structural explanation of the Plk3 activation mechanism. These findings highlight the potential widespread functions of the identified post-translational modification. The discovery of NRDC-regulated specific cleavage as a post-translational modification predicts this important mechanism that orchestrates many biological signaling pathways through NRDC proteolytic activity. There may be a widespread interest in this post-translational mechanism, and these findings suggest a few exciting avenues for future investigations.

## Methods

### Mouse studies and patient samples

All animal experiments were carried out in accordance with protocols (00001814-RN01) approved by the Institutional Animal Care and Use Committee at The University of Texas MD Anderson Cancer Center. The study is compliant with all relevant ethical regulations regarding animal research. All the experiments involving human pancreatic cancer tissues were reviewed by the Institutional Review Board at The University of Texas MD Anderson Cancer Center (LAB07-0830_MOD004). The tissues were de-identified, with only information on sex, race, age, and cause of death, and obtained from the institution tumor bank with the IRB committee's approval.

### Mouse models

The mice were exposed to a 12-h light/12-h dark cycle at 22–24 °C with 50–60% humidity, bred as specific pathogen-free mice, and given free access to food and water. A *Plk3* knockout mouse strain generated in Peter J. Stambrook's laboratory in the Department of Molecular Genetics, University of Cincinnati Cancer Institute[46], was crossed with the *p48-cre* strain[47] and *LSL-Kras^G12D* strain[48] to generate the experimental cohorts, the genotypes of which included *p48-cre;Kras^LSL-G12D;Plk3^−/−* and *p48-cre;Kras^LSL-G12D;Plk3^+/+*. KIC[8] and KPC[12] mice were bred in house at MD Anderson on a mixed background.

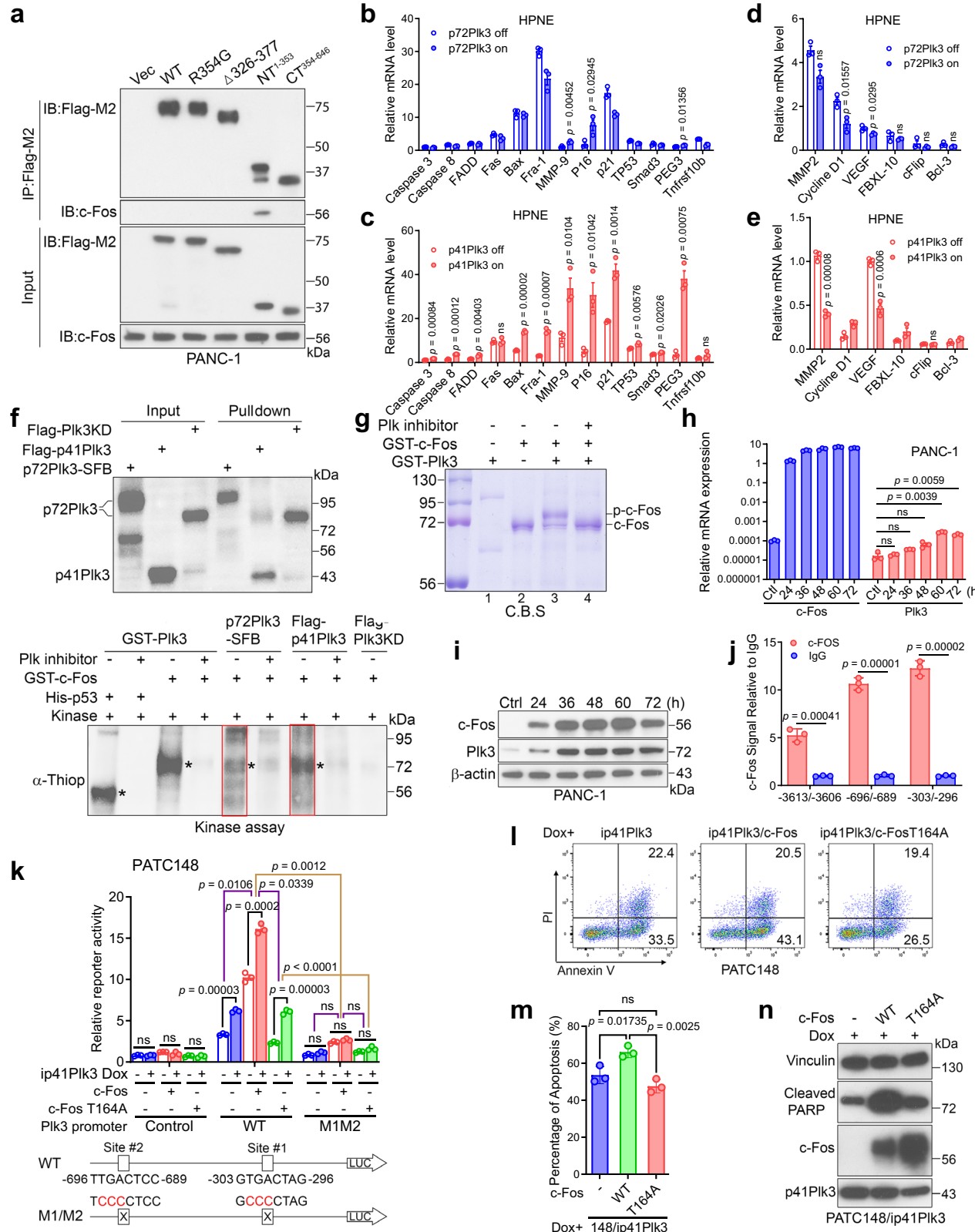

## In vivo tumorigenesis study

The experiments with orthotopic and subcutaneous mouse models were performed under protocols approved by the IACUC of MD Anderson Cancer Center. Nude mice aged 5–6 weeks were purchased from Charles River Laboratories International. When cell confluence reached 80%, cells were harvested and washed twice with PBS buffer. Cells were suspended in PBS buffer with 50% Matrigel. For orthotopic

implantation, each mouse was injected (25 μl) with $1 \times 10^5$ cells for PATC148 or $2.5 \times 10^5$ cells for PATC153 into pancreas of nude mice. For Dox treatment, mice were fed with Dox water (doxy 2 g/l, sucrose 20 g/l) upon cell inoculation and continued the treatment until termination. Mice were monitored daily except weekends for signs of disease progression and for detecting the liver metastasis. Moribund animals such as the pancreatic tumor reaching a size larger than 2 cm

**Fig. 6 | Activated p41Plk3 phosphorylates c-Fos at Thr164 and regulates a Plk3/c-Fos feed-forward pathway to promote apoptosis. a** IP and IB analysis of the interaction between Plk3WT or mutants and c-Fos in PANC-1 cells. **b–e** qRT-PCR analysis of c-Fos–regulated pro-apoptotic or cell-cycle–related (**b, c**) and anti-apoptotic (**d, e**) gene expression in HPNE cells with inducible expression of p72Plk3 or p41Plk3. **f** In vitro kinase assay using p72Plk3-SFB (boxed, S protein-FLAG-streptavidin binding peptide, tagged at Plk3 C-terminus), Flag-p41Plk3 (boxed), or Flag-p72Plk3KD (kinase-dead K91R mutant) immunoprecipitated from 293T cells to incubate with purified recombinant GST–c-Fos with or without Plk inhibitor. P53 and c-Fos phosphorylation by recombinant GST-Plk3 (a mixture of p72Plk3 and p41Plk3) was used as positive control; phosphorylation was detected by α-ThioP (asterisk). **g** In vitro kinase assay showing Plk3-mediated phosphorylation of c-Fos caused c-Fos migration to a higher molecular weight position. Shifted and unshifted c-Fos bands (Lane 3) with (Lane 4) or without (Lane 3) Plk inhibitor treatment

were analyzed by mass spectrometry to deduce phosphorylation sites. c-Fos alone (Lane 2) was analyzed by mass spectrometry to exclude the c-Fos auto-phosphorylation. Both GST-Plk3 and GST–c-Fos are purified proteins as described in (**f**). Gel was stained by Coomassie blue. **h, i** qRT-PCR (**h**) and IB (**i**) analysis of p72Plk3 expression in c-Fos–transfected PANC-1 cells. **j** Chromatin IP assay and real-time PCR comparing the ratio of anti–c-Fos antibody to IgG at the indicated Plk3 promoter region. **k** Effects of expressing c-Fos WT and T164A mutant on the activity of Plk3 reporter in inducible PATC148/ip41Plk3 cells. **l–n** Apoptosis-inducing activity (**l, m**) and immunoblot of cleaved PARP (**n**) in inducible148/ip41Plk3 cells stably transfected with c-Fos WT or T164A mutant. Data are representative of two independent experiments with similar results (**a, f, g, i, n**). Error bars, mean ± SEM (**b–e, h, j, k, m**), $n = 3$ independent experiments, two-tailed unpaired $t$-test (**b–e, j, k, m**), Source data are provided as a Source Data file.

in diameter, were sacrificed as mandated by the IACUC protocol. Endpoints includes weight loss, loss of mobility, and other signs of distress. Pancreas weights and images were recorded, and pancreatic, liver, and lung tissues were further analyzed by hematoxylin and eosin stains. Xenograft tumors were generated by subcutaneous injection of AsPC-1 cells into the flanks of nude mice at $2 \times 10^6$ cells per injection site. Data of 5 animals per experimental group are indicated in figure legends. Randomization was conducted by using an internal computer software (Indigo).

### Tissue specimens and immunohistochemistry

A human pancreatic cancer tissue microarray was constructed, and sections were obtained from paraffin blocks of primary PDAC and paired adjacent normal pancreatic tissue samples from 89 patients. Normal tissue was obtained from the region of the pancreatic neck proximal to the tumor-bearing site. Tissue specimens were collected within 1 h after surgery under a protocol approved by the MD Anderson Institutional Review Board, and written informed consent was obtained from all patients at the time of enrollment. Tissue sections were subjected to standard immunohistochemical (IHC) analysis, and images of the sections were acquired using a digital camera attached to a microscope. The IHC staining results were independently evaluated by two pathologists.

### Anoikis induction and flow cytometry

Anoikis was assayed by plating cells on polyHEMA-coated plates. Approximately $1 \times 10^6$ cells were incubated in the plate wells for 12–18 h in a humidified (37 °C, 5% $CO_2$) incubator. Cells were harvested and resuspended in 1× binding buffer containing 0.01 M HEPES (pH 7.4), 0.14 M NaCl, and 2.5 mM $CaCl_2$. Staining of cells for apoptosis was performed using APC annexin V (BD Biosciences) and propidium iodide (PI; BD Biosciences) according to the manufacturer's instructions. Cells were incubated for 15 min at 25 °C in the dark and then analyzed using flow cytometry with a BD FACSCanto II device (BD Biosciences). All data are shown as mean ± standard deviation and are representative of three independent experiments. Data analysis was performed using the FlowJo software program (version 10.7.1, Tree Star).

### Cell lines and cell culture

Human pancreatic cancer cell lines MIA-PaCa-2, PANC-1, BxPC-3, AsPC-1, Capan-1, and Colo357; the human colorectal carcinoma cell line HCT116; and the human embryonic kidney cell line HEK293T were obtained from the American Type Culture Collection (ATCC) and cultured under conditions recommended by ATCC. Pancreatic adenocarcinoma cell line, MDA-Panc-28, was obtained from the laboratory of Dr. Evans[49]. PDAC cell lines MDA-PATC43, 50, 53, 66, 69, 102, 107, 108, 124, 148, 148LM, 148LM2, 153, 153LM, 216, and 219B, which were established from PDXs, were provided by Dr. Jason B. Fleming. Of these cell lines, MDA-PATC53, 148, 148LM, 148LM2, 153, 153LM, and 219B

were derived from liver metastases as described previously[28]. Immortalized nontumorigenic HPDE and HPNE cells were described elsewhere[50,51]. Tumorigenic HPDE/Kras$^{G12V}$/Her2/shp16shp14/shSmad4 cells were established in Dr. Chiao's laboratory and were cultured as described previously[29]. KIC and KPC mouse-derived cell lines were generated independently in house from pancreatic tumors harvested from $p48$-cre;Kras$^{LSL-G12D}$;INK4a$^{F/F}$ and $p48$-cre;Kras$^{LSL-G12D}$;p53$^{LSL-H172R}$ mice[8,12]. PTEN$^{+/+}$ and PTEN$^{-/-}$ MEFs were gifts from Dr. Min Sup Song (MD Anderson). CEM leukemia cells were cultured in RPMI 1640 medium with 10% fetal bovine serum.

To establish normal mouse pancreatic epithelial cell lines, pancreas specimens were collected from p48-cre mice and minced into pieces smaller than 1 mm$^3$, dissociated using collagenase type V (Sigma) at 37 °C for 25 min, and digested with trypsin-ethylenediaminetetraacetic acid at room temperature for 5 min. Cells were then resuspended in Dulbecco's modified Eagle's medium supplemented with 5% Nu Serum IV (BD Biosciences), 25 µg/ml bovine pituitary extract (Corning), 0.5% ITS+ premix (BD Biosciences), 20 ng/ml epidermal growth factor (BD Biosciences), 100 ng/ml cholera toxin (Sigma), 5 nM 3,3',5-triiodo-L-thyronine (Sigma), 1 µM dexamethasone (Sigma), 5 mg/ml glucose (Sigma), and 1.22 mg/ml nicotinamide (Sigma)[52].

P72Plk3-WT, Plk3-heterozygous, and Plk3-KO primary MEFs were generated from 13.5-day-old (E13.5) mouse embryos. Briefly, freshly isolated fetuses from mouse embryos were minced with sterile fine forceps, digested with trypsin-ethylene-diamine tetraacetic acid for 30 min at 37 °C, and cultured in Dulbecco's modified Eagle's medium supplemented with 10% fetal bovine serum, 0.1 mM β-mercaptoethanol, and 0.5% penicillin (50 U/ml)/streptomycin (50 µg/ml; Invitrogen Gibco). These cells were immortalized using the standard 3T3 protocol[53].

All cell lines were authenticated using short tandem repeat fingerprinting at the MD Anderson Characterized Cell Line Core before use. All cell lines were tested and found to be free of mycoplasma contamination.

### Cloning procedures

Full-length human Plk3 (1-646 aa; accession number NM_004073) construct was generated in the laboratory of Dr. Chiao as described previously[17]. Deletion mutants of Plk3 (Plk3$^{1-306}$, Plk3$^{1-325}$, Plk3$^{1-340}$, Plk3$^{1-353}$, Plk3$^{354-646}$, Plk3$^{1-356}$, Plk3$^{1-362}$, Plk3$^{1-376}$, Plk3$^{1-468}$, Plk3$^{1-557}$, Plk3$^{Δ306-377}$, and Plk3$^{Δ326-377}$) were generated via PCR, and a p72Plk3 point mutant (Plk3$^{R354G}$) was generated via site-directed mutagenesis using a QuikChange Lightning site-directed mutagenesis kit (Agilent Technologies). All mutants were subcloned into the pCMV-Tag2A vector. Inducible full-length Plk3 and p41Plk3 were constructed by subcloning the corresponding coding regions of Plk3 into a pENTR/D-TOPO entry vector for Gateway cloning (Life Technologies) followed by an LR recombination reaction using Gateway LR Clonase II enzyme mix (Life Technologies) to transfer the coding regions into the Tet-inducible lentiviral vector pInducer20 (#44012, Addgene).

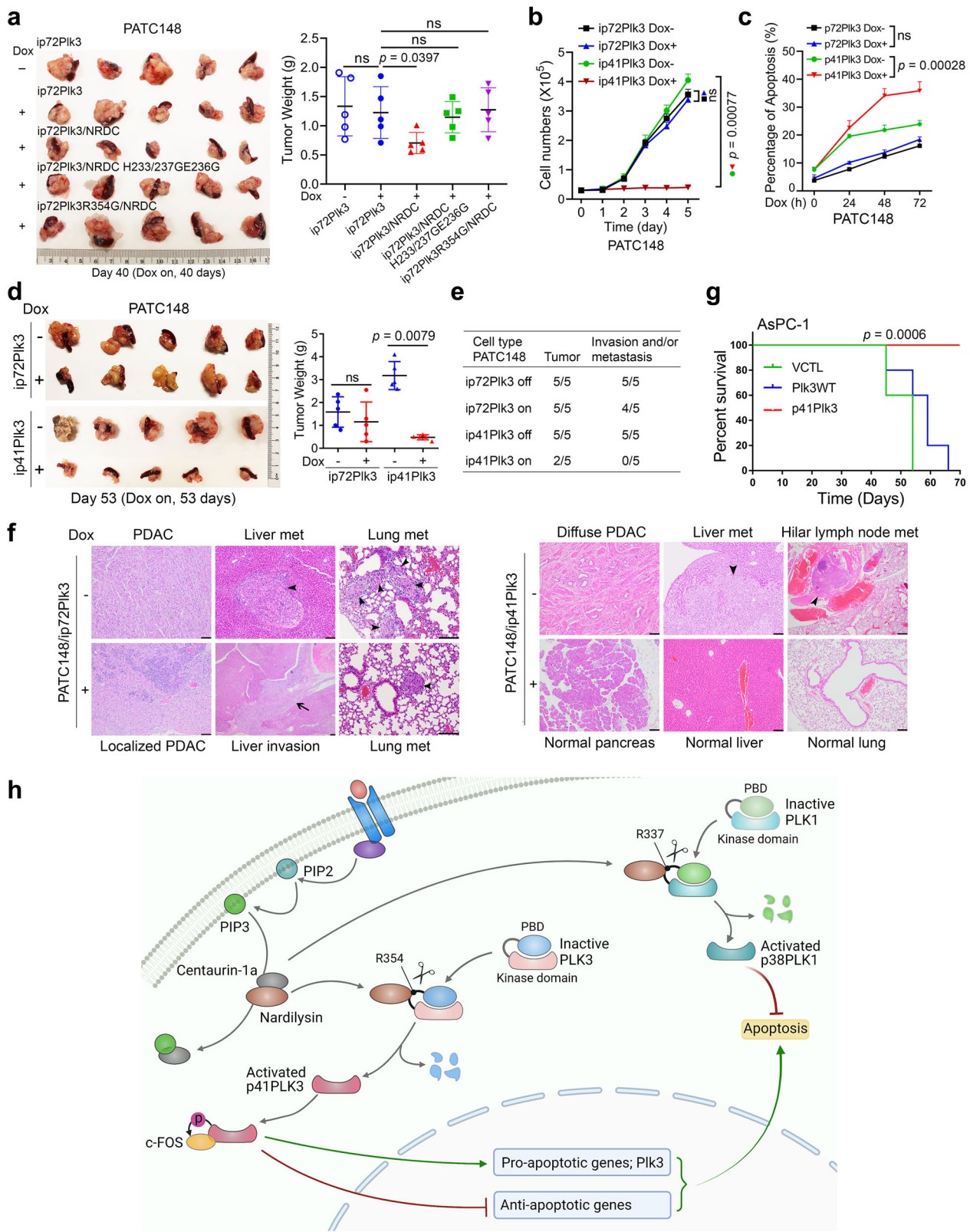

p41Plk3 mammalian expression vector for protein expression and purification was generated by performing an LR recombination reaction between the pENTR/D-TOPO entry clone and a Gateway destination vector pcDNA-DEST40 containing His-tag (Life Technologies). ShPlk3s (shRNA-A: 5'-GATCCTTTCTGGCCTCAAGTA-3'; shRNA-B: 5'-CGGCCTCATGCGCACATCCGT-3'), the Plk3 deletion mutant Flag-tagged Plk3[1-353], Flag-tagged p72Plk3* mutants (with

silent mutations of Plk3 to avoid shPlk3-mediated degradation), and p72Plk3KM (with mutated NRDC sites) for rescuing experiments were cloned into a lentiviral vector. ShRNA-A was combined with shRNA-B to ensure efficient knockdown of Plk3 expression. To generate shRNA-insensitive p72Plk3 (Plk3*) constructs, the underlined silent mutations in 5'-GGTGAGGGCCTCATGAGGAC-CAGTGTTGGGCATCAGGATGCCAGGCC-3' were introduced into the

**Fig. 7 | Activated p41Plk3 suppressed PDAC tumorigenesis and metastasis.**
**a** Left, pancreatic tissues or tumors removed on day 40 from nude mice (n = 5 mice/group) orthotopically injected with PATC148 cells (1 × 10⁵ cells) harboring indicated p72Plk3, NRDC, and mutants. Right, tumor weight analysis. **b, c** Growth curve (**b**) and flow cytometry analysis (**c**) in PATC148 cells with inducible expression of p72Plk3 and p41Plk3 under the suspension culture at the indicated times. **d** Left, pancreatic tissues or tumors removed on day 53 from nude mice (n = 5) orthotopically injected with PATC148 cells (1 × 10⁵ cells) with inducible expression of p72Plk3 or p41Plk3. Right, tumor weight analysis. **e, f** The rates of tumor formation, invasion, or metastasis (**e**) and hematoxylin and eosin stains of tissues and lesions

(**f**) of the indicated groups in **d**. Arrowheads indicate liver or lung metastasis; arrow shows liver invasion. Bar, 100 μm. **a, d–f +, on:** mice were fed with Dox-containing water upon cell inoculation and continued for the indicated times. **-, off:** mice were maintained Dox-free. **g** Kaplan-Meier analysis of mice with AsPC-1 cells that grew for 2 weeks in a subcutaneous xenograft, after which liposome-mediated gene delivery of the indicated expression vectors was performed. VCTL, vector control (n = 5 mice/group). p value was calculated by Log-rank test. **h** The proposed working model of Plk3 and Plk1 activation. Error bars, Mean ± SD (**a, d**) or ± SEM (**b, c**), n = 3 independent experiments (**b, c**), two-tailed unpaired t-test (**a–d**). Source data are provided as a Source Data file.

shRNA-B target sequence in the p72Plk3 coding region. To construct Plk3$^{R354G}$ and p72Plk3*KM mutants that resisted NRDC cleavage, the following primers were used for site-directed mutagenesis (the mutations are underlined): 5′-CGGG AGCCTCTTTGGCG-CAGCGGCCGCGAGTGCG-3′ (Plk3$^{R354G}$), 5′-CACCCCCCAACCC AGCTGCGAGTCTGTTTGCCAAAG-3′ (Plk3KMp1), 5′-GCCAAAGT-TACCGGGAGCCT CTTTGGCGC-3′ (Plk3KMp2), and 5′-CCGGGAGCCTCTTTGGCGCAGCGGCCGCGA GTGCGAATCATGCC-CAGGAG-3′ (Plk3KMp3).

## Recombinant protein
Recombinant GST-p72Plk3 was obtained from ProQinase (Germany), GST–c-Fos was purchased from Abcam, and His-p53 was obtained from R&D system.

## Structural modeling of Plk3
Human Plk3 was modeled by Rosetta structural prediction server (https://robetta.bakerlab.org/) using comparative modeling[54]. The Plk1 ortholog from zebrafish (PDB: 4j7b), which contains a protein complex of kinase domain and PBD with ~44% sequence identity, was chosen to model the human Plk3. The folding of kinase domain and PBD of Plk1 ortholog has the RMSD of 0.98 Å and 0.93 Å, respectively, to human Plk structures [PDB: 4b6l (Plk3 kinase domain) and 4rs6 (Plk2 PBD)]. The missing residues are modeled in based on the secondary structure prediction in the Rosetta program. The comparative modeling parameters have set as sample sequence register shifts up to 1 and sample fragments within template regions with probability of 0.1. The top 5 models were picked and one is represented in Fig. 5a. Notably, all five models share the common predicted helix-loop-helix secondary structure for the region containing NRDC cleavage site (GRKKK).

## Histopathological analysis
All major murine organs and tissue samples and human pancreatic tissue samples were fixed overnight in 10% formalin. Paraffin-embedded sections were stained with hematoxylin and eosin at the MD Anderson Research Histology Core Laboratory. For IHC studies, standard procedures were carried out according to the manual for the VECTASTAIN Elite ABC kit (Vector Laboratories), and the reactions were developed using a DAB Peroxidase Substrate Kit (Vector Laboratories). The primary antibodies used in IHC were an anti-Plk3 antibody (NBP2-32530; Novus Biologicals) at 1:100 dilution for mouse tissues and 1:150 for human tissues; an anti–Ki-67 antibody (Thermo Scientific) at 1:200; an anti-cleaved caspase-3 antibody (9661; Cell Signaling Technology) at 1:250.

For human tissue microarrays, IHC staining was scored semi-quantitatively in a blinded fashion by two gastrointestinal pathologists (H. Wang and J. Zhao) on a scale of 0 (no staining) to 3 (strongest staining) based on the intensity of reactivity. For mouse immunohistochemistry, whole slide scanning was carried out using a Vectra 3 automated quantitative pathology imaging system at the MD Anderson Flow Cytometry and Cellular Imaging Facility. Image and quantification analyses were performed using the inForm software program (version 2.2; PerkinElmer). By setting a tissue segment training algorithm, the immunoreactivity score was recorded by incorporating both

the intensity (ranked into four groups: +3, +2, +1, and 0) and number of positive cells per unit of tissue surface area.

## Lentiviral-based gene transduction
293T cells were transfected with corresponding shRNA or Dox-inducible constructs together with lentiviral packaging plasmids pCMV-dR8.2 and pCMV-VSV-G to produce lentivirus. Cells were infected via incubation with a virus and 10 μg/ml hexadimethrine bromide (Polybrene) followed by selection with antibiotics for 14 days. Pooled cell populations were used for various experiments. For all experiments involving the Dox-inducible lentiviral constructs, HPNE cells were treated with 1 μg/ml Dox for 24, 48, or 72 h. The expression levels for Plk3, various Plk3 mutants, and NRDC were determined using qRT-PCR or immunoblotting.

## RNA extraction from human tissues and RT-PCR
The pancreatic tumor and normal pancreatic tissue samples were ground in liquid nitrogen and homogenized fully in TRIzol reagent (Life Technologies) and liquid nitrogen, and then phenol/chloroform and 70% ethanol were added. Total RNA was isolated using an RNeasy kit (QIAGEN). RT-PCR analysis of various mRNAs was performed in 28 PCR cycles using a pair of Plk3 primers (5′-CCTGCCGCCGGTTTCCTG-3′ and 5′-GGCCTCAGGGCTTG GGTC-3′) and were separated on a 1% agarose gel.

## Real-time RT-PCR analysis
Total RNA was extracted from cells or tissue samples using TRIzol reagent (Life Technologies) as described by the manufacturer. For cDNA synthesis, 1 μg of total RNA was reverse-transcribed using iScript RT Supermix (Bio-Rad). A real-time PCR was run in triplicate using iTaq Universal SYBR Green Supermix (Bio-Rad) following the manufacturer's instructions. Real-time monitoring of PCR amplification was performed using a CFX96 real-time system (Bio-Rad). The resulting data were expressed as relative mRNA levels normalized according to the β-actin expression level in each sample and presented as means ± standard error of the mean from three independent experiments unless otherwise indicated in the figure legends. The primer sequences used in the real-time RT-PCR analysis are listed in Supplementary Table 3.

## Flow cytometry for cell-cycle analysis
For cell synchronization, cells were synchronized at the G1/S boundary by a double-thymidine (T/T; 2 mM) treatment (18 h of thymidine arrest and 8 h of release, followed by 18 h of thymidine arrest) and then released into normal medium. For preparing cells arrested in mitosis, cells were grown in 400 nM nocodazole (Sigma) for 24 h, washed, and then released into nocodazole-free media at the indicated times. Cells were fixed at −20 °C in 70% (v/v) ethanol for 30 min, before treatment with RNase A (10 μg/ml) at 37 °C for additional 1 h. Cells were resuspended in PBS containing propidium iodide (10 μg/ml in PBS) and subjected to flow cytometry. For BrdU incorporation assays, 10 μM of BrdU (550891; BD Bioscience) were added to the cell culture medium for 30 min or 1 h before harvesting. Samples were treated with 2 M HCl and 0.1 M sodium borate to neutralize any residual acid, incubated

with FITC-conjugated anti-BrdU antibody, and then analyzed by flow cytometry with a BD FACSCanto II system (BD Biosciences). Data are shown as mean ± standard deviation and are representative of three or four independent experiments. Data were analyzed using the FlowJo software program (version 10.0.8; Tree Star).

## Immunoprecipitation

Protein (1 mg) extracts from cells transfected with Flag-Plk1, Myc-NRDC, or Flag-tagged WTPlk3 and Plk3 mutants were subjected to IP with 40 µl of anti-Flag (M2) Magnetic Beads (M8823, Sigma) or anti–c-Myc Magnetic Beads (88842; Thermo Scientific) at 4 °C overnight followed by Western blotting with anti-Plk1 antibody (4513, Cell Signaling Technology), anti-NRDC (A-6) (sc-137199; Santa Cruz Biotechnology), or anti–c-Fos antibody (2250, Cell Signaling Technology).

## Colony-formation assay

A vector control plasmid and WT and mutant Plk3 expression plasmids were transfected into PANC-1 or 293T cells using FuGENE 6 transfection reagent (Roche) as described previously[17]. Stained colonies of these cells were lysed with 1% sodium dodecyl sulfate in phosphate-buffered saline. Colony lysates were diluted in 0.1% sodium dodecyl sulfate and examined for their A615 values.

## Wound healing assay

Equal numbers of cells were seeded into 6-well tissue culture plates; 24–48 h later, when the cell reached confluence, an artificial homogeneous "scratch" was created onto the monolayer with a sterile plastic 200 µl micropipette tip. At 12–20 h after wounding, the debris was removed by washing the cells with serum-free medium. Migration of cells into the wound was observed at different time points. Cells were visualized and photographed under an inverted microscope (40× objective) (Leica, Solms, Germany).

## Transwell migration assay and invasion assay

Cell migration and invasion were performed by using BioCoat Control Cell Culture Inserts and BioCoat Growth Factor Reduced Matrigel Invasion Chambers (BD Biosciences), respectively. For migration assays, $5 \times 10^4$ cells were plated in the upper chamber with the uncoated membrane (24-well insert; pore size, 8 µm). For invasion assays, $1.25 \times 10^5$ cells were plated in the upper chamber with Matrigel-coated membrane (24-well insert; pore size, 8 µm). In both assays, cells were plated in medium without serum or growth factors, and complete medium (serum containing) was used as a chemoattractant in the lower chamber. After incubation for 20 h, cells that did not migrate or invade through the pores were removed by a cotton swab. Cells on the lower surface of the membrane were fixed with 4% paraformaldehyde, stained with 0.5% crystal violet, and microscopically counted from three random fields of each membrane. The average cell number per field for triplicate membranes was used to calculate the mean with SD.

## Nonbiased functional cloning of cDNAs for inhibition of Plk3-mediated apoptosis

A premade human placenta plasmid cDNA library in the pCFB XR vector (ViraPort XR) was purchased from Stratagene. Retroviruses were produced in 293 T cells followed by infection of target 293T cells according to the manufacturer's instructions. Twenty-four hours later, pCMV-Tag2A-p72Plk3/neo constructs were transfected into retrovirus-infected 293T cells using FuGENE 6. G418 was added to the transfected cells to select stable surviving colonies. About 60 neomycin-resistant colonies grew from an initial group of plated $2 \times 10^6$ cells (0.001%), and six colonies were selected for measurement of p72Plk3 expression using Western blot analysis. Genomic DNAs were extracted from Flag-Plk3–positive colonies, and PCR was performed to recover the cDNA insert using 5′Retro (5′-GGCTGCCGACCCCGGGGGTGG-3′) and 3′pFB (5′-CGAACCCCAGAGTCCCGCTCA-3′) primers under standard cycling

conditions. The recovered cDNA inserts were sequenced and compared with the GenBank nucleotide database using the Basic Local Alignment Search Tool (National Center for Biotechnology Information) to identify related genes. Sequencing analysis of the cDNAs isolated from the surviving colonies in our functional screening revealed partial sequences and the identities of the cDNAs, which are listed in Supplementary Table 1.

## NRDC enzyme preparation and Plk3-p21 peptide digestion analysis

293T cells were washed three times with phosphate-buffered saline and lysed with a lysis buffer (50 mM Tris-HCl, pH 7.5, 150 mM NaCl, 1% NP-40, 1% Triton X-100) on ice for 30 min. Lysates of these cells were cleared via centrifugation at $20,000 \times g$ for 30 min at 4 °C. Supernatants were immunoprecipitated with an anti-NRDC antibody (A-6) (sc-137199; Santa Cruz Biotechnology). The immunoprecipitated NRDC proteins were analyzed using a sodium dodecyl sulfate–polyacrylamide gel electrophoresis (8%) gel and silver staining or Western blotting and used immediately in enzymatic assays or stored at 4 °C in 20 mM potassium phosphate (pH 7.2) containing 10% glycerol. The synthetic Plk3-P21 peptide (WT P21, 345-365 aa [AKVTKSLFG**RKKK**SKNHAQER]; from the predicted NRDC binding and processing region) and a mutated peptide (P21mt [AKVTKSLFG**MMMM**SKNHAQER]) were used in an NRDC cleavage assay. Dynorphin A (Sigma-Aldrich) was used as a cleavage-positive control. Enzymatic assays were carried out via incubation at 37 °C for 10 min of an aliquot of the enzyme preparation with Plk3-peptide or dynorphin A (12 µM) in 20 mM potassium phosphate (pH 7.0) in 100 µl of a reaction volume. The reaction was terminated by adding 10 µl of acetic acid. Digestion products were separated and identified via mass spectrometry with a MALDI–time-of-flight Voyager DE-Pro instrument (PE PerSeptive Biosystems) using the matrix 3,5-dimethoxy-4-hydroxy-cinnamic acid (Sigma-Aldrich).

## MALDI methods

Samples from cleavage reactions were diluted at 1:100 in 0.1% trifluoroacetic acid in water, and the dilution was mixed at 1:1 with a matrix solution and spotted onto a stainless steel MALDI target plate. Samples were spotted near calibration standard spots to facilitate close external calibration. The matrix employed was α-cyanohydroxycinnamic acid obtained in solution from Agilent Technologies. Mass spectra were acquired using a Voyager DE-STR MALDI–time-of-flight mass spectrometer (software version 5.1; Applied Biosystems) in reflector mode. Typically, 200 laser shots were summed per acquisition, with 2–5 acquisitions added to provide the final spectra.

## In situ proximity ligation assay

An in situ proximity ligation assay was performed to identify protein-protein interactions in cells using a Duolink In Situ Red Starter Kit (Sigma) per the manufacturer's protocol. HPNE-Tet-p72Plk3 and HPNE-Tet-p41Plk3 cells were seeded on coverslips, fixed with 4% paraformaldehyde, and permeabilized with 0.1% Triton X-100. After blocking with Duolink blocking buffer, slides were incubated with rabbit anti-Plk3 (ab123695, 1:150; Abcam) and mouse anti–c-Fos (sc-271243, 1:200; Santa Cruz Biotechnology) antibodies in a humidifying chamber at 4 °C overnight. The slides were then washed and incubated with secondary anti-rabbit and anti-mouse Duolink proximity ligation assay probes at 37 °C for 1 h. After washing, slides were incubated with a ligation-ligase solution (30 min at 37 °C) and incubated with an amplification-polymerase solution (2 h at 37 °C). Slides were then mounted using Duolink mounting medium with 4′,6-diamidino-2-phenylindole. Fluorescence signals in the cells were identified under a fluorescence microscope (Carl Zeiss) using 63× objectives and quantified in dots per cell.

## Confocal microscopy

Cells were cultured in chamber slides overnight and fixed in 4% paraformaldehyde for 15 min at 4 °C, permeabilized with 0.5% Triton X-100 for 15 min. Cells were then blocked with 5% BSA in PBS and 0.1% Tween-20 (TBST) for 1 h. After the incubation with primary antibodies overnight at 4 °C, cells were then further incubated with anti-mouse IgG (H+L), F(ab′)$_2$ fragment (Alexa Fluor 488 Conjugate, A21206, Life Technologies) or anti-rabbit IgG (H+L), F(ab′)$_2$ fragment (Alexa Fluor 594 Conjugate, A21207, Life Technologies) for 1 h at room temperature. Nuclei were stained with anti-fade mounting medium with DAPI (Invitrogen). Confocal fluorescence images were captured using a Zeiss LSM 710 laser microscope. In all cases, optical sections were obtained through the middle planes of the nuclei, as determined with use of nuclear counterstaining.

## siRNA, shRNA, and CRISPR knockdown

C-Fos siRNA and scramble RNA (negative control) were purchased from Santa Cruz Biotechnology (SC-29221), and 20 nM siRNA was transfected into HPNE cells with Lipofectamine RNAiMax reagent (Life Technologies). ON-TARGETplus SMARTpool human Plk3 siRNA (L-003257-00-0005) and ON-TARGETINGplus Non-targeting pool (D-001810-10-05) were obtained from Dharmacon. The knockdown efficiency and specificity of all siRNAs were validated with qPCR or immunoblotting. shRNA constructs that had been cloned into the pGIPZ lentiviral vector were purchased from Dharmacon. The *Plk3* shRNA sequences were as follows: shPlk3#1 (TTTCTCAGAGCACAAAGGG) and shPlk3#3 (TGAAGAGCACAGCCACACG). The *Plk1* shRNA sequences were as follows: shPlk1#1 (ATTCTGTACAATTCATATG) and shPlk1#3 (AATTAGGAGTCCCACACAG). A non-targeting shRNA (shCtrl) was used as a control. For CRISPR knockdown of *NRDC* and *Plk3*, sgRNA were designed based on the recommendations from Doench et al.[55] (http://portals.broadinstitute.org/gpp/public/analysis-tools/sgrna-design-v1). The sgRNA were then cloned into lentiCRISPR v2 (Addgene). The NRDC sgRNA sequences were as follows: sgNRDC-5_sense: CACCGCGCTGATCCAGATGACCTGC; sgNRDC-5_antisense: AAACGCAGGTCATCTGGATCAGCGC; sgNRDC-7_sense: CACCGAAGTTGAAGCTGTTGATAG; sgNRDC-7_antisense AAACCTATCAACAGCTTCAACTTC. The Plk3 sgRNA sequences were as follows: sgPlk3-1_sense: CACCGAGTGACCATACATCCGCCTC; sgPlk3-1_antisense: AAACGAGGCGGATGTATGGTCACTC; sgPlk3-2_sense: CACCGCGAAAAACGCACGATGTGG; sgPlk3-2_antisense: AAACCCACATCGTGCGTTTTTCGC.

## ChIP assay

Cell fixation and chromatin preparation was performed using SimpleChIP Plus Enzymatic Chromatin IP Kit (Cell Signaling Technology). Real-time PCR primers (site 1: forward, CAGGTTGGTCTCGAATTCCT; reverse, AGACACAGAGCTTGTCTCAAA; site 2: forward, GGCCTGGAGCATGGTAAA; reverse, CTTTAGGGCCTGGAAGGAAG; site 3: forward, AGGCGGACAGGGATCAG; reverse, CATTCCCGGCCTTACATCAC) were used to amplify the corresponding region in *Plk3* promoter containing the three c-Fos binding sites.

## In vitro kinase assay

Purified recombinant p41Plk3-His or GST-p72Plk3 with or without Plk inhibitor were incubated with recombinant GST−c-Fos or His-p53 in Kinase Assay Buffer supplemented 10 μCi of [γ-$^{32}$P] ATP for 1 h at 30 °C. The reaction was stopped by adding SDS-PAGE sample buffer and heating the sample for 6 min at 95 °C. Samples were analyzed by SDS-PAGE followed by Coomassie blue staining or autoradiography. Some kinase reaction was performed by substituting ATP-gamma-S (Abcam) in place of ATP, followed by alkylating the protein with 1.5 μl of the 50 mM PNBM stock (Abcam) at room temperature for 2 h. The kinase substrates containing thiophosphate esters were then detected by α-Thiop RabMAb (Abcam).

## Statistical analysis

In all experiments, standard deviations were calculated for three or four independent experiments and presented as error bars, with the values representing means ± standard deviation. For statistical comparison of values between more than two groups, one-way analysis of variance and the Newman-Keuls multiple comparison test were used. All other differences in datasets were evaluated using the Student *t*-test. Survival curves were generated using the Kaplan-Meier method and assessed using a log-rank test. The sample size was chosen based on the need for sufficient statistical power. Differences were considered statistically significant at p values less than or equal to 0.05, 0.01, or 0.001. Statistical analysis was performed using the Graph Pad Prism 10.0.3 software program (GraphPad Software; RRIC:SCR_000306).

## Reporting summary

Further information on research design is available in the Nature Portfolio Reporting Summary linked to this article.

## Data availability

Authors can confirm that all relevant data are included in the paper and/or its Supplementary Information files. Source data are provided with this paper. Plk3 expression in the Segara and Logsdon cancer microarray datasets are obtained from Oncomine (https://www.oncomine.com/). TCGA gene expression data were obtained from The Cancer Genome Atlas data portal (https://tcgadata.nci.nih.gov/tcga/dataAccessMatrix.htm)[27]. The Mass-spec data generated for this study are deposited in PRIDE under accession PXD042993. The following PDB were used for modeling Plk3 structure: The Plk1 ortholog from Zebrafish (PDB:4j7b), Plk3 kinase domain (PDB: 4b61), and Plk2 Polo-box domain (PDB: 4rs6). Source data are provided with this paper.

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

## Acknowledgements

*Peter Stambrook*, PhD, former chair of the Department of Cancer and Cell Biology and 2013 Daniel Drake Medalist, died Oct. 1, 2020. We are thankful to Dr. Stambrook for generously providing us with the Plk3 knockout mouse model. We thank Stephanie P. Deming and Sarah Bronson for editorial assistance, Libing Mu for assistance with graphics. We also thank Dr. Helen Piwnica-Worms for suggestions on experiment design, and Dr. Min Sup Song for his discussion and comments on cell-cycle study. This work was supported in part by grants from the NCI (CA2070313 and CA140410 to P.J.C.), the University of Texas, MD Anderson Cancer Center bridge funding to C.L., National Science and Technology Council Taiwan (NSTC 112-2639-B-039-001-ASP and T-Star Center NSTC 113-2634-F-039-001 to M-C.H.), Ministry of Health and Welfare Taiwan (MOHW113-TDU-B-222-134016 to M.-C.H.), The Featured Areas Research Center Program by the Ministry of Education (MOE) in Taiwan to M-C.H., and NIH Cancer Center Support Grant P30CA016672 to MD Anderson Cancer Center (Functional Genomics Core, Flow Cytometry and Cellular Imaging resource, Advanced Technology Genomics Core, Research Histology Core Laboratory, Cytogenetics and Cell Authentication Core, Functional Proteomics Reverse Phase Protein Array Core, and Research Animal Support Facility-Houston). J.A.T efforts are supported by Cancer Prevention Research Institute of Texas (CPRIT) RP180813, by National Institutes of Health (NIH) grants P01 CA092584 and R35 CA220430, and by a Robert A. Welch Chemistry Chair (G-0010). C-F.L. received Odyssey Fellowship from The University of Texas MD Anderson Cancer Center.

## Author contributions

J.F., J.L., C.L. and P.J.C. conceived and designed the study. J.F. and J.L. performed, and/or analyzed most of the experiments, with participation of C.-F.L., W.Y., J.Hou, P.C., Y.S., X.X., Z.J., K.F., Y.L., M.W. and J.Huang. J.L., Y.C., Q.H., E.B. and P.J.S. generated or performed experiments with genetically engineered mouse models. Y.K. and J.B.F. provided patient-derived xenograft (PDX) cell lines. C.-L.T. and J.A.T. analyzed protein structure. D.H.H. and B.-F.P. executed mass spectrum analysis. Clinical specimens were ascertained and processed by H.W. The histological analysis was performed by J.Z. and H.W. J.Y. performed bioinformatic and computational analyses. A.M., M.-C.H. and P.H. provided intellectual and material help. J.F., C.L. and P.J.C. conceptualized the study and wrote and edited the manuscript.

## Competing interests

The authors declare no competing interests.
