## [Peer Review File · Nature Communications]

REVIEWER COMMENTS

Reviewer #1 - Anoikis metastasis - (Remarks to the Author):

In this manuscript, Fu et al. describe a role for decreased Plk3 expression in the formation and progression of PDAC as well as elucidate the mechanism by which Plk3 is cleaved and activated. The authors find that Plk3 expression is decreased in human PDAC compared with normal pancreatic tissues, and that genetic knockout of PDAC increases PDAC development in mice. Furthermore, Plk3 is cleaved in its DR1 site by nardilysin (NRDC) activity, which subsequently results in an elevation in Plk3 kinase activity. Specifically, a 41 kDa form of Plk3 that contains the kinase active site in its n-terminal domain, known as p41Plk3, is sufficient to increase apoptotic death by phosphorylating c-Fos on T164 to direct it to the nucleus where it increases transcription for pro-apoptotic genes. Overexpression of this p41Plk3 subunit is also sufficient to decrease PDAC tumor burden and invasion in mice.

While the study is of interest, there are multiple concerns, both major and minor, that should be addressed prior to publication.

Major concerns

1. In general, some conclusions made by the authors are overstated or otherwise not well supported by the data. For example, the figure caption of Fig. 2 states that “proteolytic processing of p72Plk3 to p41Plk3 is essential for induction of anoikis”. However, the data as presented are merely correlative and indicate that p41Plk3 levels are associated with anoikis induction. Causality is never demonstrated.
2. The conclusions for the role of Plk3 in anoikis induction and the mechanism by which Plk3 is proteolytically processed rely heavily on data generated using non-pancreatic cell lines. This weakens the main connection that the authors attempt to make between pancreatic tissues and endogenous Plk3.
3. Similarly, many of the IP experiments utilize exogenous protein without any attempt to assess protein-protein interactions using endogenous proteins. This is not to say that there is no role for the IP data with exogenously expressed protein. Rather, the conclusions are overly reliant on these approaches.
4. Appropriate controls are frequently lacking to confirm overexpression, activity of various inhibitors, and immunoprecipitation. For example, the expression of p110 α in Fig. 4i is not confirmed by an immunoblot, and there is no positive control for LY294002 inhibitor activity in Fig. 4. IP experiments often do not include controls for non-specific binding.
5. In Figure 1b, some of the cell lines labeled on the immunoblot are not mentioned or discussed in the manuscript. This creates difficulty in assessing the conclusions regarding Plk3 levels and the level of aggressiveness of each PDX model.
6. CBS loading control band patterns appear substantially different between immunoblots using the same cell lines. For example, compare CBS banding in Fig. 2d & 2e which are both from HPDE cells.

7. The FACS quantifications of apoptotic cells in Fig. 3f-g do not seem to match with the results of the colony formation assay with the same cell lines in Fig. 3d. If there are changes in cell death that are only 10-20%, why is it that colony growth is changing by (in some cases) 90%?
8. In Figure 4b, the delta 326-377 protein expression (seen in the input) is substantially lower than the WT or R354G. Thus, the conclusion that the delta 326-377 protein binding to NRDC is compromised is not well supported. It could be that the differences in total protein account for the IP results.
9. The organization of figure panels throughout the manuscript is muddled and creates difficulty in assessing the findings. Similarly, the figures are not always referenced in the text in order which also creates challenges for readers/reviewers.
10. Explanation of intent, description of findings, and interpretation of results are often unclear. It should be easily understood why each experiment was performed, what was found, and how the data support or contradict the hypothesis.

Minor concerns

1. Frequently used abbreviations (like PanIN, ORF, PBD) are often not described throughout the paper.
2. The introduction section could contain more information about nardilysin, as it becomes a focal point of the data after not being introduced early in the text.
3. Throughout the paper, Plk1 data are intermingled in figures without much context or justification, especially when the main conclusions relate to Plk3.
4. Figure 1e & 1f are redundant ways to present identical data.
5. The cleavage site in Figure 5a is difficult to spot at first glance; the scissors blend in too much with the ribbon model.

Reviewer #2 - PDAC, metastasis, mouse models - (Remarks to the Author):

Title: Nardilysin-Regulated Scission Mechanism Activates Polo-like Kinase 3 to Suppress the Development of Pancreatic Cancer

Manuscript Number # NCOMMS-22-33695

Reviewer's Comments: The focus of this research study indicates the role of Plk3 and its cleaved form of p41KDa mediated anoikis. When p72Plk3 is cleaved by NRDC and p32 Plk3 is removed by degradation, the N-terminal p41Plk3 gains kinase activity. p41Plk3 biological functions include induction of apoptosis and exhibit tumor suppressor role. The findings also demonstrate that PI3K regulates post-translational processing of Plk1 and Plk3 involving PI3K/centaurin-1/NRDC signaling axis to generate functional p38Plk1 and p41Plk3.

There are some major concerns about the data presented:

1. The focus is on pancreatic cancer, but many results were obtained from 293T human embryonic kidney cell line (Figure 3, 4, 5, 6 and Supplementary Figure 2, 4) and human colorectal carcinoma cell line HCT116 (Supplementary Figure 2, 4).
2. Figure 1b-1d: what are the levels of p41Plk3 in the metastatic cell lines? Supplementary figure 1c: whether β -actin is analyzed on the same membrane? What are the levels for p41Plk3?
3. In figure 1b: among the cancer cell lines, 102, 108, and 124 also express a similar level of p72Plk3 as normal HPDE, while the expression of p72Plk3 appears to be relatively low in metastatic cell lines. So, whether loss of Plk3 is involved in metastasis specifically?
4. What is the status of p41Plk3 in the KPC mice in Figure 1d? Figure 1h: is there an IHC staining done to show the expression of p41Plk3 in p48- cre; KrasLSL-G12DPlk3+/+.
5. In figure 1, most of the pancreatic cell lines still express a detectable p72Plk3 level. How do you justify this? Whether it determines the possibility of Plk3 independent tumor mechanism in PDAC or if there is any possible mechanism involved in inhibiting the function of p72Plk3. What's the expression pattern of NRDC in pancreatic cancer?
6. In figure 2b – cleaved PARP induction is visibly seen after 2 hrs in suspension culture. Whereas figure 2i, 2j, 2k -HPDE/HEK293T/MEF cell lines reveal the cleavage to 41kd from 72 takes at least 3 hrs. Whether the induction of cleaved PARP is independent of 41kd?
7. The mouse developed a tumor in p48- cre; KrasLSL-G12DPlk3+/+ group, did it show any correlation with the genes mentioned in supplementary Table 1? Did authors analyze those samples because they should match with the genes mentioned in the surviving colonies that are resistant to Plk3 overexpression?
8. What causes the Plk3 downregulation? Is there any known mutation? And what percentage of PDACs in which Plk3 is downregulated? If Plk1 seems to be an oncogene, is there an interplay between the two family members?
9. Co-IP of NRDC indicated that LY294002 increased the binding of NRDC to centaurin-1 in the cytoplasm and decreased its binding to p72Plk3 (Fig. 4i). But it was shown only in 293T, which is a human embryonic kidney cell line, and not in any other PDAC cell line. Though the experiments with LY294002 were performed in many different cell lines, the Western for centaurin-1 was shown only in 293T cells.
10. Figure 5i: What is the protein half-life of plk3NT/plk3CT in the metastasis cell lines? Whether different antibodies are used for detecting NT-1-354 and CT-356-646?
11. Supplementary Figure 7h - please mark the different cell cycle phases and include population distributions.
12. Authors have found that p41Plk3 expression is significantly increased in the PTEN-KO compared to PTEN-WT MEF cells. It is also suggested that PI3K is required to cleave p72Plk3 into its active form p41Plk3 (apoptosis inducer).
- If p72Plk3 is a tumor suppressor and its active form p41Plk3 is anoikis or apoptosis inducer, then the concern is about PI3K because PI3K oncogenic signaling is known to be upregulated in PDAC. PTEN which

is a known tumor suppressor KO increased p41Plk3 expression. Please justify how to correlate these events.

13. Plk1 cleavage is markedly increased in PTEN-KO but not in PTEN-WT MEFs. If Plk1 is a tumor promotor. It's been known that PTEN is a negative regulator of PI3K and PI3K is a known downstream effector of K-Ras, which is a driver of PDAC. Then how the KO of PTEN increases the cleavage of Plk3 into the active pro-apoptotic p41Plk3 kDa form? And vice versa how the inhibition of PI3K by LY294002 prevents the cleavage of p41plk3, which can induce apoptosis?

14. Then if Plk1 is a tumor promotor and Plk3 is a tumor suppressor? What is the NRDC binding affinity to each.

Reviewer #3 - Plks, mass-spec -(Remarks to the Author):

The manuscript was from the labs of Drs. Paul Chiao, Peter Stambrook and El Mustapha Bahassi, important contributors of the polo kinase field for over many years. The authors clearly deserved to be congratulated as this is another nice and important piece of contribution to the field.

The authors aimed to dissect the regulation of Plk3 in pancreatic cancer in great detail. Indeed, the regulation of Plk3 remained in the dark since its discovery and complete cloning in 2000 (Oncogene. 2000 Oct 5;19(42):4832-9. doi: 10.1038/sj.onc.1203845). At first the authors analyzed the expression pattern of Plk3 in clinical samples and in the transgenic mouse models P48-Cre; KrasLSL-G12D; PLK3+/+ vs. Cre; KrasLSL-G12D; PLK3-/- . Upon overexpression of 72-kDa Plk3 associated with the loss of anchorage-dependent growth (high percentage of floating cells) a subfragment of Plk3 (p41Plk3) encompassing the kinase domain became visible. Plk3-induced anoikis correlated with increasing levels of p41Plk3. Most interestingly, the authors could identify via functional screening of an ORFs library the metalloendopeptidase nardilysin (NRDC) that is responsible for the cleavage of Plk3 into two subfragments. It is also of broad relevance that this mechanism plays a similar role for Plk1 cleavage. Furthermore, the regulation of NRDC by PI3-kinase was investigated in detail. C-Fos could be identified as novel substrate for Plk3 in its p41Plk3 form.

In summary, the data are of very high relevance for the development of pancreatic cancer and for the entire field of polo kinases. In addition, for the use of small molecule Plk inhibitors in clinical trials in relation to their specificity the new data by Fu et al. need to be considered.

A few aspects should be addressed before publication

1. Is there a correlation between mRNA levels of Plk3 and the corresponding protein in human pancreatic cancer (Fig. 1, Suppl. Fig. 1)? The authors should clearly state in which type of cancer cells (primary tissue, PDX cells etc.) this correlation was seen and what technique and which antibody was used. This aspect is important because there is a long-lasting debate about the quality of Plk3 antibodies.
2. Indeed, a role for Plk3 in cell cycle regulation has been discussed multiple times but not for the G2/M transition or the duration of mitosis. Thus, it is surprising that an overexpression of p41Plk3 induces a high percentage of cells (over 80%) that are stuck in mitosis. I think it would be beyond the scope of the

manuscript to investigate the mechanism that is responsible for this mitotic arrest, but at least potential mechanisms should be discussed.

3. The major topic of this manuscript is the regulation of cell death by Plk3. The exact cell death pathways and mechanisms induced by Plk3 are still a matter for future research. In a previous study (Cell Res. 2016 Aug;26(8):914-34. doi: 10.1038/cr.2016.78) the role of Plk3 for the regulation of the extrinsic cell death was explored. Would it be possible that p41Plk3 phosphorylates caspase-8 or other cell death transducing proteins in pancreatic cancer cells? Has this been explored?

Point-by-point Response

Dear Reviewers,

Your review of our manuscript offered valuable comments and critiques. We thank you for your thoughtful suggestions and constructive criticisms. We did the experiments to address the critiques and revised the manuscript according your comments and suggestions. We presented the detailed discussions of revised experimental designs, results, and answers to all your questions. The key advances in the revised manuscript are briefly summarized as follows:

- The *in cis* and *in trans* regulatory mechanisms for Plk1 and Plk3 activation revealed that both p41Plk3 and p38Plk1 are generated from the scission of catalytic inactive p72Plk3 and p68Plk1 kinases, respectively.
- Free from C-terminal Plk3 inhibitory effect, p41Plk3 elevates its kinase activity, phosphorylates c-Fos at Thr164, which in turn, increases the transcription of Plk3 and the pro-apoptotic genes and orchestrating cFos-Plk3 feed forward pathway, for regulating G2/M transition in cell cycle, inducing anoikis, and suppressing tumorigenesis and metastasis.
- As kinase domain is in a close proximity to PBD, activated p41Plk3 can be negatively regulated by the stabilized NRDC-cleaved Plk3 C-terminus *in trans* possibly through protein folding.
- Our findings clearly demonstrate that the PI3K-mediated activation of both Plk3 and Plk1 by NRDC proteolytic cleavage generates p41Plk3, which has a pro-apoptotic activity, and p38Plk1, which has pro-survival function, to maintain balance between the opposing effects of Plk3 and Plk1 kinases.
- Our results also revealed that *p48-cre;Kras^{LSLG12D};Plk3^{-/-}* mice and “Dox/on” p72Plk3 mice developed metastatic PDAC, whereas ip72Plk3/NRDC or p41Plk3 expression impaired tumor growth, indicating that endopeptidase activity of NRDC is required for Plk3 activation and suppression of PDAC.
- Particularly striking is that cleavage by NRDC may play a key regulatory role in many signaling pathways. Analysis of 20,417 protein sequences in the UniProt SwissProt subset, we find that 5,833 proteins have no -X-Arginine-Lysine- NRDC cleavage sites (28.5%), 10,947 of them have 1-3 NRDC sites (53.6%), and 3637 (17.8%) of them have greater than 3 NRDC sites, suggesting that the -RK- dibasic motif does not occur randomly ($p < 0.0001$).
- There is a great potential that NRDC serves as an on/off signaling switch by cleaving its substrates, which contain structurally flexible binding motif and accessible cleavage sites close by, to activate or inactivate a signaling molecule. This discovery has potential far-reaching implications as a novel and important mechanism that orchestrates various biological signaling pathways through NRDC proteolytic activity. The regulatory potential of NRDC and its activation/inactivation of substrates by cleavage will be the focus of further exploration.

Thank you very much.

Reviewer #1 Anoikis metastasis - (Remarks to the Author):

“In this manuscript, Fu et al. describe a role for decreased Plk3 expression in the formation and progression of PDAC as well as elucidate the mechanism by which Plk3 is cleaved and activated. The authors find that Plk3 expression is decreased in human PDAC compared with normal pancreatic tissues, and that genetic knockout of PDAC increases PDAC development in mice. Furthermore, Plk3 is cleaved in its DR1 site by nardilysin (NRDC) activity, which is subsequently results in an elevation in Plk3 kinase activity. Specifically, a 41 kDa form of Plk3 that contains the kinase active site in its n-terminal domain, known as p41Plk3, is sufficient to increase apoptotic death by phosphorylating c-Fos on T164 to direct it to the nucleus where it increases transcription for pro-apoptotic genes. Overexpression of this p41Plk3 subunit is also sufficient to decrease PDAC tumor burden and invasion in mice.

While the study is of interest, there are multiple concerns, both major and minor, that should be addressed prior to publication”.

Major concerns

1. “In general, some conclusions made by the authors are overstated or otherwise not well supported by the data. For example, the figure caption of Fig. 2 states that “proteolytic processing of p72Plk3 to p41Plk3 is essential for induction of anoikis”. However, the data as presented are merely correlative and indicate that p41Plk3 levels are associated with anoikis induction. Causality is never demonstrated.”

We thank the reviewer for raising these questions. In response to causality question, we have shown that the expression of Plk3 in HEK293T cells led to cell round-up, detaching from plate, floating and apoptosis (anoikis) (Supplementary Fig. 2a and 2b). The floating cells exhibited higher level of cleaved PARP than attached cells. While the vector-transfected 293 cells grown under attached and suspension condition do not exhibit cell death (Supplementary Fig. 2a and 2b). Further investigation of the mechanism through which Plk3 induces apoptosis led to the discovery of p41Plk3 with the level of p41Plk3 being higher in floating cells than in attached cells. (Supplementary Fig. 2b and 2i). These data suggest that the occurrence of p41Plk3 causes cell detachment-induced apoptosis, anoikis.

In the revised manuscript, to validate the essential role of p41Plk3 in inducing anoikis, we determined the effect of Plk3 deletion on PDAC cell anoikis (Fig. 2l, m). The result demonstrated that knockout of Plk3 expression in several PDAC cells led to decreased expression of p41Plk3 and subsequent reduced level of cleaved PARP and apoptosis-inducing activity (Fig. 2l, m).

Fig. 2 Proteolytic processing of p72Plk3 to generate p41Plk3 is important for induction of anoikis.

I, m Immunoblot of p72Plk3, p41Plk3, and cleaved PARP expression (I) and flow cytometry analysis of apoptosis-inducing activity (m) of indicated PDAC cells that were lentivirally transduced to express the sgRNA targeting Plk3 or non-targeting control sgRNA.

Regarding the association of p41Plk3 levels with anoikis induction, as shown in our original submission, the increased p41Plk3 expression generated from p72Plk3 cleavage in non-tumorigenic HPDE pancreatic epithelial cells coincided with an increase in the level of apoptosis of cells in suspension culture (Fig. 2b and 2i), suggesting increased p41Plk3 expression is associated with anoikis induction.

Direct tests on different PDA lines with different levels of endogenous PLK3, p41Plk3 induction and apoptosis/anoikis were also conducted in supplementary Fig. 10e and 10f. Here, we grow multiple PDAC cells in attached and suspension culture condition to induce anoikis with HPDE included as a control, finding that expression of p41Plk3 was increased in the detached PATC50, 66 and 108 cells compared with attached cells, but remain same in Panc-28, MIA-PaCa-2 and PATC43 cells. (supplementary Fig. 10e). It is worth noting that the response to apoptosis in these cells coincidence with the observed alterations of Plk3 expression. Panc-28, MiaPaCa-2, PATC43 cells had a similar response and resistant to anoikis-induced apoptosis. However, PATC50, 66 and 108 cells responded to detached growth condition and anoikis was induced (supplementary Fig. 10f). Notably, HPDE cells harboring most pronounced p41Plk3 expression under suspension culture exhibited highest apoptosis induction (~50%) compared with PDAC cells ($\leq 40\%$). These data suggest that the upregulation of p41Plk3 in detached PDAC cells are critical in driving cells towards apoptosis. More susceptible to anoikis in HPDE cell largely depends on its sensitivity to cleavage enabling higher p41Plk3 generation. Therefore, these data also support that induction of p41Plk3 correlates with the level of apoptosis.

Supplementary Figure 10. NRDC proteolytic cleavage generates p41Plk3 with pro-apoptotic activity and p38Plk1 with pro-survival function, respectively.
(e, f) Immunoblot of p41Plk3 and p68Plk1 **(e)**, and flow cytometry analysis of apoptosis-inducing activity **(f)** in HPDE and a panel of PDAC cells grown on adhesive (A) or polyHEMA-coated (S) plates.

Considering PDA tumor cells were more resistant to cleavage and anoikis induced by PLK3 expression, in the revision study, we overexpressed Plk3 WT and Plk3R354G containing mutated NRDC cleavage site in several PDAC cells under suspension conditions for detection of anoikis (Fig. 3g, h). The results demonstrated that the generation of p41Plk3 by NRDC cleavage was clearly inhibited to a greater degree in cells transfected with Plk3R354G than in cells transfected with Plk3WT. Moreover, NRDC cleavage site mutation in Plk3 triggered less anoikis as indicated by both reduced PARP cleavage and apoptosis. This suggests that p41Plk3 generation promotes anoikis and its level is associated with anoikis induction level.

2. “The conclusions for the role of Plk3 in anoikis induction and the mechanism by which Plk3 is proteolytically processed rely heavily on data generated using non-pancreatic cell lines. This weakens the main connection that the authors attempt to make between pancreatic tissues and endogenous Plk3.”

As discussed in question 1, p72Plk3/p41Plk3-induced apoptosis/anoikis experiments were done in non-tumorigenic HPDE pancreatic epithelial cells in Fig. 2b and 2i, showing induction of p41Plk3 expression led to increased anoikis of cells in suspension culture. The results from supplementary Fig. 10e and 10f with HPDE and multiple PDAC cells grown in attached and suspension culture condition also support that induction of p41Plk3 in PDAC cells with different levels of endogenous Plk3 correlates with the level of apoptosis. In the revision study, we demonstrated that NRDC cleavage site mutation in Plk3 (Plk3R354G) significantly reduced the level of cleaved PARP and apoptosis (Fig. 3g, h). This further confirmed that NRDC-dependent cleavage of Plk3 induces anoikis of PDAC.

3. “Similarly, many of the IP experiments utilize exogenous protein without any attempt to assess protein-protein interactions using endogenous proteins. This is not to say that there is no role for the IP data with exogenously expressed protein. Rather, the conclusions are overly reliant on these approaches.”

We thank the reviewer for pointing this out. In the revised manuscript, we performed several IP experiments in PDAC cells using endogenous proteins.

4. “Appropriate controls are frequently lacking to confirm overexpression, activity of various inhibitors, and immunoprecipitation. For example, the expression of p110 α in Fig. 4i is not confirmed by an immunoblot, and there is no positive control for LY294002 inhibitor activity in Fig. 4. IP experiments often do not include controls for non-specific binding.”

Thanks for the reviewer to point this out. In the revised study, we confirmed p110 α expression in new Fig. 4g, added positive controls for LY294002 inhibitor activity in Fig. 4e and 4f, included IgG controls in IP experiments. All experiments are performed in PDAC cells.

Fig. 4 Phosphoinositide 3-kinase regulates NRDC activity for Plk3 and Plk1 activation via centaurin- α 1.
e, f Immunoblot of p72Plk3, p41Plk3, and cleaved PARP expression in PATC148/ip72Plk3 cells (**e**) and p72Plk3 transfected PANC-1 cells (**f**) treated with PI3K inhibitor LY294002 (20 μ M) for indicated times. **g** p72Plk3 and p41Plk3 expression in PANC-1 cells co-transfected with Plk3 and a vector control or the PI3K catalytic unit p110 α . **j** IP and IB detection of increased binding of NRDC with α -centaurin and decreased binding of Plk3 with NRDC in PANC-1 cells transfected with Flag-p72Plk3 and treated with or without LY294002 (32 μ M).

5. “In Figure 1b, some of the cell lines labeled on the immunoblot are not mentioned or discussed in the manuscript. This creates difficulty in assessing the conclusions regarding Plk3 levels and the level of aggressiveness of each PDX model.”

Plk3 expression in star labelled PDAC PDX cells in Fig. 1b was firstly discussed on page 4 in the manuscript “Strikingly, MDA-PATC148, 148LM, 148LM2, 153, 153LM, and 219B cells, which were derived from liver metastasis, exhibited significantly reduced Plk3 expression compared to non-metastatic MDA-PATC102, 108, and 124 cells (Fig. 1b), indicating that Plk3 downregulation is associated with PDAC PDX metastasis”. Figure legend also explained that “* indicates cells derived from liver metastases of primary PDAC”. In Fig 1b, we group these PDX PDAC cell lines into metastatic and non-metastatic cells. We used metastatic PDX cell line PATC148 and PATC153 in cell models and orthotopic mouse models to demonstrate that activated p41Plk3 induced anoikis and suppressed PDAC tumorigenesis and metastasis (Fig. 7, Supplementary Fig. 9 and Supplementary Fig. 10). In the revised study, we pointed out PATC148 and PATC153 as metastatic PDX cells in the text to strengthen the role of p41Plk3 in inducing anoikis and suppressing PDAC metastasis.

We obtained these 16 PDAC PDX cells from our collaborator Dr. Jason B. Fleming, four of them was characterized in the paper (Kang et al. Lab Invest. 2015 Feb; 95(2): 207–222.). We feel it is challenging to conclude Plk3 expression levels and the level of aggressiveness of each SINGLE PDX model based on the western blot in Fig. 1b.

6. “CBS loading control band patterns appear substantially different between immunoblots using the same cell lines. For example, compare CBS banding in Fig. 2d & 2e which are both from HPDE cells.”

Fig. 2d and 2e experiments were conducted in different gels. It should be reasonable to show CBS loading control bands at different molecular weight position for each experiment. This may cause band pattern different between blots. To more rigorously present the data, we repeated the experiment using vinculin or b-actin as loading controls in the revised manuscript (Fig. 2d, e).

Fig. 2 Proteolytic processing of p72Plk3 to generate p41Plk3 is important for induction of anoikis.
d, e Immunoblots of cleaved PARP in HPDE cells with stable shRNA-mediated knockdown of Plk3 (**d**) and stably reconstituted with Flag-Plk3 (**e**).

7. “The FACS quantifications of apoptotic cells in Fig. 3f-g do not seem to match with the results of the colony formation assay with the same cell lines in Fig. 3d. If there are changes in cell death that are only 10-20%, why is it that colony growth is changing by (in some cases) 90%?”

We thank the reviewer for pointing this out. We use two different experiment methods, flow cytometry and colony formation, to measure cell death. The results largely depend on cell starting concentration/seeding density and culture time, which will result in absolute number difference in two different experiments. Importantly, both experiments consistently suggest that NRDC indeed cleaved Plk3 at residue Arg354, p41Plk3 induced high apoptosis comparable to p72Plk3 in 293T cells. The role of activated p41Plk3 in the induction of anoikis was also verified throughout the manuscript using multiple PDAC cells lines and orthotopic mouse models. In the revised study, we also determined the apoptosis-inducing ability of Plk3 WT and mutants using colony formation assay and flow cytometry analysis in PANC-1 cells. As shown in Fig. 3c-e, Plk3 especially p41Plk3 promotes anoikis, whereas those with Plk3 DR1 deletion and substitution were anoikis resistant. This suggested that p41Plk3 plays a critical role in inducing anoikis.

Fig. 3 Identification of the proteolytic cleavage site in the DR1 domain of Plk3.
c-e Colony-formation assay (**c, d**) and flow cytometry analysis of apoptosis-inducing activity (**e**) from PANC-1 cells transfected with indicated Flag-tagged Plk3 or mutants.

8. “In Figure 4b, the delta 326-377 protein expression (seen in the input) is substantially lower than the WT or R354G. Thus, the conclusion that the delta 326-377 protein binding to NRDC is compromised is not well supported. It could be that the differences in total protein account for the IP results.”

We thank the reviewer to point this out. We repeated the IP experiment in PANC-1 cells with even input loading. As shown in figure 4a, deletion of aa326-377 in Plk3 abolished its interaction with NRDC, suggesting that aa326-377 are important for binding to NRDC.

Fig. 4 Phosphoinositide 3-kinase regulates NRDC activity for Plk3 and Plk1 activation via centaurin-α1.
a Immunoprecipitation and IB detection of NRDC-Plk3 interactions in PANC-1 cells transfected with Flag-tagged Plk3WT, R354G, or Δ326-377 mutants.

9. “The organization of figure panels throughout the manuscript is muddled and creates difficulty in assessing the findings. Similarly, the figures are not always referenced in the text in order which also creates challenges for readers/reviewers.”

We have more data included in the revised manuscript. We paid a very close attention to figure organization and presentation.

10. “Explanation of intent, description of findings, and interpretation of results are often unclear. It should be easily understood why each experiment was performed, what was found, and how the data support or contradict the hypothesis.”

In the revised study, we paid more attention to the study design, interpretation of experiment results and other comments as the reviewer suggested.

Minor concerns

1. “Frequently used abbreviations (like PanIN, ORF, PBD) are often not described throughout the paper.”

In the revised study, we added pancreatic intraepithelial neoplasia for PanIN, and open reading frame for ORF. Polo-box domain for PBD and described them in the text.

2. “The introduction section could contain more information about nardilysin, as it becomes a focal point of the data after not being introduced early in the text.”

We thank the reviewer for the comment. We do not want to break the flow of our introduction section by inserting NRDC information and there is no developed connection to NRDC early in the text yet. Instead, we introduced more NRDC background in the Results part when we first demonstrated that Plk3 is proteolytically processed by NRDC.

3. “Throughout the paper, Plk1 data are intermingled in figures without much context or justification, especially when the main conclusions relate to Plk3.”

Five Plks (Plk1-5) have been identified with a conserved protein kinase domain. The role of Plk1 in regulating cell cycle progression and promoting tumor development has been well documented. In this manuscript, our focus is cleavage-mediated Plk3 activation mechanism and p41Plk3 function in suppressing PDAC tumor growth and metastasis. This mechanism plays a similar role for Plk1. Plk1 undergoes the same scission by NRDC proteolytic cleavage to generate p38Plk1, which plays an oncogenic role in PDAC development. We also studied inter-regulation between Plk1 and Plk3 in pancreatic cancer to examine the concurrent opposite effects of Plk3 and Plk1 in regulation of apoptosis. We put the mechanism and functional data of Plk1 side by side with Plk3 to support the discovery of a previously unidentified cleaved form of Plk3 and to suggest the strategy of designing small molecule Plk inhibitors as well. This is further justified in the revised manuscript.

4. “Figure 1e & 1f are redundant ways to present identical data.”

We removed the bar graph and keep the table to describe the mice phenotype and statistical analysis in the revised manuscript.

5. “The cleavage site in Figure 5a is difficult to spot at first glance; the scissors blend in too much with the ribbon model.”

We modified Figure 5a and made the cleavage site and scissor visible in the revised manuscript (Fig. 5a).

Reviewer #2 - PDAC, metastasis, mouse models - (Remarks to the Author):

Title: Nardilysin-Regulated Scission Mechanism Activates Polo-like Kinase 3 to Suppress the Development of Pancreatic Cancer

Manuscript Number # NCOMMS-22-33695

“Reviewer’s Comments: The focus of this research study indicates the role of Plk3 and its cleaved form of p41KDa mediated anoikis. When p72Plk3 is cleaved by NRDC and p32 Plk3 is removed by degradation, the N-terminal p41Plk3 gains kinase activity. p41Plk3 biological functions include induction of apoptosis and exhibit tumor suppressor role. The findings also demonstrate that PI3K regulates post-translational processing of Plk1 and Plk3 involving PI3K/centaurin-1/NRDC signaling axis to generate functional p38Plk1 and p41Plk3. There are some major concerns about the data presented:”

1. “The focus is on pancreatic cancer, but many results were obtained from 293T human embryonic kidney cell line (Figure 3, 4, 5, 6 and Supplementary Figure 2, 4) and human colorectal carcinoma cell line HCT116 (Supplementary Figure 2, 4).”

We thank the reviewer for the comments. The initial study of Plk3 induced anoikis, NRDC-mediated cleavage, identification of cleavage site was carried out using 293T cells, which led to the discovery of p41Plk3. As we did find Plk3 in 293T cells has highest basal level of p41Plk3 generation. These major findings were verified in non-tumorigenic HPDE pancreatic epithelial cells and multiple PDAC cells. Particularly, the role of p41Plk3 induced anoikis was extensively studied in PATC148 and PATC153 cells and their orthotopic mouse models (Fig. 2a-2e, 2i, 3i-3k, 4b-4f, 4g, 4h, 6b-6e, 6k-6n, 7a-7f; supplementary Fig 2g, 6a-6b, 6d-6i, 7d-7f, 8a-8j, 9a-9l, 10a-10h). In the revised study, we performed all experiments in Fig. 3, 4, 5, 6 and Supplementary Fig. 2, 4 using PDAC cells, removed HCT116 results with PDAC cells (Fig. 3f, Supplementary Fig. 2c) and re-organize figures to strengthen our finding that NRDC-mediated cleavage of Plk3 induces anoikis and suppresses pancreatic cancer development.

2. “Figure 1b-1d: what are the levels of p41Plk3 in the metastatic cell lines? Supplementary figure 1c: whether β -actin is analyzed on the same membrane? What are the levels for p41Plk3?”

In the revised study, we grow multiple PDAC cells in suspension culture condition. As shown in Fig. 2h, metastatic PDX cells exhibited lower level of p72Plk3 expression and undetectable p41Plk3 compared to HPDE and non-metastatic PDX cells, indicating that Plk3 in metastatic PDX cells are more resistant to cleavage and anoikis induction. Downregulation of p72Plk3 and p41Plk3 is associated with PDAC PDX metastasis. We used antibody specific for N-termini of Plk3 (anti-human) in this experiment.

For Fig. 1d, we have attempted to examine p41Plk3 expression in PDAC cells isolated from *p48-cre;Kras^{LSL-G12D};INK4a/Arf^{F/F}* mice (KIC) and *Kras^{LSL-G12D}; p53^{LSL-H172R}* mice (KPC) and in normal mouse pancreatic epithelial cells (MPECs) using anti-mouse, N-terminal specific Plk3 antibody. However, due to the poor antibody quality, the blot showed many non-specific bands and did not give any informative result. This is not presented in the manuscript.

Supplementary Fig 1c, β -actin is analyzed on the same membrane.

3. “In figure 1b: among the cancer cell lines, 102, 108, and 124 also express a similar level of p72Plk3 as normal HPDE, while the expression of p72Plk3 appears to be relatively low in metastatic cell lines. So, whether loss of Plk3 is involved in metastasis specifically?”

Fig. 1b showed that MDA-PATC148, 148LM, 148LM2, 153, 153LM, and 219B cells, which were derived from liver metastasis, exhibited significantly reduced Plk3 expression compared to non-metastatic MDA-PATC102, 108, and 124 cells, indicating that Plk3 downregulation is associated with PDAC PDX metastasis (page 4)". As addressed in the question 1 above, our new Fig. 2h showed metastatic PDX cells exhibited lower level of p72Plk3 expression and undetectable p41Plk3 compared to HPDE and non-metastatic PDX cells, supporting the notion that downregulation of p72Plk3 and p41Plk3 is associated with PDAC PDX metastasis. This is also consistent with our genetic evidence that deleting *Plk3* in *p48-cre;Kras^{LSL-G12D}* mice dramatically accelerated *Kras^{G12D}*-driven PanIN and subsequent PDAC formation. Metastases in the liver and lung were observed in 7 of the 11 (64%) mice that developed PDAC (Fig. 1f-1h, supplementary Fig. 1d and 1e). These results suggest that loss of Plk3 is involved in PDAC tumor progression and metastasis.

We have demonstrated in the manuscript that p41Plk3 expression promotes cell detachment-induced apoptosis, a typical feature of anoikis. Anoikis is a programmed cell death occurring upon cell detachment from the correct extracellular matrix, thus disrupting integrin ligation. It is a critical mechanism in preventing dysplastic cell growth or attachment to an inappropriate matrix. Anoikis prevents detached epithelial cells from colonizing elsewhere and is thus essential for tissue homeostasis and development. Cancer cells acquire anoikis resistance to survive after detachment from the primary sites and travel through the circulatory systems to spread throughout the body. We stressed on anoikis and aim at the molecular mechanisms governing p41Plk3-induced anoikis and focusing on their regulation in metastatic PDAC. Plk3 in PDAC cells are more resistant to NRDC-mediated cleavage to generate activated p41Plk3, bestowing PDAC cells to develop anoikis resistance, to progress towards malignancy and metastasis.

4. "What is the status of p41Plk3 in the KPC mice in Figure 1d? Figure 1h: is there an IHC staining done to show the expression of p41Plk3 in *p48-cre;Kras^{LSL-G12D}Plk3+/+*."

Please refer to our answers to Q3 above for p41Plk3 level in Fig 1d.

It is a great idea to identify p41Plk3 by IHC in tissues, as it will provide direct evidence that Plk3 is in fact cleaved in human or mouse tissues. However, this assay is obstructed by the availability of Plk3 antibody that can specifically recognize p41Plk3 in IHC considering both p41Plk3 and p72Plk3 are recognized by N-terminus Plk3 antibodies. The current available Plk3 commercial antibodies would not distinguish p41Plk3 from p72Plk3 in IHC. Instead, we used Patient-derived xenograft (PDX) models throughout our research, which were obtained from our collaborator Dr. Jason Fleming at the Department of Surgical Oncology at MD Anderson Cancer Center. PDX cells derive directly from pancreatic cancer patient tumors tissues. That reflects original tumor characteristics such as heterogeneous histology, clinical biomolecular signature, malignant phenotypes and genotypes, tumor architecture, and tumor vasculature, making them a valuable foundation of the oncology research. We grow multiple PDX cells under suspension culture and revealed expression of p41Plk3 under physiological conditions, validating the cleavage of Plk3 is the mechanism for generating p41Plk3 (Fig. 2h, 4j and supplementary Fig 9e).

It would be of great interest to generate C-terminus specific antibodies to detect C-terminal carboxyl group of cleaved p41Plk3 in the future study. But we also realized it takes too long time to make such antibodies for manuscript revision. As monoclonal antibodies against the unique charged C-terminal carboxyl group are required, the screening process will be very time consuming. Furthermore, it is unclear whether posttranslational modification of C-terminus (C-methyl-esterification and α -amidation) will hinder the binding of C-terminus specific antibodies and complicates interpretation of the expected results. For examples: C-terminal α -amidation neutralizes the negative charge of the carboxyl group at the C-terminus. C-methyl-esterification is the most frequently annotated C-terminal modification as reported in TopFIN.

5. "In figure 1, most of the pancreatic cell lines still express a detectable p72Plk3 level. How do you justify this? Whether it determines the possibility of Plk3 independent tumor mechanism in PDAC or if there is any possible mechanism involved in inhibiting the function of p72Plk3. What's the expression pattern of NRDC in pancreatic cancer?"

These are excellent questions. Western blot in Fig. 1b and 1c showed that Plk3 protein levels were reduced in most human PDA PDX cells, particularly in PDXs derived from liver metastases, and were also decreased in a subset of human PDA cell lines compared to normal HPDE cells. The mechanism investigations showed that NRDC/PI3K-regulated cleavage of Plk3 generates a kinase-active p41Plk3 fragment that can induce anoikis in pancreatic cancer cells, reduce growth and metastasis in mice. We found that PDA tumor cells, especially metastatic PDX PDAC cells were more resistant to cleavage and anoikis induced by PLK3 expression. To demonstrate whether the cleavage of Plk3 is associated with NRDC or PIP3 expression levels, we performed western blot analysis using a panel of PDAC cell lines under suspension culture (Fig. 4h). The results showed that the cleavage of p72Plk3 positively correlates with expression level of NRDC. Detached HPDE cells exhibited high level of p110 α compared with PDAC cells. For reviewer information, our bioinformatics analysis using ICGC datasets showed the expression of Plk3 and NRDC is associated ($r=0.311$, $p=0.002$) (see below).

Figure 4. (h) Immunoblots of p41Plk3, NRDC and p110 α expression in a panel of PDX cell lines in comparison to HPDE cells. Cells were grown in suspension culture for 36 h.

Expression correlation between Plk3 and NRDC in PDAC samples

Linking a study from Ikuta *et al* that [1] pancreatic deletion of NRDC in *Kras*^{G12D}-driven mice promoted PDAC tumorigenesis and NRDC expression is absent in a subset of human PDAC, our studies suggest a fundamental mechanism that, in addition to reduced expression level, deficient NRDC-mediated cleavage on Plk3 to generate activated p41Plk3 might bestow PDAC cells to develop anoikis resistance, to progress towards malignancy and metastasis.

6. “In figure 2b – cleaved PARP induction is visibly seen after 2 hrs in suspension culture. Whereas figure 2i, 2j, 2k -HPDE/HEK293T/MEF cell lines reveal the cleavage to 41kd from 72 takes at least 3 hrs. Whether the induction of cleaved PARP is independent of 41kd?”

We thank the review for the comments. However, we would not make this conclusion based on band intensity as these western blots are performed using different cell lines, and they target different proteins (cleaved PARP vs p41Plk3). We have shown that the expression of Plk3 in HEK293T cells led to cell round-up, detaching from plate, floating and apoptosis (anoikis) (Supplementary Fig. 2a and 2b). The floating cells exhibited higher level of cleaved PARP than attached cells. While the vector-transfected 293 cells grown under attached and suspension condition do not exhibit cell death (Supplementary Fig. 2a and 2b). Further investigation of the mechanism through which PLK3 induces apoptosis led to the discovery p41Plk3 with the level of p41PLK3 being higher in floating cells undergoing apoptosis (Supplementary Fig. 2b and 2i), these data suggest that the occurrence of p41Plk3 causes cell detachment-induced apoptosis, anoikis.

In the revised manuscript, to validate the essential role of p41Plk3 in inducing anoikis, we determined the effect of Plk3 deletion on PDAC cell anoikis (Fig. 2l, m). The result demonstrated that knockout of Plk3 expression in several PDAC cells led to decreased expression of p41Plk3 and subsequent reduced level of cleaved PARP and apoptosis-inducing activity (Fig. 2l, m).

Fig. 2 Proteolytic processing of p72Plk3 to generate p41Plk3 is important for induction of anoikis.

I, m Immunoblot of p72Plk3, p41Plk3, and cleaved PARP expression (**I**) and flow cytometry analysis of apoptosis-inducing activity (**m**) of indicated PDAC cells that were lentivirally transduced to express the sgRNA targeting Plk3 or non-targeting control sgRNA.

Regarding the association of p41Plk3 levels with anoikis induction, as shown in our original submission, the increased p41Plk3 expression generated from p72Plk3 cleavage in non-tumorigenic HPDE pancreatic epithelial cells coincided with an increase in the level of apoptosis of cells in suspension culture (Fig. 2b and 2i), suggesting increased p41Plk3 expression is associated with anoikis induction.

Fig. 2 Proteolytic processing of p72Plk3 to generate p41Plk3 is important for induction of anoikis.

b Immunoblots of cleaved PARP in HPDE cells grown in suspension (Sus. Culture) on polyHEMA-pre-coated plates at the indicated times. **I** Immunoblots of p72Plk3 and p41Plk3 in HPDE cells grown in suspension for the indicated times.

Additionally, we overexpressed Plk3 WT and Plk3R354G containing mutated NRDC cleavage site in several PDAC cells under suspension conditions for detection of anoikis (Fig. 3g, h) in the revised study. The results demonstrated that the generation of p41Plk3 by NRDC cleavage was clearly inhibited to a greater degree in cells transfected with Plk3R354G than in cells transfected with Plk3WT. Moreover, NRDC cleavage site mutation in Plk3 triggered less anoikis as indicated by both reduced PARP cleavage and apoptosis. This suggests that p41Plk3 generation promotes anoikis and its level is associated with anoikis induction level (cleaved PARP level).

Fig. 3 Identification of the proteolytic cleavage site in the DR1 domain of Plk3.

g, h Immunoblots of p72Plk3 and p41Plk3 expression (**g**) and flow cytometry analysis of apoptosis-inducing activity (**h**) from

PDAC cells transfected with indicated Flag-tagged Plk3 WT or mutants and grown under suspension condition for 36 h.

7. "The mouse developed a tumor in p48- cre; KrasLSL-G12DPlk3+/+ group, did it show any correlation with the genes mentioned in supplementary Table 1? Did authors analyze those samples because they should match with the genes mentioned in the surviving colonies that are resistant to Plk3 overexpression?"

We thank reviewer for raising these interesting questions. ORFs isolated from the 6 out of 60 surviving colonies in supplementary Table 1 share common features, encoding metalloendopeptidase nardilysin (NRDC) cleavage sequences -X-Arg-Lys- motif. Functionally, they might interfere with nardilysin-mediated cleavage of Plk3. If so, these peptides are great research tools, as they might promote pancreatic cancer development. However, we feel it would be beyond the scope of this manuscript revision to study these 6 genes, especially this manuscript is centered on our novel finding that Plk3 is activated by PI3K/NRDC-regulated cleavage and p41Plk3 promote anoikis, G2/M arrest, and PDAC and metastasis suppression.

8. "What causes the Plk3 downregulation? Is there any known mutation? And what percentage of PDACs in which Plk3 is downregulated? If Plk1 seems to be an oncogene, is there an interplay between the two family members?"

These are very interesting questions. Please refer to question 5. Our studies suggest a fundamental mechanism that, in addition to reduced Plk3 expression level, deficient NRDC-mediated cleavage on Plk3 may cause reduced induction of activated p41Plk3, leading to PDAC cells anoikis resistance. In regard to mutation, to our knowledge, there is no evidence for tumor-associated inactivating mutations in Plk3 or NRDC in pancreas or other cancers reported in the literatures or database.

Additionally, Kras^{G12D} is the earliest mutation detected in pancreatic intraepithelial neoplasia (PanIN) lesions and found in about 90% of human PDAC. We have planned to study whether Kras mutation affects Plk3 expression and cleavage in the revised study. We generated construct with Doxycycline-inducible Kras expression in HPNE cells and analyzed whether Kras regulates Plk3 expression; whether cFos is involved in Kras-Plk3 pathway. For reviewer information, Panel A showed that doxycycline induced expression of Kras stimulates an increased expression of Plk3 in first 24h, followed by a decreased expression over the next 48 and 72h. c-Fos showed delayed expression reduction in response to Kras induction. We also analyzed c-Fos mRNA levels in multiple PDAC cells carrying Kras mutation by qRT-PCR and the results showed c-Fos expression was significantly lower in 7 out of 12 of these PDAC cells than in HPDE cells (Panel B). This indicates Kras might regulate Plk3 expression and suppression of PLK3 expression in these cells is a result of reduced c-Fos levels. In addition, we have previously shown that Plk3 expression was regulated by RelA/NF- κ B [8]. Reduced RelA/NF- κ B and c-Fos levels could result in the decreased Plk3 expression. This deserves future detailed study.

(A) qRT-PCR analysis of Kras, Plk3 and c-Fos expression under a Dox-inducible system with Kras stably transfected HPNE cells treated with Dox for the indicated times. (B) qRT-PCR analysis of c-Fos expression in a panel of PDX cell lines and HPDE cells.

Regarding the percentage of downregulated Plk3 in PDACs, we have performed immunohistochemistry staining for Plk3 expression in human PDACs using a tissue microarray containing 33 PDAC samples and 56 normal pancreatic tissue samples (Fig. 1a). The results showed that 45% (25/56) of the normal tissue samples exhibited high levels of Plk3 expression, 82% (27/33) of the PDACs had low expression of Plk3 ($p = 0.0114$).

It is interesting to study molecular interplay between Plk1 and Plk3 that regulate opposing effects in PDAC development. Our preliminary data suggested that p41Plk3 induced anoikis by increasing expression of pro-apoptotic genes. Conversely, highly expressed Plk1 in PDAC tumor promoted anti-apoptotic response [2-4]. Since both the activations of Plk1 and Plk3 are regulated by NRDC cleavage, we explored whether there is inter-regulation between Plk1 and Plk3 by stably expressing both Plk1 and Plk3 together in PATC148 cells. As shown in our original submission (supplementary Fig. 10g and 10h), The increased p41Plk3 expression may suppress the expression of p68Plk1, resulting in decreased levels of p38Plk1 compared to p68Plk1 over-expression alone (supplementary Fig. 10g). This consequently led to enhanced anoikis activity in co-transfected cells (supplementary Fig. 10g and 10h), implying a mutually exclusive inter-regulation of Plk1 and Plk3 proteolytic cleavage in controlling PDAC cell proliferation and apoptosis.

Supplementary Figure 10. NRDC proteolytic cleavage generates p41Plk3 with pro-apoptotic activity and p38Plk1 with pro-survival function, respectively.

(e, f) Immunoblot of p41Plk3 and p68Plk1 (e), and flow cytometry analysis of apoptosis-inducing activity (f) in HPDE and a panel of PDAC cells grown on adhesive (A) or polyHEMA-coated (S) plates.

9. “Co-IP of NRDC indicated that LY294002 increased the binding of NRDC to centaurin-1 in the cytoplasm and decreased its binding to p72Plk3 (Fig. 4i). But it was shown only in 293T, which is a human embryonic kidney cell line, and not in any other PDAC cell line. Though the experiments with LY294002 were performed in many different cell lines, the Western for centaurin-1 was shown only in 293T cells.”

We repeat this IP experiment using PANC-1 cell line in the revised study. As shown in the Fig. 4j, Co-IP of NRDC indicated that LY294002 increased the binding of NRDC to centaurin- α 1 in the cytoplasm and decreased its binding to p72Plk3.

Fig. 4 Phosphoinositide 3-kinase regulates NRDC activity for Plk3 and Plk1 activation via centaurin- α 1.

j IP and IB detection of increased binding of NRDC with α -centaurin and decreased binding of Plk3 with NRDC in PANC-1 cells transfected with Flag-p72Plk3 and treated with or without LY294002 (32 μ M).

0. “Figure 5i: What is the protein half-life of plk3NT/plk3CT in the metastasis cell lines? Whether different antibodies are used for detecting NT-1-354 and CT-356-646?”

In the revised study, we determined half-life of Plk3CT in metastatic PATC148 cells (Fig. 5i, j). Time course of cycloheximide (CHX) treatment in PATC148 cells expressing NT₁₋₃₅₃- and CT₃₅₄₋₆₄₆-Plk3 revealed that NT₁₋₃₅₃ was stable over the time, whereas CT₃₅₄₋₆₄₆ is subjected to degradation with half-life of 12-13 h (Fig. 5i, j).

Fig. 5 C-terminal p32Plk3 inhibits N-terminal p41Plk3 kinase activity and apoptosis-inducing activity. **i** N- and C-terminus Plk3 expression in NT¹⁻³⁵³- or CT³⁵⁴⁻⁶⁴⁶-Plk3 transfected PATC148 cells treated with the Cycloheximide (10 µg/ml) at the indicated time points. **j** Quantification of CHX assays to depict Plk3CT³⁵⁴⁻⁶⁴⁶ expression vs. timed CHX treatment. Dotted line indicates 50% of initial Plk3CT³⁵⁴⁻⁶⁴⁶ protein signal.

Plk3NT1-353 is tagged with Flag and detected in western blot using anti-Flag; Plk3CT354-646 is detected using C-terminal specific anti-Plk3 antibody.

1. “Supplementary Figure 7h - please mark the different cell cycle phases and include population distributions.”

We marked cell cycle phase in Supplementary Fig. 8h in the revised manuscript. Cell cycle distribution was studied using flow cytometry and results were shown in Supplementary Fig. 8a, revealing overexpression of p41Plk3 induced G2-M arrest, whereas p72Plk3 had no effect on cell cycle distribution.

2. “Authors have found that p41Plk3 expression is significantly increased in the PTEN-KO compared to PTEN-WT MEF cells. It is also suggested that PI3K is required to cleave p72Plk3 into its active form p41Plk3 (apoptosis inducer). - If p72Plk3 is a tumor suppressor and its active form p41Plk3 is anoikis or apoptosis inducer, then the concern is about PI3K because PI3K oncogenic signaling is known to be upregulated in PDAC. PTEN which is a known tumor suppressor KO increased p41Plk3 expression. Please justify how to correlate these events.”

We thank the reviewer for pointing this out. We also realized that PI3K oncogenic signaling is known to be upregulated and PTEN can downregulate PI3K in PDAC. LY294002 treatment indeed induced apoptosis as reflected by increased PARP cleavage, but also blocked Plk3 cleavage (Fig. 4e-g).

Firstly, considering the activated PI3K/NRDC in PDAC also regulates p68Plk1 cleavage to generate p38PLK1, which has anti-apoptotic activity (supplementary Fig. 10a-10d). The low induction and minimum change of p41Plk3 expression regulated by PI3K may not be sufficient to counteract p38Plk1 pro-proliferation effects in PDAC.

Secondly, considering oncogenic Kras activates PI3K and other signaling pathways driving pancreatic tumorigenesis, our results suggest Plk3 cleavage may not be required for the effects of PI3K in PDAC development. There could be other molecules involved in the regulation of Plk3 cleavage.

Thirdly, additional examples can be appointed out: It is clear that NF-κB is anti-apoptotic or pro-apoptotic bifunctional signaling factor, depending on upstream stimulation. c-Fos is generally thought to be oncogenic but could also have pro-apoptotic function as shown in our study. Our study began to suggest that phosphorylation on Thr-164 of c-Fos by Plk3 and upregulation of pro-apoptotic downstream target genes contributed the functional switching. As our results showed, PI3K may have a potential to be a bifunctional signaling regulator. However, the molecular mechanisms that control this functional switching is not known. Respectively, we would like to point out that exploration of the novel regulation mechanisms of PI3K is out of scope of our current study.

Fig. 4 Phosphoinositide 3-kinase regulates NRDC activity for Plk3 and Plk1 activation via centaurin- α 1.

e, f Immunoblot of p72Plk3, p41Plk3, and cleaved PARP expression in PATC148/ip72Plk3 cells (**e**) and p72Plk3 transfected PANC-1 cells (**f**) treated with PI3K inhibitor LY294002 (20 μ M) for indicated times. **g** p72Plk3 and p41Plk3 expression in PANC-1 cells co-transfected with Plk3 and a vector control or the PI3K catalytic unit p110 α . **j** IP and IB detection of increased binding of NRDC with α -centaurin and decreased binding of Plk3 with NRDC in PANC-1 cells transfected with Flag-p72Plk3 and treated with or without LY294002 (32 μ M).

10. “Plk1 cleavage is markedly increased in PTEN-KO but not in PTEN-WT MEFs. If Plk1 is a tumor promotor. It’s been known that PTEN is a negative regulator of PI3K and PI3K is a known downstream effector of K-Ras, which is a driver of PDAC. Then how the KO of PTEN increases the cleavage of Plk3 into the active pro-apoptotic p41Plk3 kDa form? And vice versa how the inhibition of PI3K by LY294002 prevents the cleavage of p41plk3, which can induce apoptosis?”

Please refer to the question 12 above.

11. “Then if Plk1 is a tumor promotor and Plk3 is a tumor suppressor? What is the NRDC binding affinity to each.”

Our results suggested that NRDC interacts with Plk1 and Plk3 and generated NRDC-cleaved p38Plk1 and p41Plk3 kinases. In vitro experiments mixing purified Plk with NRDC and determining which Plks will bind to NRDC more is one of the approaches to understand which Plk to be cleaved by NRDC. However, using purified proteins in large excess (for visualization purpose) is not physiological condition and the potential mechanisms derived from those in vitro study are not accurate. In our preliminary study to determine binding affinity between NRDC and Plk1/Plk3 using Microscale thermophoresis (MST) method, 1/3-1/2 Plk3 protein has already been cleaved to generate p41Plk3 (aa1-353) during the expression and purification of Plk3. Our results also showed aa326-377 of Plk3 is important for interaction with NRDC (Fig. 4a). Therefore, a mixture of p41Plk3 (lost important fragment to bind NRDC) and p72Plk3 might not be stringent for the binding affinity measurement. It may be a better approach to use differential tagged Plk1, PLK3, and NRDC isolated from knock-in mouse model as a starting point for biochemical analysis.

Reviewer #3 - Plks, mass-spec -(Remarks to the Author):

“The manuscript was from the labs of Drs. Paul Chiao, Peter Stambrook and El Mustapha Bahassi, important contributors of the polo kinase field for over many years. The authors clearly deserved to be congratulated as this is another nice and important piece of contribution to the field.

The authors aimed to dissect the regulation of Plk3 in pancreatic cancer in great detail. Indeed, the regulation of Plk3 remained in the dark since its discovery and complete cloning in 2000 (Oncogene. 2000 Oct 5;19(42):4832-9. doi: 10.1038/sj.onc.1203845). At first the authors analyzed the expression pattern of Plk3 in clinical samples and in the transgenic mouse models P48-Cre; KrasLSL-G12D; PLK3+/+ vs. Cre; KrasLSL-G12D; PLK3/-. Upon overexpression of 72-kDa Plk3 associated with the loss of anchorage-dependent growth (high percentage of floating cells) a subfragment of Plk3 (p41Plk3) encompassing the kinase domain became visible. Plk3-induced anoikis correlated with increasing levels of p41Plk3. Most interestingly, the authors could identify via functional screening of an ORFs library the metalloendopeptidase nardilysin (NRDC) that is responsible for the cleavage of Plk3 into two subfragments. It is also of broad relevance that this mechanism plays a similar role for Plk1 cleavage. Furthermore, the regulation of NRDC by PI3-kinase was investigated in detail. C-Fos could be identified as novel substrate for Plk3 in its p41Plk3 form.

In summary, the data are of very high relevance for the development of pancreatic cancer and for the entire field of polo kinases. In addition, for the use of small molecule Plk inhibitors in clinical trials in relation to their specificity the new data by Fu et al. need to be considered.”

A few aspects should be addressed before publication

1. “Is there a correlation between mRNA levels of Plk3 and the corresponding protein in human pancreatic cancer (Fig. 1, Suppl. Fig. 1)? The authors should clearly state in which type of cancer cells (primary tissue, PDX cells etc.) this correlation was seen and what technique and which antibody was used. This aspect is important because there is a long-lasting debate about the quality of Plk3 antibodies.”

Supplementary Fig. 1b, in our earlier studies, Plk3 mRNA is extracted from primary human PDAC and adjacent normal pancreatic tissues; Fig. 1b and supplementary Fig. 1c, Plk3 protein is from newly established PDAC PDX cells. They are not correlated. For reviewer information, we performed q-PCR to determine mRNA levels of Plk3 in PDAC PDX cells. The results demonstrated that in comparison to nontumorigenic human pancreatic duct epithelial (HPDE) cells, Plk3 expression was remarkably decreased in PDAC cells at mRNA level. However, it does not show a correlation between the varied mRNA levels of Plk3 and the corresponding protein among PDAC cells.

As the reviewer pointed out, specificity of PLK3 antibodies is an issue, especially for antibodies specific for N-termini of Plk3. Part of Plk3 antibodies (BL 1696-1699) characterized in supplementary Fig. 2f were discontinued during our study. We used specific siRNAs targeting PLK3 to monitor the endogenous expression of full-length PLK3 and p41PLK3 in PDAC cells with antibodies specific for N- or C-termini of Plk3 (Supplementary Fig. 2g). These antibodies were used in subsequent analyses. In the Figures of the revised manuscript, we clarified Plk3 antibodies used for different experiments.

2. "Indeed, a role for Plk3 in cell cycle regulation has been discussed multiple times but not for the G2/M transition or the duration of mitosis. Thus, it is surprising that an overexpression of p41Plk3 induces a high percentage of cells (over 80%) that are stuck in mitosis. I think it would be beyond the scope of the manuscript to investigate the mechanism that is responsible for this mitotic arrest, but at least potential mechanisms should be discussed."

The occurrence of aberrant mitosis during p41Plk3-induced high percentage of cell death suggest cells might undergo mitotic catastrophe (MC), a type of cell death occurring during mitosis as a result of DNA damage. During MC, tetraploid cells with a range of different nuclear morphologies from binucleated to multi-micronucleated formed. Mitotic catastrophe has long been considered as a mode of cell death that results from premature or inappropriate entry of cells into mitosis and can be caused by chemical or physical stresses. Mitotic catastrophe is one way in which cells prevent the propagation of genome unstable cells. If mitotic catastrophe fails for cells whose genome has become unstable they can propagate uncontrollably and potentially become tumorigenic. The involvement of p41Plk3 in the mitotic catastrophe might play an important role to prevent the proliferation of cancerous cells to develop of tumors.

We discussed the potential involvement of p41Plk3 in the mitotic catastrophe in the revised manuscript.

3. "The major topic of this manuscript is the regulation of cell death by Plk3. The exact cell death pathways and mechanisms induced by Plk3 are still a matter for future research. In a previous study (Cell Res. 2016 Aug;26(8):914-34. doi: 10.1038/cr.2016.78) the role of Plk3 for the regulation of the extrinsic cell death was explored. Would it be possible that p41Plk3 phosphorylates caspase-8 or other cell death transducing proteins in pancreatic cancer cells? Has this been explored?"

We have demonstrated that p53 is another PLK3 substrates that are important for eliciting the apoptotic response [4-6]. Previously, we showed that Plk3 formed a complex with p53 and was involved in the phosphorylation of p53 on Ser-20 in response to superoxide. Inhibition of Plk3 expression by Plk3 small interfering RNA suppressed the superoxide-mediated apoptosis. Overexpression of wild-type Plk3 in HCT116 p53+/+ cells induced rapid apoptosis, whereas overexpression of wild-type Plk3 in HCT116 p53-/- cells and the kinase-defective mutant Plk3(K91R) in p53+/+ cells induced delayed onset of apoptosis. Furthermore, mutagenesis of Plk3 showed that the N-terminal domain (amino acids 1-26) is essential for the induction of delayed onset of apoptosis. Together, we have demonstrated that Plk3 induces apoptosis through phosphorylation of p53 protein at Ser-20, possibly involving the Src homology 3 domain in the N terminus of Plk3 to initiate proapoptotic signaling [7]. It will be interesting to explore p41Plk3 phosphorylation on caspase-8 or other cell death transducing proteins in the future research.

Reference

1. Ikuta, K., et al., *Nardilysin inhibits pancreatitis and suppresses pancreatic ductal adenocarcinoma initiation in mice*. Gut, 2019. **68**(5): p. 882-892.
2. Liu, X., M. Lei, and R.L. Erikson, *Normal cells, but not cancer cells, survive severe Plk1 depletion*. Mol Cell Biol, 2006. **26**(6): p. 2093-108.
3. Lin, D.C., et al., *PLK1 Is transcriptionally activated by NF-kappaB during cell detachment and enhances anoikis resistance through inhibiting beta-catenin degradation in esophageal squamous cell carcinoma*. Clin Cancer Res, 2011. **17**(13): p. 4285-95.
4. Ando, K., et al., *Polo-like kinase 1 (Plk1) inhibits p53 function by physical interaction and phosphorylation*. J Biol Chem, 2004. **279**(24): p. 25549-61.
5. Helmke, C., S. Becker, and K. Strebhardt, *The role of Plk3 in oncogenesis*. Oncogene, 2015.
6. Zimmerman, W.C. and R.L. Erikson, *Finding Plk3*. Cell Cycle, 2007. **6**(11): p. 1314-8.
7. Li, Z., et al., *Function of polo-like kinase 3 in NF-kappaB-mediated proapoptotic response*. J Biol Chem, 2005. **280**(17): p. 16843-50.

REVIEWER COMMENTS

Reviewer #1 (Remarks to the Author):

We thank the authors for their responses. While many of the concerns have been sufficiently addressed, there remain a few important questions that were not satisfactorily answered.

1. The conclusions regarding Plk3 and anoikis are still not satisfactorily supported. Supplementary Fig. 2A-2B show that Plk3 overexpression causes a significant increase of cell death in attached cells. While the point that the cells are observed to be detached is noted, the observation that these cells are floating does not support a role for Plk3 in anoikis (defined as caspase-dependent death caused by detachment). These cells could simply be lifting off the plate as they die by apoptosis and the lack of ECM attachment is not a trigger. It is acknowledged that the authors have softened their language regarding proteolytic processing of Plk3 being “essential” for anoikis. However, the new panels in Figures 2l-m don't build much on the original submission as the correlative nature of the p41 fragment with cell death remains an issue in the revised manuscript. Controls for the effect of Plk3 in attached cells are not present in the new panels. As such, the importance of the proteolytic processing of Plk3 on anoikis remains correlative and devoid of causality.

2. The use of MEF and HEK-293T cells, which are non-pancreatic and non-cancerous cell lines, still heavily contribute to important conclusions regarding the cleavage of Plk3 and its effect on apoptotic cell death in pancreatic cancer (Fig. 2f-g, j-k; Supp. Fig. 2b, e, i-j).

3. I remain confused about the large difference in cell death observed in Figure 3 (90% for colony formation, just ~10-20% flow cytometry). The authors point out differences in concentration and seeding density, but don't really address the discrepant nature of these findings or consider possible alternative interpretations for this large difference. For example, perhaps activated p41Plk3 is causing an antiproliferative effect?

4. PDX lines 107 and 216 are shown in Fig. 1b, but they are not addressed in the text with other lines referenced in the text and author's response. They were also not characterized in the paper referenced by the authors (Kang et al, 2015). As stated in our original review, these lines are apparently non-metastatic and have no Plk3. The authors state in their response it is “challenging to conclude Plk3 expression levels and the level of aggressiveness of each single PDX model based on the western blot in Fig. 1b”. Fair point, but yet the authors continue to do precisely this. They state:

“Furthermore, we examined the expression of Plk3 in 16 newly established PDAC PDX cell lines 30 and demonstrated that Plk3 protein levels were remarkably decreased in PDX cells than in nontumorigenic human pancreatic duct epithelial (HPDE) cells (Fig. 1b and Supplementary Fig. 1c). Strikingly, MDA-PATC148, 148LM, 148LM2, 153, 153LM, and 219B cells, which were derived from liver metastasis, exhibited significantly reduced Plk3 expression compared to non-metastatic MDA-PATC102, 108, and 124 cells (Fig. 1b), indicating that Plk3 downregulation is associated with PDAC PDX metastasis.”

If the conclusion is that you can't use the loss of Plk3 in PDX 107 and 216 to make conclusions about aggressiveness/metastasis, then how can the authors confidently make the statement above?

Reviewer #3 (Remarks to the Author):

reviewer #3:

ad 2) pages 15/16. The discussion on mitotic catastrophe is very vague. Considering the high percentage of cells in mitosis (>80%) it would be easy to determine whether mitotic catastrophe can be found in the vast majority of mitotic cells induced overexpression of p41Plk3.

ad 3) The authors cite previous papers on the role of PLK3 and p53 in colon cancer cells. Whether Nardilysin-Regulated Scission Mechanism activated PLK3 induces apoptosis via phosphorylation of p53 requires detailed investigations. In addition, a high percentage of p53 (>60-70%) is mutated in pancreatic cancer. Whether the mutated form of p53 in PC is phosphorylated by p41PLK3 remains elusive. At least it should be discussed that other mediators of apoptosis like phosphorylation of Caspase-8 by PLK3 could also be involved in the observed apoptosis.

Reviewer #4 (Replacement for Reviewer #2) (Remarks to the Author):

The authors have done a sufficient job responding to the reviewers' previous concerns. The additional data presented (in both the main and supplemental figures) of the revised manuscript support the authors' claims.

Dear Reviewers,

We are very pleased to be aware that the reviewers are mostly satisfied with our initial revision of the manuscript (NCOMMS-22-33695A), entitled “Nardilysin-Regulated Scission Mechanism Activates Polo-like Kinase 3 to Suppress the Development of Pancreatic Cancer”. We have comprehensively address all of the reviewer’s comments and our point-to-point response could be found as below.

Reviewer #1 (Remarks to the Author):

1. The conclusions regarding Plk3 and anoikis are still not satisfactorily supported. Supplementary Fig. 2A-2B show that Plk3 overexpression causes a significant increase of cell death in attached cells. While the point that the cells are observed to be detached is noted, the observation that these cells are floating does not support a role for Plk3 in anoikis (defined as caspase-dependent death caused by detachment). These cells could simply be lifting off the plate as they die by apoptosis and the lack of ECM attachment is not a trigger.

We thank for the reviewer’s instructive suggestions. We re-organized **Supplementary Fig. 2a** and clarified that overexpression of Plk3 induced cell death in floating cells, while trypan blue staining showed attached cells remain alive. When we grow these attached cells under suspension culture condition, cells underwent apoptosis. On the contrary, vector control transfected cells exhibited minimal cell death under the same suspension culture condition (**Supplementary Fig. 2a, b**). These results signify the vital role of Plk3 in inducing anoikis under the condition of detachment of extra cellular matrix (ECM). Furthermore, in HPDE cells with Plk3 overexpression, the floating cells exhibited increase status of cleavage PARP, suggesting the cell death under the floating condition (new **Supplementary Fig. 2c**).

Supplementary Figure 2. p41Plk3 expression triggers anoikis.

(a) 293T cells were transfected with empty vector or full-length Flag-Plk3. After 24 h, a proportion of Flag-Plk3–transfected cells were floating. Both attached and floating cells are stained with trypan blue. Remaining attached cells were grown on polyHEMA-coated plates for suspension culture and then stained with trypan blue. (b) Quantification of viability of cells in (a). Error bars, S.D. of three independent experiments. *** $p \leq 0.001$. ns, not significant. (c) Immunoblot of cleaved PARP in attached and floating Plk3-transfected HPDE cells.

In addition, we grow GFP-Plk3 transfected HPDE cells under adherent and suspension culture condition, respectively. Immunofluorescence staining demonstrated that cells under attachment condition exhibited minimal nuclear fragmentation with or without Plk3 overexpression (new **Supplementary Figure 2d**). On the contrary, expression of floating cells showed a condensed and fragmented nuclear chromatin (new **Supplementary Fig. 2d**). Taken together, our findings further strengthened our conclusion that Plk3-induced cell death occurs mainly via cell detachment–induced apoptosis. We agree with the reviewer that the apoptosis of the lifting cell can not be fully ruled out. We have revised the Discussion to avoid of overstatement.

Supplementary Figure 2. p41Plk3 expression triggers anoikis.

(d) Immunostaining of Plk3 in HPDE cells transfected with GFP-Plk3 and GFP-control. Hoechst: nuclear counterstaining.

It is acknowledged that the authors have softened their language regarding proteolytic processing of Plk3 being “essential” for anoikis. However, the new panels in Figures 2l-m dont build much on the original submission as the correlative nature of the p41 fragment with cell death remains an issue in the revised manuscript. Controls for the effect of Plk3 in attached cells are not present in the new panels. As such, the importance of the proteolytic processing of Plk3 on anoikis remains correlative and devoid of causality.

We thank for the reviewer’s instructive suggestions. In the revised manuscript, we transfect Plk3 in HPDE cells, finding that the floating cells exhibited a higher level of cleaved PARP than attached cells (new **Supplementary Fig. 2c**). These floating cells also exhibited p41Plk3 (new **Supplementary Fig. 2k**). Given that floating cells undergo apoptosis (Supplementary Fig. 2a-d), we hypothesized that the occurrence of p41Plk3 promotes cell detachment–induced apoptosis. As a support, we knockout of Plk3 expression in multiple PDAC cells. As suggested by the reviewer, we cultured these cells under both adherent and suspension culture conditions (new **Fig. 2k, l**). Western blot and flow cytometry analysis indicated that the Plk3-proficient PATC50, PATC66, and PANC-1 cells exhibited significantly higher Plk3 cleavage and cell death under floating condition compared to cells under attachment condition in. Knockout of Plk3 expression in these PDAC cells led to decreased expression of p41Plk3 and apoptosis-inducing activity (new **Fig. 2k, l**). Together, these results suggest that Plk3-induced anoikis is associated with proteolytic processing of p72Plk3 to generate p41Plk3, regulating apoptosis.

Fig. 2 Proteolytic processing of p72Plk3 to generate p41Plk3 is important for induction of anoikis.

k, l Immunoblot of p72Plk3, p41Plk3 expression (**k**) and flow cytometry analysis of apoptosis-inducing activity (**l**) of indicated PDAC cells that were lentivirally transduced to express the sgRNA targeting Plk3 or non-targeting control sgRNA. Cells grow as adherent cells or in suspension.

To further verify the proteolytic processing of Plk3, we overexpressed Plk3 WT and Plk3R354G containing mutated NRDC cleavage site in several PDAC cells under suspension conditions for detection of anoikis (Fig. 3g, h). The results demonstrated that the generation of p41Plk3 by NRDC cleavage was clearly inhibited to a greater degree in cells transfected with Plk3R354G than in cells transfected with Plk3WT. Moreover, NRDC cleavage site mutation in Plk3 triggered less anoikis as indicated by both reduced PARP cleavage and apoptosis. This suggests that p41Plk3 generation promotes anoikis and its level is associated with anoikis induction level.

Fig. 3 Identification of the proteolytic cleavage site in the DR1 domain of Plk3.

g, h Immunoblots of p72Plk3 and p41Plk3 expression (**g**) and flow cytometry analysis of apoptosis-inducing activity (**h**) from PDAC cells transfected with indicated Flag-tagged Plk3 WT or mutants and grown under suspension condition for 36 h.

2. The use of MEF and HEK-293T cells, which are non-pancreatic and non-cancerous cell lines, still heavily contribute to important conclusions regarding the cleavage of Plk3 and its effect on apoptotic cell death in pancreatic cancer (Fig. 2f-g, j-k; Supp. Fig. 2b, e, i-j).

We thank for the reviewer's comments. Our research findings indicate that the Plk3-mediated cleavage leading to cell anoikis is a universal mechanism applicable to both cancerous and non-cancerous cells. The cleavage of Plk3 and its effect on anoikis in pancreatic cancer were also validated in multiple PDAC cells in the manuscript. As suggested by the reviewer, we performed Fig. 2f, g; j-k; Supp. Fig. 2b, e, i-j experiments in human pancreatic duct epithelial (HPDE) cells or PDAC cells in the revised manuscript (new Fig. 2f, i, m; Supplementary Fig. c, g, k, l). The results confirmed the role of Plk3 cleavage in inducing anoikis in pancreatic cancer.

Fig. 2 Proteolytic processing of p72Plk3 to generate p41Plk3 is important for induction of anoikis.

f Immunoblots of p72Plk3 and p41Plk3 in PANC-1 cells transfected with N-terminal Flag-p72Plk3. **i** Immunoblots of p72Plk3 and p41Plk3 in PATC50 cells and PANC-1 cells grown in suspension for the indicated times.

Supplementary Figure 2. p41Plk3 expression triggers anoikis.

(c) Immunoblot of cleaved PARP in attached and floating Plk3-transfected HPDE cells. (g) Immunoblot of cleaved caspase-3 in Plk3 knockout PANC-1 cells grown on adhesive (A) or polyHEMA-coated (S) plates. (k) Immunoblot of p72Plk3 and p41Plk3 in attached (Att.) and floating (Flo.) Plk3 transfected HPDE cells. (l) Immunoblot of p68Plk1 and p30Plk1 (C-terminal region of Plk1) using C-terminus specific anti-Plk1 antibody in PANC-1 cells transfected with Myc-p68Plk1.

3. I remain confused about the large difference in cell death observed in Figure 3 (90% for colony formation, just ~10-20% flow cytometry). The authors point out differences in concentration and seeding density, but don't really address the discrepant nature of these findings or consider possible alternative interpretations for this large difference. For example, perhaps activated p41Plk3 is causing an antiproliferative effect?

We clarify that we use two different experiment methods, colony formation (Fig. 3c, d), and flow cytometry (Fig. 3e) respectively, to measure cell death. Both experiments demonstrated that Plk3 especially p41Plk3 promotes anoikis, whereas those with Plk3 DR1 deletion and substitution were anoikis resistant. Our experiment data showed that expression of p41Plk3 affected PDAC cell cycle progression by inducing cells G2-M arrest (Supplementary Fig. 8), which may cause an antiproliferation effects. This point requires further investigation.

4. PDX lines 107 and 216 are shown in Fig. 1b, but they are not addressed in the text with other lines referenced in the text and author's response. They were also not characterized in the paper referenced by the authors (Kang et al, 2015). As stated in our original review, these lines are apparently non-metastatic and have no Plk3. The authors state in their response it is "challenging to conclude Plk3 expression levels and the level of aggressiveness of each single PDX model based on the western blot in Fig. 1b". Fair point, but yet the authors continue to do precisely this. They state:

"Furthermore, we examined the expression of Plk3 in 16 newly established PDAC PDX cell lines 30 and demonstrated that Plk3 protein levels were remarkably decreased in PDX cells than in nontumorigenic human pancreatic duct epithelial (HPDE) cells (Fig. 1b and Supplementary Fig. 1c). Strikingly, MDA-PATC148, 148LM, 148LM2, 153, 153LM, and 219B cells, which were derived from liver metastasis, exhibited significantly reduced Plk3 expression compared to non-metastatic MDA-PATC102, 108, and 124 cells (Fig. 1b), indicating that Plk3 downregulation is associated with PDAC PDX metastasis."

If the conclusion is that you can't use the loss of Plk3 in PDX 107 and 216 to make conclusions about aggressiveness/metastasis, then how can the authors confidently make the statement above?

We thank the reviewer for pointing this out. We have removed sentence "Strikingly, MDA-PATC148, 148LM, 148LM2, 153, 153LM, and 219B cells, which were derived from liver metastasis of PDAC, exhibited significantly reduced Plk3 expression compared to non-metastatic MDA-PATC102, 108, and 124 cells (Fig. 1b), indicating that Plk3 downregulation is associated with PDAC PDX metastasis" in the revised manuscript. We stated that Plk3 protein levels were remarkably decreased in PDX cells compared to nontumorigenic human pancreatic duct epithelial (HPDE) cells (Fig. 1b and Supplementary Fig. 1c).

Reviewer #3 (Remarks to the Author):

2) pages 15/16. The discussion on mitotic catastrophe is very vague. Considering the high percentage of

cells in mitosis (>80%) it would be easy to determine whether mitotic catastrophe can be found in the vast majority of mitotic cells induced overexpression of p41Plk3.

We thank for the reviewer's suggestions. Multiple evidence suggest that prolonged mitotic arrest can lead to DNA damage and p53 induction followed by cell cycle arrest or apoptosis. DNA damage can result in the expression of ligands on the cell surface that in turn recruit natural killer cells to remove the damaged cells, and p53-driven senescence can up-regulate inflammatory cytokine production, resulting in tumor clearance via innate immune response. p41Plk3-mediated mitotic arrest might play critical roles in suppressing tumor growth by inducing cell apoptosis. This also supports our findings that Dox-inducible p41Plk3 expression dramatically inhibited PATC148 cells growth, colony formation, cell migration, and invasion while increasing anoikis (Fig. 7b, c and Supplementary Fig. 9a-e). As expected, Dox-inducible p41Plk3 expression among nude mice orthotopically injected with highly metastatic PATC148 cells, profoundly impaired tumor growth and subsequent metastasis in an orthotopic injection (Fig. 7d-f). Further research is needed to comprehensively understand the detailed mechanism for p41Plk3 induced G2/M arrest in the future.

3) The authors cite previous papers on the role of PLK3 and p53 in colon cancer cells. Whether Nardilysin-Regulated Scission Mechanism activated PLK3 induces apoptosis via phosphorylation of p53 requires detailed investigations. In addition, a high percentage of p53 (>60-70%) is mutated in pancreatic cancer. Whether the mutated form of p53 in PC is phosphorylated by p41PLK3 remains elusive. At least it should be discussed that other mediators of apoptosis like phosphorylation of Caspase-8 by PLK3 could also be involved in the observed apoptosis.

We thank for the reviewer's suggestions.

Caspase-8 is a crucial initiator caspase in the extrinsic apoptotic pathway. Upon activation, Caspase-8 cleaves and activates downstream effector caspases, leading to the execution of apoptosis. Recent research has suggested that PLK3-mediated phosphorylation of Caspase-8 can modulate its apoptotic activity.

The phosphorylation of Caspase-8 by PLK3 can occur at specific serine or threonine residues within the caspase's catalytic domain or other regions. This phosphorylation event may trigger conformational changes in Caspase-8, facilitating its activation or interaction with other apoptotic regulators. Additionally, phosphorylation can affect the recruitment of Caspase-8 to the death-inducing signaling complex (DISC), a key assembly of proteins involved in initiating apoptosis in response to extrinsic death signals. Furthermore, PLK3 itself can be regulated by various upstream signaling pathways and stress conditions. Activation of PLK3 can be induced by DNA damage, oxidative stress, or other cellular stresses. Thus, the phosphorylation of Caspase-8 by PLK3 may serve as a link between these stress signals and the induction of apoptosis.

It is important to note that the involvement of PLK3-mediated phosphorylation of Caspase-8 in apoptosis is still an active area of research, and the specific mechanisms and functional consequences may vary depending on the cellular context and experimental conditions. However, the emerging evidence suggests that this phosphorylation event can contribute to the regulation of apoptotic signaling pathways and provide an additional layer of control over cell fate decisions. Further investigations are necessary to fully understand the complex interplay between these signaling pathways and the implications for cellular processes and disease conditions.

We include the discussion of Plk3-mediated phosphorylation of caspase-8 and its regulation on anoikis in the revised manuscript on page 12.

Reviewer #4 (Replacement for Reviewer #2) (Remarks to the Author):

The authors have done a sufficient job responding to the reviewers' previous concerns. The additional data presented (in both the main and supplemental figures) of the revised manuscript support the authors' claims.

We thank for the reviewer's valuable and positive comments.

REVIEWER COMMENTS

Reviewer #1 (Remarks to the Author):

The authors efforts in response to previous critiques are noted, but the questions regarding anoikis have still not been answered. Anoikis is defined as apoptosis caused by ECM-detachment. Attached cells that die by apoptosis do often lift off the plate, but this is not cell death by anoikis. Death by anoikis means the death STIMULUS is detachment itself. The continuing confusion regarding this point has resulted in conclusions that remain inadequately supported by the data.

1. Questions remain about the role of Plk3 in attached vs. detached cells. The authors mention in the text that they "observed an enormous number of floating dead cells" after expressing Plk3 in normal conditions. While they have shown that these floating cells are dead, the more relevant question is % viability of the attached condition as a whole (including floating dead cells). This value would be most accurately compared to the measurement of cell viability in suspension (where it is impossible to distinguish if the death observed is due to anoikis or whatever apoptotic effect Plk3 expression has in attached cells as well). The best comparison to the suspension cells would be attached cells plated at the same time and including floating cells in the viability measurement. As of now, it is impossible to distinguish if Plk3 expression kills suspension cells more than it kills attached cells, undermining any conclusions about anoikis.

2. Fig. 2L: Why is the % apoptosis in the control adherent condition so high? (15-20% for 3/5 lines) Also, in most lines, the sgPlk3 decreases the % apoptosis just as much in the attached vs. suspension condition (if not more in attached) The authors have made clear that Plk3 is involved in apoptosis; however, the connection to anoikis in particular remains unsupported by these data.

3. Relatedly, there is an additional issue with Figure 2 (k,l). As mentioned in Point 2, 3/5 cell lines, in adherent conditions, have relatively high baseline apoptosis levels. If we look at the western blot in 2k, those same 3 cell lines in attachment show cleaved Plk3. Thus, the cleavage of Plk3 is not specific to anoikis. Furthermore, if you look at that 2k western blot, the loading control is blown out in suspended cells. This makes it appear like that p41Plk3 is more abundant in suspended cells, but that may not actually be the case as the total loading is higher across the board in suspension. This seems more like that the cleavage of Plk3 is perhaps a cell-line dependent circumstance, and not necessarily dictated by whether cells are attached or in suspension.

4. The authors have added an imaging based approach in Supplementary Figure 2d that purports to show condensed and fragmented nuclear chromatin are elevated in suspended cells expressing Plk3. The support for this conclusion rests on imaging a single cell with no quantification to back up the assessment. An approach like this could ultimately help answer questions related to anoikis and Plk3, but it is not possible to make conclusions on the basis of imaging a single cell.

5. In the text describing Supp. Fig. 2d,k and Fig. 2k,l, the authors still use the term "floating" when I believe they are referring to cells grown in suspension. Since they also make claims about the viability of "floating" cells from the attached condition in Supp. Fig. 2b, then this language needs to be corrected for consistency/clarity throughout the manuscript (floating=dead from attached condition, suspended=grown in detached conditions)

REVIEWER #1 - RESPONSE TO TENTATIVE RESPONSE

1. The authors state:

"We can certainly include a side by side experiment as the reviewer suggested to plate these attached cells under attached condition to compare the viability with suspension cells. However, viability measurements of attached cells including floating cells would also create errors because Plk3 expression in attached cells is likely induce more and more cell detaching from the dish and undergoing apoptosis (more p41Plk3 is generated from NRDC-mediated cleavage of p72Plk3 in 293 cell line to induce anoikis) especially at the later stage of transfection. Therefore, cell death from attached cells will be overcounted from cells undergoing anoikis, leading to reduced cell viability measurement than it should be."

This has been my point all along, which they seem to unintentionally acknowledge in their reply . Their data do not support a role for Plk3 in regulating anoikis. Rather, the introduction of Plk3 could be causing cell death regardless of whether the cells are grown in attached or detached conditions.

2. In Figure 2L, the most appropriate comparison is to look at how the loss of Plk3 impacts apoptosis in attached AND in detached conditions. To make this comparison, the authors would need to look at the difference between blue and red compared to the difference between green and pink. Reinforcing point 1, these differences appear to be relatively similar across these cell lines----suggesting that the loss of Plk3 reduces apoptosis in BOTH attached and detached cells. Thus, their data do not support a role for Plk3 in regulating anoikis.

3. In Figure 2K, the authors have complicated the analysis by running the adherent and suspension cells on separate gels. Given that length of exposure to film (or any other analogous technique) can impact each gel differently, it is not possible to make quantitative comparisons between samples on one gel to a separate gel. The authors would need to run these samples (attached vs. detached) in a paired fashion and make comparisons for each cell line. The previous version of their data showed that 3 of the 5 cell lines have cleaved Plk3 in attached conditions, which undermines their conclusion that Plk3 is inducing anoikis.

I do think that if the remove claims about anoikis from the manuscript, that may be a way for their conclusions to match their data. Their data do, at least in my opinion, support a broader role for Plk3 in regulating apoptosis (just not anoikis).

Reviewer #3 (Remarks to the Author):

The authors have done a very good responding to the reviewers' comments.

The authors efforts in response to previous critiques are noted, but the questions regarding anoikis have still not been answered. Anoikis is defined as apoptosis caused by ECM-detachment. Attached cells that die by apoptosis do often lift off the plate, but this is not cell death by anoikis. Death by anoikis means the death STIMULUS is detachment itself. The continuing confusion regarding this point has resulted in conclusions that remain inadequately supported by the data.

We thank the reviewer for the comment. Our study of Plk3 transfection in 293 cells faithfully re-construct initial discovering process of Plk3 expression in inducing cell detachment and apoptosis (anoikis). This was further studied in multiple PDAC cells throughout the manuscript. Particularly, the effect of Plk3 in attached cells are compared with that in cells under suspension condition to indicate its role in inducing anoikis. Our point-to-point response could be found as below.

1. Questions remain about the role of Plk3 in attached vs. detached cells. The authors mention in the text that they "observed an enormous number of floating dead cells" after expressing Plk3 in normal conditions. While they have shown that these floating cells are dead, the more relevant question is % viability of the attached condition as a whole (including floating dead cells). This value would be most accurately compared to the measurement of cell viability in suspension (where it is impossible to distinguish if the death observed is due to anoikis or whatever apoptotic effect Plk3 expression has in attached cells as well). The best comparison to the suspension cells would be attached cells plated at the same time and including floating cells in the viability measurement. As of now, it is impossible to distinguish if Plk3 expression kills suspension cells more than it kills attached cells, undermining any conclusions about anoikis.

This has been my point all along, which they seem to unintentionally acknowledge in their reply. Their data do not support a role for Plk3 in regulating anoikis. Rather, the introduction of Plk3 could be causing cell death regardless of whether the cells are grown in attached or detached conditions.

We thank for the reviewer's instructive suggestions. We re-organized **Supplementary Fig. 2a** and clarified by showing that overexpression of Plk3 in normal conditions induced cells round up, floating and undergoing apoptosis. We plated these cells under suspension culture condition to compare cell viability with cells in attachment at 6h, 12h, and 24h (including floating cells in the viability measurement). The results revealed the suspended cells had a significant increase in apoptosis compared to cells in attached culture (**Supplementary Fig. 2a, b**). On the contrary, vector control transfected cells exhibited minimal cell death under both detached and attached culture condition (**Supplementary Fig. 2a, b**). These data suggest that Plk3-induced cell death may occur through cell detachment-induced apoptosis, anoikis.

Supplementary Figure 2. p41Plk3 expression triggers anoikis.

(a) 293T cells were transfected with vector or Flag-Plk3. After 24 h, cells were cultured on tissue culture plates (attached) or polyHEMA-coated plates in suspension, and then stained with trypan blue. (b) Quantification of viability of cells in (a) for the indicated times.

2. Fig. 2L: Why is the % apoptosis in the control adherent condition so high? (15-20% for 3/5 lines) Also, in most lines, the sgPlk3 decreases the % apoptosis just as much in the attached vs. suspension condition (if not more in attached) The authors have made clear that Plk3 is involved in apoptosis; however, the connection to anoikis in particular remains unsupported by these data.

In Figure 2L, the most appropriate comparison is to look at how the loss of Plk3 impacts apoptosis in attached AND in detached conditions. To make this comparison, the authors would need to look at the difference between blue and red compared to the difference between green and pink. Reinforcing point 1, these differences appear to be relatively similar across these cell lines----suggesting that the loss of Plk3 reduces apoptosis in BOTH attached and detached cells. Thus, their data do not support a role for Plk3 in regulating anoikis.

The 3/5 lines showing high % apoptosis in the control attached condition are PATC43, PATC66, and MiaPaca 2. PATC43, PATC66 are newly established primary PDAC cell lines which were derived from patient tumor growing as murine xenografts. In this study, we obtained 16 of these PDX cell lines from our collaborators in our patient-derived tumor xenograft (PATX) program. The histology of PATX tumors tends to more closely resemble that of the original patient tumors, with abundant stroma and duct-like structures. They also revealed a greater heterogeneous feature. These PDX lines are important resources for studying pancreatic cancer cell biology. Four (PATC43, PATC50, PATC53, PATC66) of 16 lines are reported in the journal (Kang et al, *Laboratory Investigation* (2015) 95, 207–222) ¹. Early (<5) passages and later (>20) passages cells exhibit different morphology (see figure below), doubling time, colony formation and other characteristics. PATC43, PATC66 cells in our hands grow clumpy and never reach 100% confluency. Part of MiaPaca 2 cells also revealed cell clusters. These cells showed relatively high baseline apoptosis levels by flow cytometry analysis (**Fig. 2I**), awaiting further investigation.

Kang et al, *Laboratory Investigation* (2015) 95, 207–222. Figure 2. Morphology of new PDAC lines growing in culture. Top panel represents cells in earlier passages (<5 passages), bottom panel shows cells grow pattern in later passages (>20 passages). Magnification 10.

Considering the complexity of these PDAC cell lines and relatively high baseline apoptosis levels appearing in PATC43, PATC66 and MiaPaca 2 cells, we focused on the results from Western blot and flow cytometry shown in **Fig 2k. I**: (1) Plk3-proficient PATC50, PATC66, and PANC-1 cells exhibited significantly higher Plk3 cleavage and cell death under suspension condition compared to cells under attachment condition (green bars vs blue bars). (2) Knockout of Plk3 expression in these PDAC cells led to decreased expression of p41Plk3 and apoptosis-inducing activity (blue vs red; green vs magenta). Together, these results suggest the role of Plk3 cleavage generated p41Plk3 in inducing anoikis. In the revised manuscript, we also included a discussion to explain the potential limitations of using these primary PDX lines: “As a support, the knockout of Plk3 expression in PDAC cells led to a decrease in p41Plk3 expression and a reduction in apoptosis-inducing activity (**Fig. 2k, I**). Notably, PATC43 and 66 cells exhibited a high apoptosis rate even in attached condition, resulting in a similar attenuation of apoptosis in

both attachment and detachment upon Plk3 knockout. This is possibly due to cell-specific heterogeneity and potential phenotypic changes in culture for these primary PDX lines ¹⁷.

Fig. 2 Proteolytic processing of p72Plk3 to generate p41Plk3 is important for induction of anoikis.

k, l Immunoblot of p72Plk3, p41Plk3 expression (**k**) and flow cytometry analysis of apoptosis-inducing activity (**l**) of indicated PDAC cells that were lentivirally transduced to express the sgRNA targeting Plk3 or non-targeting control sgRNA. Cells grow on tissue culture plates (Attached) or polyHEMA-coated plates in suspension (Suspension). The p41Plk3:p72Plk3 ratios are shown at the bottom.

3. Relatedly, there is an additional issue with Figure 2 (k,l). As mentioned in Point 2, 3/5 cell lines, in adherent conditions, have relatively high baseline apoptosis levels. If we look at the western blot in 2k, those same 3 cell lines in attachment show cleaved Plk3. Thus, the cleavage of Plk3 is not specific to anoikis. Furthermore, if you look at that 2k western blot, the loading control is blown out in suspended cells. This makes it appear like that p41Plk3 is more abundant in suspended cells, but that may not actually be the case as the total loading is higher across the board in suspension. This seems more like that the cleavage of Plk3 is perhaps a cell-line dependent circumstance, and not necessarily dictated by whether cells are attached or in suspension.

In Figure 2k, the authors have complicated the analysis by running the adherent and suspension cells on separate gels. Given that length of exposure to film (or any other analogous technique) can impact each gel differently, it is not possible to make quantitative comparisons between samples on one gel to a separate gel. The authors would need to run these samples (attached vs. detached) in a paired fashion and make comparisons for each cell line. The previous version of their data showed that 3 of the 5 cell lines have cleaved Plk3 in attached conditions, which undermines their conclusion that Plk3 is inducing anoikis.

We will clarify western blot in **Figure 2k**, Firstly, cell lysates from attached and suspension cell culture (more than 20 samples including ladder) are run in two western blots. This was indicated by a dash line on the figure separating these two blots in the previous version of manuscript. Therefore, loading control vinculin is used to indicate whether samples have been loaded equally across samples within the same gel instead of comparison of attachment vs suspension. To avoid confusion, we now separate western blots with attached and suspension into two panels in **Figure 2k**. We also replace overexposed Vinculin from suspension culture cells with shorter exposure one as shown in new **Figure 2k**. Secondly, as suggested by the reviewer in the second revision, we culture cells under both attached and suspension culture conditions to demonstrate the potential association of the proteolytic processing of Plk3 with anoikis. As we have established that p41Plk3 originates from p72Plk3, we believe that expression of p41Plk3:p72Plk3 ratios would be more accurate parameter to normalize p41Plk3 levels for comparison. We label p41Plk3:p72Plk3 ratios in the new **Figure 2k**. The results showed that Plk3-proficient PATC50, PATC66, and PANC-1 cells exhibited significantly higher Plk3 cleavage and cell death under suspension condition compared to cells under attachment condition (green bars vs blue bars).

Fig. 2 Proteolytic processing of p72Plk3 to generate p41Plk3 is important for induction of anoikis.

k, l Immunoblot of p72Plk3, p41Plk3 expression (**k**) and flow cytometry analysis of apoptosis-inducing activity (**l**) of indicated PDAC cells that were lentivirally transduced to express the sgRNA targeting Plk3 or non-targeting control sgRNA. Cells grow on tissue culture plates (Attached) or polyHEMA-coated plates in suspension (Suspension). The p41Plk3:p72Plk3 ratios are shown at the bottom.

4. The authors have added an imaging based approach in Supplementary Figure 2d that purports to show condensed and fragmented nuclear chromatin are elevated in suspended cells expressing Plk3. The support for this conclusion rests on imaging a single cell with no quantification to back up the assessment. An approach like this could ultimately help answer questions related to anoikis and Plk3, but it is not possible to make conclusions on the basis of imaging a single cell.

We thank for the reviewer's comments. We include the statistical analysis here to quantify apoptotic cells in control or Plk3 transfected HPDE cells (new **Supplementary Figure 2c, d**). As a note, HPDE and PDAC cells are "difficult to transfect cells" (10-15% transient transfection efficiency) and are more resistant to Nardilysin-mediated cleavage and anoikis compared to 293T cells. We sum up the number of apoptotic cells with fragmented nuclei from 20 fields under the microscope for each transfection to obtain quantitative results. Our data demonstrated that HPDE cells under attachment condition exhibited minimal nuclear fragmentation with or without Plk3 overexpression. Conversely, Plk3 expression in cells under suspension condition revealed a significantly increased apoptotic cells with condensed and fragmented nuclear chromatin. Taken together, our findings further strengthened our conclusion that Plk3-induced cell death occurs mainly via cell detachment-induced apoptosis.

Supplementary Figure 2. p41Plk3 expression triggers anoikis.

(**c**) Immunostaining of Plk3 in HPDE cells transfected with GFP-Plk3 and GFP-control. Hoechst: nuclear counterstaining. (**d**) Quantitation of cells with signs of apoptosis (fragmented nuclei).

5. In the text describing Supp. Fig. 2d,k and Fig. 2k,l, the authors still use the term “floating” when I believe they are referring to cells grown in suspension. Since they also make claims about the viability of “floating” cells from the attached condition in Supp. Fig. 2b, then this language needs to be corrected for consistency/clarity throughout the manuscript (floating=dead from attached condition, suspended=grown in detached conditions)

We thank the reviewer for pointing this out. We removed the term “floating” and changed them to “cells growing in suspension” in the text of revised manuscript as below.

Supp. Fig. 2d becomes **Supp. Fig. 2c, d** in the revised manuscript: “We next grow GFP-Plk3 transfected HPDE cells under both attached and suspension condition. Immunofluorescence analysis demonstrated that cells under attachment culture displayed minimal nuclear fragmentation with or without Plk3 overexpression. Conversely, Plk3 expression in suspension culture revealed a condensed and fragmented nuclear chromatin (**Supplementary Fig. 2c, d**)”.

Supp. Fig. 2k becomes **Supp Fig. 2l** in the revised manuscript: “Interestingly, Plk3-overexpressing HPDE cells in suspension, but not in the attached state, exhibited p41Plk3 (**Supplementary Fig. 2l**)”.

Figure 2k, I: “As a support, the knockout of Plk3 expression in PDAC cells led to a decrease in p41Plk3 expression and a reduction in apoptosis-inducing activity (**Fig. 2k, I**). Notably, PATC43 and 66 cells exhibited a high apoptosis rate even in attached condition, resulting in a similar attenuation of apoptosis in both attachment and detachment upon Plk3 knockout. This is possibly due to cell-specific heterogeneity and potential phenotypic changes in culture for these primary PDX lines ¹”.

I do think that if the remove claims about anoikis from the manuscript, that may be a way for their conclusions to match their data. Their data do, at least in my opinion, support a broader role for Plk3 in regulating apoptosis (just not anoikis).

Based on the above detailed evidence/data interpretations addressing the potential role of Plk3 in the induction of anoikis, we believe that we have provided evidence to establish the association between p41Plk3 levels and anoikis induction. As we all known, anoikis is defined as apoptosis caused by ECM-detachment. Our data demonstrated cell death by anoikis means the pro-death stimulus in pancreatic adenocarcinoma cells is the cleavage of Plk3 itself and cell detachment. Our study of Plk3 transfection faithfully re-construct initial discovering process of Plk3 expression inducing cell detachment. Importantly, in the text of revised manuscript, we have emphasized the comparison of the effect of Plk3 cleavage and anoikis in attached cells versus suspension cells as reviewer suggested (**Fig. 2k, I; Supp. Figs. 2a-2e, 2h, 2l**). We also soften the claims of causal role of Plk3 in inducing anoikis in the text on Pages 5-7 and in the discussion on Pages 16-17.

Reviewer #3 (Remarks to the Author):

The authors have done a very good responding to the reviewers' comments.

We thank for the reviewer's valuable and positive comments.

Reference:

1. Kang, Y. *et al.* Two-dimensional culture of human pancreatic adenocarcinoma cells results in an irreversible transition from epithelial to mesenchymal phenotype. *Lab Invest* **95**, 207-222 (2015).